# FROM NOISE TO FACTORS: DIFFUSION-BASED UNSUPERVISED SEQUENTIAL DISENTANGLEMENT

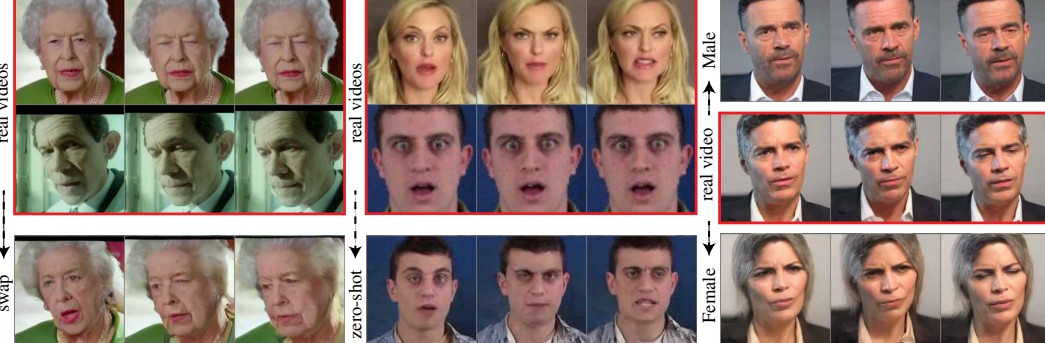

Figure 1: We present swap (left), zero-shot (middle), and multifactor disentanglement (right) results on multiple real-world and high-resolution visual datasets. See Sec. 5 for further details.

## ABSTRACT

Unsupervised representation learning, in particular, sequential disentanglement, where the goal is to learn disentangled static and dynamic factors of variation, remains a significant challenge due to the absence of labels. Existing models, based on variational autoencoders and generative adversarial networks, achieved success in certain domains, but they often struggle with disentangling sequences, especially when dealing with real-world complexity and variability. Further, there is no real-world evaluation protocol for assessing the effectiveness of sequential disentanglement models. Recently, diffusion autoencoders have emerged as a new promising generative model, offering semantically rich representations by gradual noise-to-data transformations. Despite their advantages, these models face limitations: they are non-sequential, fail to disentangle the latent space effectively, and are computationally intensive, making them difficult to scale to sequences. In this work, we introduce our diffusion sequential disentanglement autoencoder (*DiffSDA*), a novel approach effective on real-world visual data and accompanied by a new and challenging evaluation protocol. DiffSDA is based on a new probabilistic modeling and is implemented using latent diffusion models and efficient samplers, facilitating processing of high-resolution videos. We test our approach on several real-world datasets and metrics, and we demonstrate its effectiveness in comparison to recent state-of-the-art sequential disentanglement methods.

## 1 INTRODUCTION

As deep learning models progress, the need for extensive labeled data made unsupervised learning crucial (Bengio et al., 2012). Within unsupervised learning, *disentangled representation learning* is particularly important (Bengio et al., 2013). This approach aims to factorize latent encodings from data so that each factor represents a distinct variation, enhancing explainability (Liu et al., 2020), reducing biases (Creager et al., 2019), and improving generalizability (Zhang et al., 2022). A key challenge is modeling sequential data, such as videos, where latent encodings are split into static and dynamic factors. For example, in a video of a person talking, facial appearance is the static factor, and facial motions encode the dynamic factors. This paper focuses on disentangling and evaluating real-world, high-quality visual sequential information in an unsupervised manner.

Most existing sequential disentanglement works, e.g., (Tulyakov et al., 2018; Yingzhen & Mandt, 2018; Bai et al., 2021; Han et al., 2021; Naiman et al., 2023) are based on variational autoencoders (VAEs) (Kingma, 2013) and generative adversarial networks (GANs) (Goodfellow et al., 2014). Unfortunately, processing real-world data is challenging for VAEs as they produce blurry images and although several approaches have been proposed to address this issue (Razavi et al., 2019; Vahdat & Kautz, 2020), these methods often result in a significantly larger hierarchical latent space, potentially complicating disentanglement. In practice, state-of-the-art sequential disentanglement models struggle to capture the complexities of real-world, high-quality datasets, and thus, they mostly demonstrate their applicability on simple, often toy, examples, see Fig. 3. Another related limitation is the lack of a real-world benchmark. Such absence goes beyond the current evaluation on simple data—there is no standardized, robust, and reliable evaluation protocol involving real-world, high-quality visual sequential data and unsupervised metrics. We advocate that improving these two objectives, namely, *designing new models* and *proposing new evaluation protocols* for real-world data and tasks may lead to fundamental advances in unsupervised sequential disentanglement.

One approach for improving real-world capabilities of sequential disentanglement techniques is to incorporate them within diffusion models (Sohl-Dickstein et al., 2015), which have been showing superior results over GANs' and VAEs' visual generative quality (Ho et al., 2020; Preechakul et al., 2022). Unfortunately, the latent variables of diffusion models lack a semantic structure that is crucial for disentangled learning. Consequently, recent work has extended diffusion models to have an autoencoder structure, facilitating the learning of meaningful representations (Preechakul et al., 2022; Wang et al., 2023). However, diffusion autoencoder (DiffAE) models can not be used for unsupervised sequential disentanglement directly due to three main limitations. First, these models were primarily designed for non-sequential information. Second, while semantic information becomes available in DiffAE, it is not factorized to separate disentangled factors. Finally, existing DiffAE works are computationally restrictive, requiring many resources to process high-quality and high-resolution data.

We address the first objective above by proposing a novel *diffusion sequential disentanglement autoencoder (DiffSDA)* that extends DiffAE works by alleviating their challenges. We improve the first two issues by basing our encoder on a new probabilistic disentanglement model that has no constraints on the prior distribution of the static and dynamic latents, unlike previous models. This flexibility facilitates learning of expressive representations, it significantly reduces the number of required hyper-parameters, and it is easy to code using vanilla recurrent modules and a U-Net (Ronneberger et al., 2015). To alleviate the third challenge, our decoder exploits the recent advances on stochastic differential equations (SDEs) (Song et al., 2021) and diffusion models, supporting training via a simple score matching loss. In addition, we adapt an efficient and lightweight sampling framework known as EDM (Karras et al., 2022) to our setup, allowing fast sampling with only 63 network evaluations. Finally, to process high-resolution visual sequential data with reasonable computational requirements, we incorporate a latent diffusion module (LDM) (Rombach et al., 2022) into our pipeline. Overall, we develop an SDE-based LDM-EDM decoder.

Toward addressing the second objective, we introduce a new *evaluation protocol for visual sequential disentanglement* models. Inspired by the existing quantitative and qualitative evaluation standard protocol (Bai et al., 2021), we adopt current datasets and metrics, as well as introduce new components. For data, we use three real-world and high-resolution visual datasets that have not been previously used for sequential disentanglement. For metrics, we utilize qualitative conditional and unconditional swap tasks. We also measure quantitatively by borrowing estimators from animation for assessing if objects and motions are preserved and suggest a new unsupervised swap metric. Further, we propose a new test for assessing multifactor disentanglement capabilities. Finally, we suggest a new realistic zero-shot test case where DiffSDA is trained on one dataset but evaluated on unseen data. Through extensive tests, we show that DiffSDA disentangles real-world data well while outperforming recent SOTA approaches. Our contributions can be summarized as follows:

1. We introduce DiffSDA, a novel diffusion sequential disentanglement autoencoder model, based on a new probabilistic bias and implemented using common neural modules.
2. We extend current disentanglement evaluation protocols to include real-world, high-quality visual data, metrics, and new multifactor exploration and zero-shot tests.
3. We extensively demonstrate our model's superiority in qualitative and quantitative disentanglement tasks in comparison to existing SOTA models.

## 2 RELATED WORK

**Disentangled representation learning** is a long-standing problem (Bengio et al., 2013). Many works proposed variational autoencoder models for non-sequential (Higgins et al., 2017; Chen et al., 2018; Kim & Mnih, 2018) and sequential (Hsu et al., 2017; Yingzhen & Mandt, 2018; Zhu et al., 2020; Bai et al., 2021; Han et al., 2021; Naiman et al., 2023; Berman et al., 2024; Simon et al., 2025) information. However, a significant drawback of VAE-based approaches is their low reconstruction quality, often resulting in blurry images when applied to real-world data (Kingma et al., 2016; Berg et al., 2018; Vahdat & Kautz, 2020; Bredell et al., 2023). Another research direction has focused on GANs for disentanglement with non-sequential (Tran et al., 2017; Karras et al., 2020; Ren et al., 2021) and sequential (Villegas et al., 2017; Tulyakov et al., 2018) approaches. Unfortunately, GAN-based techniques are challenging to train and they suffer from learned distributions with insufficient expressiveness, exhibiting mode collapse issues (Goodfellow, 2016; Lucic et al., 2018). Currently, most sequential disentanglement approaches present factorized results on simple real-world datasets that are far from in-the-wild, with the exception of some preliminary results in SPYL (Naiman et al., 2023). Recently, the rise of diffusion models has led to new non-sequential disentanglement approaches (Kwon et al., 2022; Yang et al., 2023; Wang et al., 2023; Yang et al., 2024; Zhu et al., 2024; Baumann et al., 2024), producing high-resolution images with high-quality disentangled factors. However, to the best of our knowledge, we are the first to suggest a two-factor (i.e., static and dynamic) diffusion sequential disentanglement model that is evaluated on a real-world and high-resolution visual benchmark. Finally, a related body of works in animation (Siarohin et al., 2019; Hu, 2024; Xu et al., 2024), which leverages video priors for disentangling object and movement, can be applied to similar swapping tasks as demonstrated in Sec. 5.1 and Sec. 5.2. However, our method is designed for general sequential data, as it also extends to audio Sec.5.5 and also enables the learning of multifactor disentangled representations, as shown in Sec. 5.3

**Diffusion Models** (Sohl-Dickstein et al., 2015) and score matching (Hyvärinen & Dayan, 2005; Vincent, 2011) have emerged as strong alternatives to VAEs and GANs (Ho et al., 2020; Dhariwal & Nichol, 2021). They excel in generating high-quality images through iterative denoising of latent variables and are unified in a score-based modeling framework (Song et al., 2021). However, diffusion models construct non-semantic latent codes, and thus, recent efforts have focused on structuring their latent representations. For instance, DiffAE (Preechakul et al., 2022) design an autoencoder to manipulate visual features, whereas InfoDiffusion (Wang et al., 2023) adds loss regularizers. Still, these approaches are not designed to handle sequences: they do not produce static and dynamic factors and are computationally demanding due to high-dimensional latents and long sampling. Recent advances in diffusion models have partially addressed their computational representation and sampling shortcomings. In particular, latent diffusion models (LDM) (Rombach et al., 2022) design their denoising process in a small latent space, significantly reducing computational time and memory requirements. Finally, multiple approaches focused on improving diffusion samplers (Song et al., 2023; Song & Dhariwal, 2024), with EDM (Karras et al., 2022; 2024) standing out as an efficient and lightweight framework with fast sampling times and high generation quality.

## 3 BACKGROUND

### 3.1 DIFFUSION MODELS

Diffusion models (Sohl-Dickstein et al., 2015) are a family of SOTA generative models, that were recently described using stochastic differential equations (SDEs), diffusion processes, and score-based modeling (Song et al., 2021). We will use diffusion models and score-based models interchangeably. These models include two processes: the forward process and the reverse process. The forward process (often not learnable) is an iterative procedure that corrupts the data by progressively adding noise to it. Specifically, the infinitesimal change to the state $\mathbf{x}_t$ can be formally described by

$$d\mathbf{x}_t = \mathbf{f}(\mathbf{x}_t, t)dt + g(t)d\mathbf{w} , \tag{1}$$

where $\mathbf{w}$ is the standard Wiener process, $\mathbf{f}(\cdot, t)$ is a vector-valued function called the drift coefficient, and $g(\cdot)$ is a scalar function known as the diffusion coefficient. From a probabilistic viewpoint, Eq. 1 is associated with modeling the transition from the given data distribution, $\mathbf{x}_0 \sim p_0$, to $p_t$, the probability density of $\mathbf{x}_t$, $t \in [0, T]$. Typically, the prior distribution $p_T$ is a simple Gaussian

distribution with fixed mean and variance that contains no information of $p_0$. The reverse process, which is learnable, de-noises the data iteratively. The reverse of a diffusion process is also a diffusion process, depending on the score function $\nabla_{\mathbf{x}} \log p_t(\mathbf{x})$ and operating in reverse time (Anderson, 1982). In our approach, we utilize the conditioned reverse process

$$d\mathbf{x}_t = [\mathbf{f}(\mathbf{x}_t, t) - g(t)^2 \nabla_{\mathbf{x}} \log p_t(\mathbf{x}_t \mid \mathbf{u})]d\bar{t} + g(t)d\bar{\mathbf{w}}, \tag{2}$$

where $\bar{\mathbf{w}}$ is a standard Wiener process as time progresses backward from $T$ to $0$, $d\bar{t}$ is an infinitesimal negative timestep, and $\mathbf{u}$ is a condition variable. Diffusion models are generative by sampling from $p_T$ and use $\nabla_{\mathbf{x}} \log p_t(\mathbf{x}_t \mid \mathbf{u})$ to iteratively solve Eq. 2 until samples from $p_0$ are recovered.

## 3.2 DIFFUSION AUTOENCODERS

Although diffusion models are powerful generative tools, they are not inherently designed to learn meaningful representations of the data. To address this limitation, several works (Preechakul et al., 2022; Wang et al., 2023) have adapted diffusion models into autoencoders, resulting in diffusion autoencoders (DiffAEs). These models have demonstrated the ability to learn semantic representations of the data, allowing certain modifications of the resulting samples by altering their latent vectors. To this end, DiffAEs introduce a semantic encoder, taking a data sample $x_0$ and returning its semantic latent encoding $z_{\text{sem}}$. Then, the latter vector conditions the reverse process, enhancing the model's ability to reconstruct and manipulate data samples. In practice, the denoiser is also conditioned on a feature map $h$ and the time $t$, combined using an adaptive group normalization (AdaGN) layer (Dhariwal & Nichol, 2021). The AdaGN block is defined as

$$\text{AdaGN}(h, t, z_{\text{sem}}) = z_s \left( t_s \, \text{GroupNorm}(h) + t_b \right), \tag{3}$$

where $z_s$ is the output of a linear layer applied to $z_{\text{sem}}$, $t_s$ and $t_b$ are the outputs of a multi-layer perceptron (MLP) applied to the time $t$, and multiplications are done element-wise.

## 4 METHOD

In this section and the subsequent ones, the subscripts represent time in the diffusion process, and superscripts indicate time in the sequence, e.g., a sequence state of the diffusion process is denoted by $\mathbf{x}_t^\tau$, $t \in [0, T]$ and $\tau \in \{1, \ldots, V\}$. $T$ and $V$ represent the maximum diffusion and sequence times, respectively. In particular, note that we consider discrete time sequences of continuous time diffusion processes; however, our modeling can be extended to additional settings.

### 4.1 PROBABILISTIC MODELING

To improve real-world capabilities of existing sequential disentangling tools, we propose a diffusion-based sequential disentanglement model, harnessing the recent advances in generative diffusion modeling. Alas, vanilla diffusion models do not naturally learn semantic latent variables, which was recently addressed by diffusion autoencoders. In what follows, we extend DiffAE to the sequential setting, incorporating explicitly the disentangled static and dynamic factors into our modeling. Our generative disentanglement method is based on two diffusion models. The first model details the state-independent distribution density of the static (time-invariant) and dynamic (time-variant) factors, $s_0$ and $d_0^{1:V}$, respectively. The second model specifies the state distribution and its dependence on the disentangled factors. Formally, the joint distribution is given by

$$p(\mathbf{x}_0^{1:V}, \mathbf{x}_T^{1:V}, s_0, s_T, d_0^{1:V}, d_T^{1:V}) = p_{T0}(s_0, d_0^{1:V} \mid s_T, d_T^{1:V}) \prod_{\tau=1}^{V} p_{T0}(\mathbf{x}_0^\tau \mid \mathbf{x}_T^\tau, s_0, d_0^\tau), \tag{4}$$

where $p_{T0}(s_0, d_0^{1:V} \mid s_T, d_T^{1:V})$ is a standard diffusion process with $p_{T0}(\cdot)$ being the transition distribution from time $T$ to time $0$. The state distribution of $p_{T0}(\mathbf{x}_0^\tau \mid \mathbf{x}_T^\tau, s_0, d_0^\tau)$ is conditioned on the latent $\mathbf{x}_T^\tau$ and the factors $s_0$ and $d_0^\tau$.

Importantly, our probabilistic approach differs from existing work (Bai et al., 2021; Naiman et al., 2023) in that our static and dynamic factors are dependent on one another. We motivate our model by three main reasons: i) expressiveness—the overall dependence facilitates learning of different state trajectories, leading to higher expressivity in the marginals $p_{t0}(\cdot)$; and ii) efficiency—our sampler

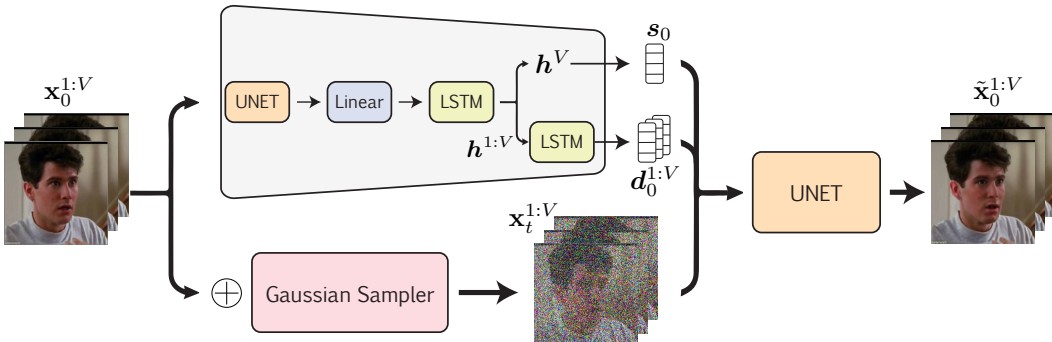

Figure 2: *DiffSDA* processes sequences $\mathbf{x}_0^{1:V}$ via semantic and stochastic encoders (top and bottom). Their outputs $(\boldsymbol{s}_0, \boldsymbol{d}_0^{1:V}, \mathbf{x}_t^{1:V})$ are fed to a stochastic decoder yielding a denoised $\tilde{\mathbf{x}}_0^{1:V}$ (right).

is not autoregressive, allowing for fast and parallelized sampling; and iii) causality—our model has the ability to learn intricate relationships between the static and dynamic factors, if needed.

Given the data sequence $\mathbf{x}_0^{1:V} \sim p_0(\mathbf{x}_0^{1:V})$, the posterior distribution of the latent variables $\mathbf{x}_t^{1:V}$ and latent factors $\boldsymbol{s}_0$ and $\boldsymbol{d}_0^{1:V}$ is composed of three independent distributions. Further, unlike the non-autoregressive prior in Eq. 4, here, we explicitly assume temporal dependence. The posterior distribution reads

$$p(\mathbf{x}_t^{1:V}, \boldsymbol{s}_0, \boldsymbol{d}_0^{1:V} \mid \mathbf{x}_0^{1:V}) = p_{0t}(\mathbf{x}_t^{1:V} \mid \mathbf{x}_0^{1:V}) p(\boldsymbol{s}_0 \mid \mathbf{x}_0^{1:V}) \prod_{\tau=1}^{V} p(\boldsymbol{d}_0^{\tau} \mid \boldsymbol{d}_0^{<\tau}, \mathbf{x}_0^{\leq\tau}) , \qquad (5)$$

where $\mathbf{x}_t^{1:V}$ and $\boldsymbol{s}_0$ are conditioned on the entire data sequence $\mathbf{x}_0^{1:V}$, and the dynamic factors only depend on previous dynamic factors and current and previous data elements. To optimize the above probabilistic model, we employ score matching (Hyvärinen & Dayan, 2005; Song et al., 2021), minimizing for the denoising parametric map $\boldsymbol{D}_\theta$. The map $\boldsymbol{D}_\theta$ takes the noisy latent $\mathbf{x}_t^{\tau}$, time $t$, and disentangled factors $\mathbf{z}_0^{\tau} := (\boldsymbol{s}_0, \boldsymbol{d}_0^{\tau})$, and it returns an estimate of the score function $\nabla_{\mathbf{x}} \log p_{0t}(\mathbf{x}_t^{\tau} \mid \mathbf{x}_0^{\tau})$. Overall, the optimization objective reads

$$\boldsymbol{\theta}^* = \arg\min_{\boldsymbol{\theta}} \mathbb{E}_t \Big\{ \lambda_t \mathbb{E}_{\mathbf{x}_t^{\tau}, \mathbf{z}_0^{\tau}, \mathbf{x}_0^{\tau}} \big[ \|\boldsymbol{D}_\theta(\mathbf{x}_t^{\tau}, t, \mathbf{z}_0^{\tau}) - \nabla_{\mathbf{x}} \log p_{0t}(\mathbf{x}_t^{\tau} \mid \mathbf{x}_0^{\tau}) \|_2^2 \big] \Big\} , \qquad (6)$$

where $\lambda_t \in \mathbb{R}^+$ is a positive weight, $t \sim \mathcal{U}[0, T]$ is uniformly sampled over $[0, T]$, the variables $\mathbf{x}_t^{\tau}$, $\mathbf{x}_0^{\tau}$ are sampled from their respective distributions, $p_{0t}(\cdot), p_0(\cdot)$, and $\mathbf{z}_0^{\tau}$ via the densities in Eq. 5. Importantly, $p_{T0}$ of $\boldsymbol{s}_0, \boldsymbol{d}_0^{1:V}$ is not used in Eq. 6, and thus its optimization can be separated.

## 4.2 DIFFUSION SEQUENTIAL DISENTANGLEMENT AUTOENCODER

Below, we detail our architectural contributions in extending DiffAE neural networks (Preechakul et al., 2022; Wang et al., 2023), toward achieving unsupervised sequential disentanglement. Our architecture, shown in Fig. 2, addresses three limitations of current DiffAEs: 1) it processes sequential data, 2) it factorizes data into separate static and dynamic components, and 3) it is computationally efficient. From a high-level viewpoint, our network is similar to DiffAEs in that it is an autoencoder, consisting a (sequential) semantic encoder, a stochastic encoder, and a stochastic decoder.

**Encoders.** Inspired by works in sequential disentanglement (Yingzhen & Mandt, 2018; Bai et al., 2021; Naiman et al., 2023), we design a novel *sequential semantic encoder* to extract $\boldsymbol{s}_0$ and $\boldsymbol{d}_0^{1:V}$. Particularly, it consists of a U-Net (Ronneberger et al., 2015) and linear modules that operate on each sequence frame independently. Then, there is an LSTM module that summarizes the sequence into a latent representation $\boldsymbol{h}^{1:V}$. The last hidden, $\boldsymbol{h}^V$, is passed to a linear layer to produce $\boldsymbol{s}_0$, whereas $\boldsymbol{h}^{1:V}$ are processed with another LSTM and a linear layer to produce $\boldsymbol{d}_0^{1:V}$. Our stochastic encoder is EDM (Karras et al., 2022). This encoder adds noise to every frame $\mathbf{x}_0^{\tau}$ by sampling $\epsilon \sim \mathcal{N}(0, \sigma_t^2 I)$, yielding $\mathbf{x}_t^{\tau} = \mathbf{x}_0^{\tau} + \epsilon$. These encoders realize in practice the posterior in Eq. 5.

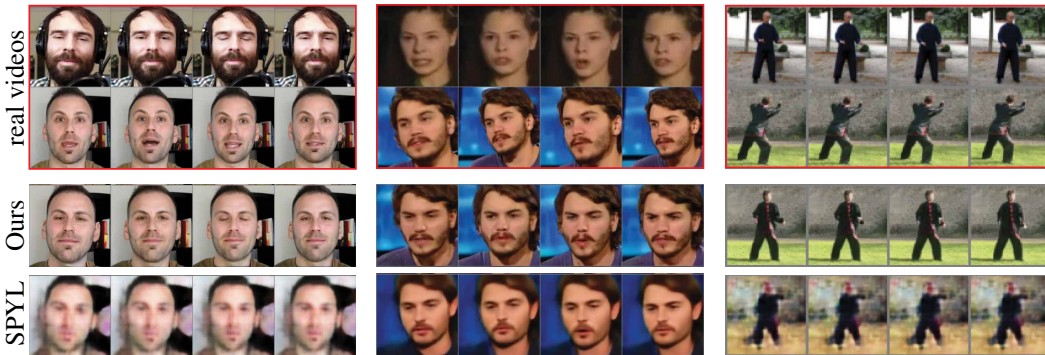

Figure 3: We present dynamic swap results of our approach (third row) and SPYL (fourth row) on CelebV-HQ (left), VoxCeleb (middle), and TaiChi-HD (right).

**Decoder.** The stochastic decoder in DiffAEs is DDIM (Song et al., 2020), a computationally-demanding sampler as it requires thousands of neural function evaluations (NFEs). To facilitate processing of real-world sequential information, we follow the decoding in EDM (Karras et al., 2022), featuring only 63 NFEs during inference. Our decoder $\boldsymbol{D}_\theta$ takes as inputs the noisy input $\mathbf{x}_t^\tau$ and disentangled factors $\mathbf{z}_0^\tau := (\boldsymbol{s}_0, \boldsymbol{d}_0^\tau)$, and it returns a denoised version of $\mathbf{x}_t^\tau$, denoted by $\tilde{\mathbf{x}}_0^\tau$. Given any $t \in [0, T]$ and $\tau \in \{1, \ldots, V\}$, the decoder is parameterized independently from other times $t', \tau'$ as follows

$$\tilde{\mathbf{x}}_0^\tau := \boldsymbol{D}_\theta(\mathbf{x}_t^\tau, t, \mathbf{z}_0^\tau) = c_t^{\text{skip}}\mathbf{x}_t^\tau + c_t^{\text{out}}\boldsymbol{F}_\theta\left(c_t^{\text{in}}\mathbf{x}_t^\tau, \mathbf{z}_0^\tau, c_t^{\text{noise}}\right) , \quad (7)$$

where $c_t^{\text{skip}}$ modulates the skip connection, $c_t^{\text{in}}$ and $c_t^{\text{out}}$ scale the input and output magnitudes, and $c_t^{\text{noise}}$ maps noise level at time $t$ into a conditioning input for the neural network $\boldsymbol{F}_\theta$, conditioned on $\mathbf{z}_0^\tau$ through AdaGN.

**Loss.** While prior sequential disentanglement works depend on intricate prior modeling, regularization terms, and mutual information losses, leading to many hyper-parameters and challenging training, we opt for a simpler objective containing a single loss term that is based on Eq. 6,

$$\mathbb{E}_{t,\mathbf{x}_t^\tau,\mathbf{z}_0^\tau,\mathbf{x}_0^\tau}\left[\lambda_t(c_t^{\text{out}})^2\|\boldsymbol{F}_\theta\left(c_t^{\text{in}}\mathbf{x}_t^\tau, \mathbf{z}_0^\tau, c_t^{\text{noise}}\right) - \frac{1}{c_t^{\text{out}}}(\mathbf{x}_0^\tau - c_t^{\text{skip}} \cdot \mathbf{x}_t^\tau)\|_2^2\right] . \quad (8)$$

While our loss in Eq. 8 does not include auxiliary terms, it promotes disentanglement due to two main reasons: i) the static factor $\boldsymbol{s}_0$ is shared across $\tau$, and thus it will not hold dynamic information, and ii) the dynamic factors $\boldsymbol{d}_0^\tau \in \mathbb{R}^k$ are low-dimensional (i.e., $k$ is small), making it difficult for $\boldsymbol{d}_0^\tau$ to store static features. Finally, we briefly mention that to support high-resolution sequences, we incorporate latent diffusion models (LDM) (Rombach et al., 2022), using a pre-trained VQ-VAE autoencoder to reduce the high-dimensionality of input frames. Instead of factorizing all the equations above with new symbols for the features VQ-VAE produces, we denote by $\mathrm{x}_0^{1:V}$ the input sequence, and we abuse the notation $\mathbf{x}_0^{1:V}$ to denote the latent features, i.e., $\mathbf{x}_0^{1:V} = \mathcal{E}(\mathrm{x}_0^{1:V})$ and $\mathrm{x}_0^{1:V} = \mathcal{D}(\mathbf{x}_0^{1:V})$, where $\mathcal{E}$ and $\mathcal{D}$ are the VQ-VAE encoder and decoder, respectively.

**Reconstruction and generation** require computing the reverse process, where we modified the reverse sampler (Karras et al., 2022) to depend on $\mathbf{x}_T^\tau$ and also on $\mathbf{z}_0^\tau$, see Alg. 1. For reconstruction, we extract $\mathbf{z}_0^{1:V}$ from the given $\mathbf{x}_0^{1:V}$ using the sequential semantic encoder described above, and we sample $\mathbf{x}_T^{1:V}$ either from $\mathbf{x}_T^{1:V} \sim p_T = \mathcal{N}(0, \sigma_T I)$ or by using Alg. 2 which is similar to $\mathbf{x}_T^{1:V} \sim p_{0T}(\mathbf{x}_T^{1:V} \mid \mathbf{x}_0^{1:V})$, to improve reconstruction quality. For generation, we extract $z_0^{1:V}$ by reversing a separate DDIM model (App. A.3) for $p_{T0}(\boldsymbol{s}_0, \boldsymbol{d}_0^{1:V} \mid \boldsymbol{s}_T, \boldsymbol{d}_T^{1:V})$ starting from $\mathbf{z}_T^{1:V} := (\boldsymbol{s}_T, \boldsymbol{d}_T^{1:V}) \sim \mathcal{N}(0, I)$, and we sample $\mathbf{x}_T^{1:V} \sim p_T = \mathcal{N}(0, \sigma_T I)$.

## 5 RESULTS

In this section, we empirically evaluate the modeling capabilities of DiffSDA in comparison to recent state-of-the-art methods, SPYL (Naiman et al., 2023) and DBSE (Berman et al., 2024). We consider

Table 1: Preservation of objects (AED) and motions (AKD) is estimated across several datasets and methods. The labels 'static frozen' and 'dynamics frozen' correspond to samples $\mathbf{z}^s$ and $\mathbf{z}^d$.

| | AED↓ (static frozen) | | | AKD↓ (dynamics frozen) | | |
|---|---|---|---|---|---|---|
| | SPYL | DBSE | Ours | SPYL | DBSE | Ours |
| MUG ($64 \times 64$) | 0.766 | 0.773 | **0.751** | 1.132 | 1.118 | **0.802** |
| VoxCeleb ($256 \times 256$) | 1.058 | 1.026 | **0.846** | 4.705 | 10.96 | **2.793** |
| CelebV-HQ ($256 \times 256$) | 0.631 | 0.751 | **0.540** | 39.16 | 28.69 | **6.932** |
| TaiChi-HD ($64 \times 64$) | 0.443 | **0.325** | 0.326 | 7.681 | 6.312 | **2.143** |

both quantitative and qualitative experiments conducted on three high-resolution, real-world visual datasets that have not been previously used for sequential disentanglement representation learning: VoxCeleb (Nagrani et al., 2017), CelebV-HQ (Zhu et al., 2022), and TaiChi-HD (Siarohin et al., 2019), along with the MUG dataset (Aifanti et al., 2010). Detailed descriptions of the datasets and their pre-processing can be found in App. C. For brevity, we omit below the subscript indicating the diffusion step for clean samples (corresponding to time step 0).

## 5.1 CONDITIONAL SWAPPING

We begin our evaluation with the classic conditional swapping task (Yingzhen & Mandt, 2018; Bai et al., 2021). Given two sample videos $\mathbf{x}$, $\hat{\mathbf{x}} \sim p_0$, the goal in this experiment is to create a new sample $\bar{\mathbf{x}}$, conditioned on the static factor of $\mathbf{x}$ and dynamic features of $\hat{\mathbf{x}}$. This is done by extracting the latent factors $\mathbf{z} = (\boldsymbol{s}, \boldsymbol{d}^{1:V})$ and $\hat{\mathbf{z}} = (\hat{\boldsymbol{s}}, \hat{\boldsymbol{d}}^{1:V})$ for $\mathbf{x}$ and $\hat{\mathbf{x}}$, respectively. The new sample $\bar{\mathbf{x}}$ is defined to be the reconstruction of $\bar{\mathbf{z}} = (\boldsymbol{s}, \hat{\boldsymbol{d}}^{1:V})$ through Alg. 1, see Sec. 4. In an ideal swap, $\bar{\mathbf{x}}$ preserves the static characteristics of $\mathbf{x}$ while presenting the dynamics of $\hat{\mathbf{x}}$, thus demonstrating strong disentanglement capabilities of the swapping method. We show in Fig. 1 (left) a swap example of DiffSDA, where the top two rows are real videos, and the third row shows the new sample obtained by preserving the static features of the first row and using the dynamics of the second row. Remarkably, while the people in these sequences are very different, many fine details are transferred, including head angle and orientation, as well as mouth and eyes orientation and openness. In Fig. 3, we present additional swap results on CelebV-HQ (left), VoxCeleb (middle), and TaiChi-HD (right), comparing DiffSDA (third row) to SPYL (fourth row). Notably, our approach produces high-quality samples, while swapping the dynamics of the second row into the first row, whereas SPYL struggles both with the reconstruction and swap.

In addition to the above qualitative evaluation, we also want to quantitatively assess the effectiveness of DiffSDA. We report in App. E results from the traditional quantitative benchmark, where a pre-trained judge (classifier) is used to determine if swapped content is correct (Bai et al., 2021). However, there are two main issues with the benchmark: i) it depends on labeled data, making it relevant to only a small number of datasets; and ii) results are sensitive to the expressivity and generalizability of the judge. For instance, swapping a smiling expression from person A to person B, may result in person B having a smile, different from the one in the data. In these cases, the judge may wrongly classify a different expression to the smiling person B, see App. E.

Towards addressing these issues, we propose new *unsupervised* swapping metrics to quantitatively measure the model's disentanglement abilities. We adopt estimators commonly used in animation for assessing whether objects and motions are preserved (Siarohin et al., 2019). Specifically, we utilize the *average Euclidean distance* (AED) that is based on the distances between the latent representations of images. Further, we also employ the *average keypoint distance* (AKD) which

Table 2: Reconstruction errors are measured in terms of AED, AKD, and MSE across several datasets and models. We find DiffSDA to be orders-of-magnitude better than other methods.

| | AED↓ | | | AKD↓ | | | MSE↓ | | |
|---|---|---|---|---|---|---|---|---|---|
| | SPYL | DBSE | Ours | SPYL | DBSE | Ours | SPYL | DBSE | Ours |
| MUG ($64 \times 64$) | 0.491 | 0.486 | **0.113** | 0.465 | 0.479 | **0.062** | 0.001 | 0.001 | **3.0e−7** |
| VoxCeleb ($256 \times 256$) | 0.987 | 1.027 | **0.374** | 2.267 | 2.428 | **1.092** | 0.005 | 0.003 | **4.6e−4** |
| CelebV-HQ ($256 \times 256$) | 0.701 | 0.777 | **0.292** | 15.00 | 13.78 | **1.256** | 0.012 | 0.006 | **5.9e−4** |
| TaiChi-HD ($64 \times 64$) | 0.319 | 0.294 | **0.001** | 4.311 | 3.833 | **0.099** | 0.018 | 0.007 | **2.0e−7** |

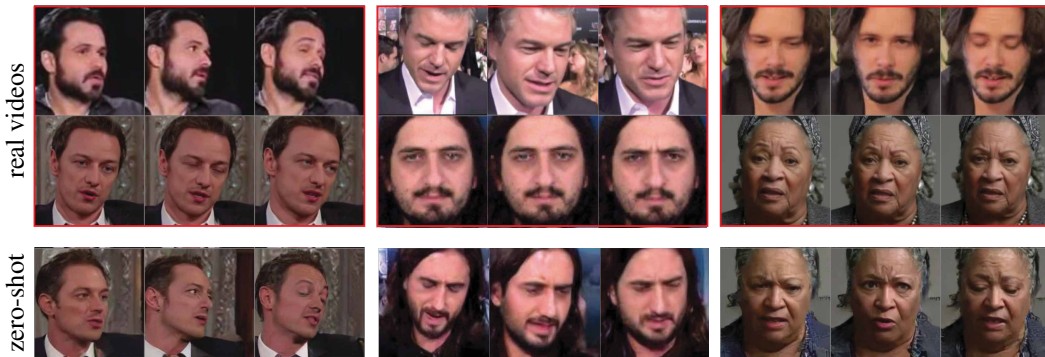

Figure 4: Zero-shot swap results, training on VoxCeleb and tested on CelebV-HQ or MUG.

computes the distances between selected keypoints in images. Intuitively, AED and AKD have been designed to identify the preservation of objects and motions in images, respectively. See App. D for comprehensive definitions. Equipped with these new metrics, we perform conditional swapping over a pre-defined random list of sample pairs, $\mathbf{x}, \hat{\mathbf{x}}$. Particularly, we reconstruct new samples of the form $\mathbf{z}^s := (\boldsymbol{s}, \hat{\boldsymbol{d}}^{1:V})$ and $\mathbf{z}^d := (\hat{\boldsymbol{s}}, \boldsymbol{d}^{1:V})$, encoding dynamic and static swaps, respectively. We compute the AED of $\mathbf{z}^s$ with respect to $\mathbf{z}$ (arising from $\mathbf{x}$), expecting their static features to be similar. Following the same logic, we compute the AKD of $\mathbf{x}^d$ (reconstructed from $\mathbf{z}^d$) and $\mathbf{x}$, as they share the dynamic factors. Our findings are presented in Tab. 1, where DiffSDA outperforms SOTA previous (SPYL, DBSE) approaches across all datasets, except for AED on TaiChi-HD, where we attain the second best error. Notably, our AKD errors are significantly lower than SPYL and DBSE. Further, we apply these metrics to assess reconstruction performance, as well as the mean squared error (MSE), with the results shown in Tab. 2. Again, DiffSDA is superior to current SOTA methods.

## 5.2 ZERO-SHOT DISENTANGLEMENT

In the previous sub-section, the conditional swap was performed on the held-out test set of each dataset on which we trained on. In contrast to previous work, for the first time, we perform the same task on a dataset unseen during training. We show an example in Fig. 1 (middle) of zero-shot swap, where our model was trained on the VoxCeleb dataset (1st row) and the inferred sequence was taken from MUG (2nd row). Particularly, we froze the static features of the MUG sample and swapped the dynamic factors with those of VoxCeleb (3rd row). Remarkably, in addition to changing the facial expression of the person, DiffSDA also adds the necessary details to mimic the body pose. We emphasize that the MUG dataset does not include sequences similar to the third row in Fig. 1, but rather zoomed-in facial videos as shown in the second row, thus, our zero-shot results present a significant adaptation to the new data. Additionally, we include in Fig. 4 zero-shot examples where DiffSDA is trained on VoxCeleb and evaluated on CelebV-HQ or MUG. These results further highlight the effectivity of our approach in transferring dynamic features across different datasets. Finally, we provide more zero-shot examples in App. F.4.

## 5.3 TOWARD MULTIFACTOR DISENTANGLEMENT

Multifactor sequential disentanglement is a challenging problem, where the objective is to produce several static factors and several dynamic factors per frame (Berman et al., 2023). Here, we show that our model has the potential to further disentangle the static and dynamic features into additional factors of variation. Inspired by DiffAE (Preechakul et al., 2022), we propose to explore the learned latent space in an unsupervised linear fashion, particularly, using principal component analysis (PCA). For instance, to obtain fine-grained semantic static factors of variation, we sample a large batch of static vectors $\hat{\boldsymbol{s}}_j \in \mathbb{R}^h$, with $h$ the static latent size, $j = 1, \ldots, b = 2^{15}$. Then, we compute PCA on the matrix formed by arranging $\{\hat{\boldsymbol{s}}_j\}$ in its columns, yielding the principal components $\{v_i\}_{i=1}^h$, given that $b \geq h$. We can utilize the latter pool of static variability by exploring

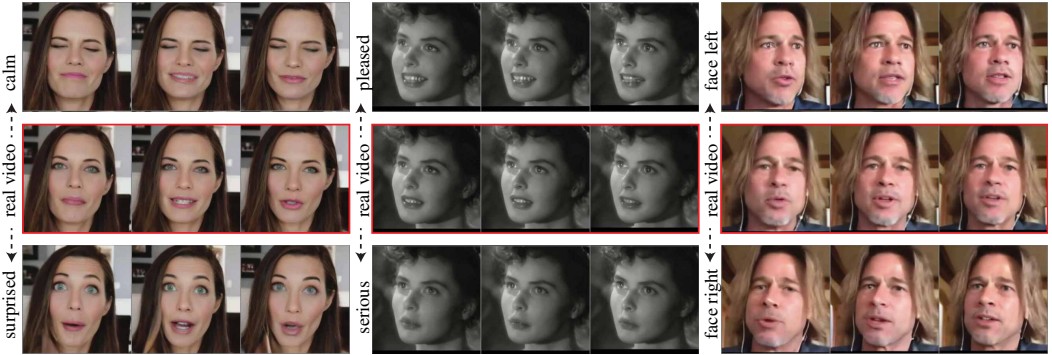

Figure 5: Traversing the latent space of DiffSDA via PCA reveals multiple dynamic variations on CelebV-HQ, including surprised and serious expressions, and different head orientations.

the latent space from a static code $s$ of a real example $\mathbf{x}$ in the test set, i.e.,

$$\bar{s} = \left( \frac{s - \mu_{\hat{s}}}{\sigma_{\hat{s}}} + \alpha v_i \cdot \sqrt{h} \right) \cdot \sigma_{\hat{s}} + \mu_{\hat{s}} , \tag{9}$$

where $\mu_{\hat{s}}$ and $\sigma_{\hat{s}}^2$ are the mean and variance of the sampled static features, $\{\hat{s}_j\}_{j=1}^b$, and $\alpha \in [-\kappa, \kappa]$, notice that $\alpha = 0$ recovers the original sequence. The new sample $\bar{\mathbf{x}}$ is obtained by reconstructing the new static features $\bar{s}$ with the original dynamic factors $d^{1:V}$ of $\mathbf{x}$.

We demonstrate a static PCA exploration in Fig. 1 (right) on VoxCeleb. The middle row is the real video, whereas the top and bottom rows use positive and negative $\alpha$ values, respectively. Our results show that traversing in the positive direction yields more masculine appearances, and in contrast, going in the negative direction produces more feminine characters. Importantly, we highlight that other static features and the dynamics are fully preserved across the sequence. We plot additional samples in Fig. 5 on CelebV-HQ, where the PCA directions alter facial expressions and head orientation. In App. F.5, we present further results on full sequences using multiple $\alpha$ values to demonstrate the gradual transition in the latent space. Notably, we find in our exploration principal components that control other features such as skin tone, image blurriness, and more.

### 5.4 UNCONDITIONAL GENERATION AND SWAPPING

In addition to the conditional and zero-shot shot tasks considered above, we can also perform such tasks in an unconditional manner. Specifically, given a real sequence $\mathbf{x}^{1:V}$ with its factors $(s, d^{1:V})$, we can unconditionally sample new $(\hat{s}, \hat{d}^{1:V})$ using our separate DDIM model (see Sec. 4). We then reconstruct the static swap $(\hat{s}, d^{1:V})$ and the dynamic swap $(s, \hat{d}^{1:V})$ similarly as described above. In Fig. 6, we present unconditional swap results on CelebV-HQ (left), VoxCeleb (middle), and TaiChi-HD (right). The middle rows represent the original sequences, whereas the top and bottom rows demonstrate dynamic and static swaps, respectively. Across all datasets and swap settings, our approach succeeds in modifying the swapped features while preserving the frozen factors, either in the static or in the dynamic examples. We show additional examples in App. F.3.

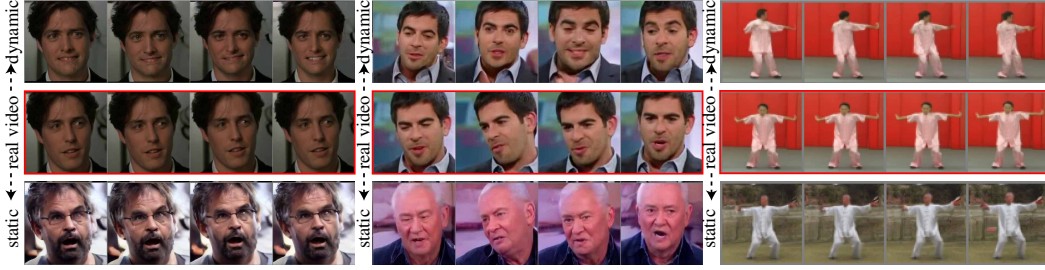

Figure 6: Unconditional dynamic (top) and static (bottom) swap results on CelebV-HQ (left), Vox-Celeb (middle), and TaiChi-HD (right).

Table 3: Disentanglement metrics on TIMIT.

| Method | static EER↓ | dynamic EER ↑ | Disentanglement Gap ↑ |
|--------|-------------|---------------|------------------------|
| FHVAE | 5.06% | 22.77% | 17.71% |
| DSVAE | 5.64% | 19.20% | 13.56% |
| R-WAE | 4.73% | 23.41% | 18.68% |
| S3VAE | 5.02% | 25.51% | 20.49% |
| SKD | 4.46% | 26.78% | 22.32% |
| C-DSVAE | 4.03% | 31.81% | 27.78% |
| SPYL | **3.41**% | 33.22% | 29.81% |
| DBSE | 3.50% | 34.62% | 31.11% |
| Ours | 4.43% | **46.72**% | **42.29**% |

## 5.5 AUDIO MODALITY

Our method is inherently modality-agnostic and not constrained to the video domain. Unlike video-focused methods, which often require substantial adjustments when applied to new modalities, our approach can adapt to various modalities with only minor modifications to the backbone architecture. For instance, to support audio, we replace the U-Net with a simple MLP. This flexibility aligns with prior sequential disentanglement methods such as SPYL (Naiman et al., 2023) and DBSE (Berman et al., 2024). In Tab. 3, we demonstrate the adaptability of our model by successfully disentangling audio data on the TIMIT dataset, a widely used benchmark in speech-related tasks. Following the speaker identification benchmarks established by Yingzhen & Mandt (2018) and Zhu et al. (2020), we evaluate disentanglement quality using the Equal Error Rate (EER), a standard metric in speech tasks. Notably, our model improves the disentanglement gap by over 11%, achieving 42.29% dynamic EER compared to 31.11% by DBSE, thereby outperforming current state-of-the-art methods. This result underscores the effectiveness of our approach in handling the audio modality. Additional details about the dataset, metrics, and implementation are provided in the appendix.

## 6 CONCLUSIONS

The analysis and results of this study underscore the potential of the proposed DiffSDA model to address key limitations in sequential disentanglement, specifically in the context of complex real-world visual data and speech audio. By leveraging a novel probabilistic framework, diffusion autoencoders, efficient samplers, and latent diffusion models, DiffSDA provides a robust solution for disentangling both static and dynamic factors in sequences, outperforming existing state-of-the-art methods. Moreover, the introduction of a new real-world visual evaluation protocol marks a significant step towards standardizing the assessment of sequential disentanglement models. Nevertheless, while DiffSDA shows promise in handling high-resolution videos and varied datasets, future research should focus on optimizing its computational efficiency and extending its applicability to more diverse sequence modalities, such as sensor data, and general time series. Each of these modalities presents unique challenges, such as varying temporal characteristics and distinct data patterns, which may require adapting the model architecture and training strategies. Additionally, our suggested benchmark is designed for visual or speech information. Consequently, extending our work to additional modalities also requires designing new benchmarks to account for, e.g., trend prediction or temporal coherence. Finally, while the benchmark presents significant progress in handling real-world data in a sequential disentanglement context, it primarily consists of structured data, such as single objects or single views. Modeling and evaluating unstructured data, such as multiple objects within a video or multiple viewpoints (Jabri et al., 2024; Wu et al., 2024), remains an open challenge in the context of sequential disentanglement. This raises an important and compelling direction for future research.

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

## A DIFFSDA MODELING

### A.1 UNSUPERVISED SEQUENTIAL DISENTANGLEMENT

Unsupervised sequential disentanglement is a challenging problem in representation learning, aiming to decompose a given dataset to its static (time-independent) and dynamic (time-dependent) factors of variation. Let $\mathcal{D} = \{\mathbf{x}_j^{1:V}\}_{j=1}^N$ be a dataset with $N$ sequences $\mathbf{x}_j^{1:V} := \{\mathbf{x}_j^1, \ldots, \mathbf{x}_j^V\}$, where $\mathbf{x}_j^\tau \in \mathbb{R}^d$. We omit the subscript $j$ for brevity, unless noted otherwise. The goal of sequential disentanglement is to extract an alternative representation of $\mathbf{x}^{1:V}$ via a single static factor $\mathbf{s}$ and multiple dynamic factors $\mathbf{d}^{1:V}$. Note that $\mathbf{s}$ is shared across the sequence.

We can formalize the sequential disentanglement problem as a *generative task*, where every sequence $\mathbf{x}^{1:V}$ from the data space $\mathcal{X}$ is conditioned on some $\mathbf{z}^{1:V}$ from a latent space $\mathcal{Z}$. We aim to maximize the probability of each sequence under the entire generative process

$$p(\mathbf{x}^{1:V}) = \int_{\mathcal{Z}} p(\mathbf{x}^{1:V} \mid \mathbf{z}^{1:V}) p(\mathbf{z}^{1:V}) \, \mathrm{d}\, \mathbf{z}^{1:V} \;, \tag{10}$$

where $\mathbf{z}^{1:V} := (\mathbf{s}, \mathbf{d}^{1:V})$. One of the main challenges with directly maximizing Eq. (10) is that the latent space $\mathcal{Z}$ is too large to practically integrate over. Instead, a separate distribution, denoted here as $q(\mathbf{z}^{1:V} \mid \mathbf{x}^{1:V})$, is used to narrow search to be only over $\mathbf{z}^{1:V}$ associated with sequences from the dataset $\mathcal{D}$. Importantly, the distributions $p(\mathbf{x}^{1:V} \mid \mathbf{z}^{1:V})$ and $q(\mathbf{z}^{1:V} \mid \mathbf{x}^{1:V})$ take the form of a decoder and an encoder in practice, suggesting the development of *autoencoder* sequential disentanglement models (Yingzhen & Mandt, 2018). The above $p(\mathbf{x}^{1:V} \mid \mathbf{z}^{1:V})$ and $q(\mathbf{z}^{1:V} \mid \mathbf{x}^{1:V})$ are denoted by $p_{T0}(\mathbf{x}_0^\tau \mid \mathbf{x}_T^\tau, \mathbf{s}_0, \mathbf{d}_0^\tau)$ and $p(\mathbf{x}_t^{1:V}, \mathbf{s}_0, \mathbf{d}_0^{1:V} \mid \mathbf{x}_0^{1:V})$, respectively, in Eq. 4 and Eq. 5.

### A.2 HIGH-RESOLUTION DISENTANGLED SEQUENTIAL DIFFUSION AUTOENCODER

In addition to transitioning to real-world data, our goal is to manage high-resolution data for unsupervised sequential disentanglement, for the first time. Drawing inspiration from Rombach et al. (2022), we incorporate perceptual image compression, which combines an autoencoder with a perceptual loss (Zhang et al., 2018) and a patch-based adversarial objective (Dosovitskiy & Brox, 2016; Esser et al., 2021; Isola et al., 2017). Specifically, we explore two main variants of the autoencoder. The first variant applies a small Kullback–Leibler penalty to encourage the learned latent space to approximate a standard normal distribution, similar to a VAE (Kingma, 2013; Rezende et al., 2014). The second variant integrates a vector quantization layer (Van Den Oord et al., 2017; Razavi et al., 2019) within the decoder. Empirically, we find that the VQ-VAE-based model performs better when combined with our method. Given a pre-trained encoder $\mathcal{E}$ and decoder $\mathcal{D}$, we can extract $\mathbf{x}_0^\tau = \mathcal{E}(\mathrm{x}_0^\tau)$, which represents a low-dimensional latent space where high-frequency, imperceptible details are abstracted away. Finally, $\mathrm{x}_0^\tau$ can be reconstructed from the latent $\mathbf{x}_0^\tau$ by applying the decoder $\mathrm{x}_0^\tau = \mathcal{D}(\mathbf{x}_0^\tau)$. The EDM formulation in Eq. 7 makes relatively strong assumptions about the mean and standard deviation of the training data. To meet these assumptions, we opt to normalize the training data globally rather than adjusting the value of $\sigma_{\text{data}}$, which could significantly affect other hyperparameters (Karras et al., 2024). Therefore, we keep $\sigma_{\text{data}}$ at its default value of 0.5 and ensure that the latents have a zero mean during dataset preprocessing. When generating sequence elements, we reverse this normalization before applying $\mathcal{D}$.

### A.3 PRIOR MODELING

We model the prior static and dynamic distribution with $p_{T0}(\boldsymbol{s}_0, \boldsymbol{d}_0^{1:V} \mid \boldsymbol{s}_T, \boldsymbol{d}_T^{1:V})$. To sample static and dynamic factors, we train a separate latent DDIM model (Song et al., 2020). Then, we can extract the factors by sampling noise, and reversing the trained model. Specifically, we learn $p_{\Delta t}(\mathbf{z}_{t-1}^{1:V} \mid \mathbf{z}_t^{1:V})$ where $\mathbf{z}_0 = (\boldsymbol{s}_0, \boldsymbol{d}_0^{1:V})$ are the outputs of our sequential semantic encoder. The training is done by simply optimizing the $\mathcal{L}_{\text{latent}}$ with respect to DDIM's output $\varepsilon_\phi(\cdot)$:

$$\mathcal{L}_{\text{latent}} = \sum_{t=1}^T \mathbb{E}_{\mathbf{z}^{1:V}, \varepsilon_t} \left[ \|\varepsilon_\phi(\mathbf{z}_t^{1:V}, t) - \varepsilon_t\| \right] \tag{11}$$

where $\varepsilon_t \in \mathbb{R}^{dV+s} \sim \mathcal{N}(\mathbf{0}, \mathbf{I})$, $V$ is the sequence length, $s, d$ are the static and dynamic factors dimensions respectively. Additionally, $\mathbf{z}_t^{1:V}$ is the noise version of $\mathbf{z}_t$ as described in Song et al.

---

**Algorithm 1** Conditioned Stochastic Sampler with $\sigma(t) = t$ and $s(t) = 1$.

---

1: **procedure** CONDITIONEDSTOCHASTICSAMPLER($D_\theta$, $t_{i\in\{0,\dots,N\}}$, $\gamma_{i\in\{0,\dots,N-1\}}$, $\mathbf{z}_0^{1:V}$, $\mathbf{x}_0^{1:V}$, $S_{\text{noise}}^2$)
2:     **if** $\mathbf{x}_0^{1:V} \neq$ None **then**
3:         $\mathbf{x}_N^{1:V} \leftarrow$ **Algorithm 2 output**
4:     **else**
5:         **sample** $\mathbf{x}_N^{1:V} \sim \mathcal{N}(\mathbf{0},\, t_N^2\,\mathbf{I})$
6:     **for** $i \in \{N, \dots, 1\}$ **do**          $\triangleright\ \gamma_i = \begin{cases} \min\left(\frac{S_{\text{churn}}}{N}, \sqrt{2}-1\right) & \text{if } t_i \in [S_{\text{tmin}}, S_{\text{tmax}}] \\ 0 & \text{otherwise} \end{cases}$
7:         **sample** $\boldsymbol{\epsilon}_i \sim \mathcal{N}(\mathbf{0},\, S_{\text{noise}}^2\,\mathbf{I})$
8:         $\hat{t}_i \leftarrow t_i + \gamma_i t_i$         $\triangleright$ Select temporarily increased noise level $\hat{t}_i$
9:         $\hat{\mathbf{x}}_i^\tau \leftarrow \mathbf{x}_i^\tau + \sqrt{\hat{t}_i^2 - t_i^2}\,\boldsymbol{\epsilon}_i$         $\triangleright$ Add new noise to move from $t_i$ to $t_i$
10:         $\boldsymbol{d}_i \leftarrow \big(\mathbf{x}_i^\tau - D_\theta(\mathbf{x}_i^\tau, \mathbf{z}_0^\tau; \hat{t}_i)\big)/\hat{t}_i$         $\triangleright$ Evaluate $\mathrm{d}\mathbf{x}/\mathrm{d}t$ at $t_i$
11:         $\mathbf{x}_{i-1}^\tau \leftarrow \mathbf{x}_i^\tau + (t_{i-1} - \hat{t}_i)\boldsymbol{d}_i$         $\triangleright$ Take Euler step from $t_i$ to $t_{i-1}$
12:         **if** $t_{i-1} \neq 0$ **then**
13:             $\boldsymbol{d}_i' \leftarrow \big(\mathbf{x}_{i-1} - D_\theta(\mathbf{x}_{i-1}, \mathbf{z}_0^\tau; t_{i-1})\big)/t_{i-1}$         $\triangleright$ Apply $2^{\text{nd}}$ order correction
14:             $\mathbf{x}_{i-1}^\tau \leftarrow \hat{\mathbf{x}}_i^\tau + (t_{i-1} - \hat{t}_i)\big(\frac{1}{2}\boldsymbol{d}_i + \frac{1}{2}\boldsymbol{d}_i'\big)$
15:     **return** $\mathbf{x}_0$

---

**Algorithm 2** Stochastic Encoding with $\sigma(t) = t$ and $s(t) = 1$.

---

1: **procedure** STOCHASTICENCODER($D_\theta$, $t_{i\in\{0,\dots,N\}}$, $\gamma_{i\in\{0,\dots,N-1\}}$, $\mathbf{x}_0^{1:V}$, $\mathbf{z}_0^{1:V}$)
2:     **for** $i \in \{0, \dots, N-1\}$ **do**
3:         $\boldsymbol{d}_i \leftarrow \big(\mathbf{x}_i^\tau - D_\theta(\mathbf{x}_i^\tau, \mathbf{z}_0^\tau; t_i)\big)/t_i$         $\triangleright$ Evaluate $\mathrm{d}\mathbf{x}^\tau/\mathrm{d}t$ at $t_i$
4:         $\mathbf{x}_{i+1}^\tau \leftarrow \mathbf{x}_i^\tau + (t_{i+1} - t_i)\boldsymbol{d}_i$         $\triangleright$ Take Euler step from $t_i$ to $t_{i+1}$
5:         **if** $t_{i+1} \neq \sigma_{\max}$ **then**
6:             $\boldsymbol{d}_i' \leftarrow \big(\mathbf{x}_{i+1}^\tau - D_\theta(\mathbf{x}_{i+1}^\tau, \mathbf{z}_0^\tau; t_{i+1})\big)/t_{i+1}$         $\triangleright$ Apply $2^{\text{nd}}$ order correction
7:             $\mathbf{x}_{i+1}^\tau \leftarrow \mathbf{x}_i^\tau + (t_{i+1} - t_i)\big(\frac{1}{2}\boldsymbol{d}_i + \frac{1}{2}\boldsymbol{d}_i'\big)$
8:     **return** $\mathbf{x}_N^{1:V}$

---

(2020). For designing the architecture of our latent model, we follow Preechakul et al. (2022) and it is based on 10 MLP layers. Our network architecture and hyperparamters are provided in Tab. 5.

### A.4 REVERSE PROCESSES

The detailed reverse sampling algorithm is provided in Alg. 1. We follow Karras et al. (2022) sampling techniques, however, each step in our reverse process is conditioned on the latent static and dynamic factors extracted by our sequential semantic encoder. As in Preechakul et al. (2022), we observe that auto-encoding is improved significantly when using the stochastic encoding technique. Since we have a different reverse process, we provide the algorithm for stochastic encoding for our modeling in Alg. 2. Finally, when performing conditional swapping, we observe that performing stochastic encoding on the sample from which we borrow the dynamics and using it as an input to Alg. 1, improves the results empirically. That is, given two sample videos $\mathbf{x}, \hat{\mathbf{x}} \sim p_0$, to create a new sample $\bar{\mathbf{x}}$, conditioned on the static factor of $\mathbf{x}$ and dynamic features of $\hat{\mathbf{x}}$, we use the stochastic encoding of $\hat{\mathbf{x}}$ in Alg. 1.

## B HYPER-PARAMETERS

The hyperparameters used in our autoencoder are listed in Tab. 4, detailing the configurations for each dataset: MUG, TaiChi-HD, VoxCeleb, CelebV-HQ and TIMIT. We provide the values of essential parameters such as sequence lengths, batch sizes, learning rates, and the use of $P_{\text{mean}}$ and $P_{\text{std}}$ to manage noise disturbance during training. In addition, the table specifies whether VQ-VAE was employed. Tab. 5 outlines the architecture of our latent DDIM model, including batch size, number of epochs, MLP layers, hidden sizes, and the $\beta$ scheduler. These details are essential for understanding the model's structure and its training process. For the VQ-VAE model, we utilized the pre-trained model from (Rombach et al., 2022) with hyperparameters $f = 8$, $Z = 256$, and $d = 4$, which encodes a frame of size $3 \times 256 \times 256$ into a latent representation of size $4 \times 32 \times 32$.

Table 4: Hyperparameters for all datasets.

| Dataset | MUG | TaiChi-HD | VoxCeleb | CelebV-HQ | TIMIT |
|---|---|---|---|---|---|
| $P_{\text{maen}}$ | $-1.2$ | $-1.2$ | $-0.4$ | $-0.4$ | $-0.4$ |
| $P_{\text{std}}$ | $1.2$ | $1.2$ | $1.0$ | $1.0$ | $1.0$ |
| NFE | 71 | 63 | 63 | 63 | 63 |
| VQ-VAE | ✗ | ✗ | ✓ | ✓ | ✗ |
| lr | 1e$-4$ | 1e$-4$ | 1e$-4$ | 1e$-4$ | 1e$-4$ |
| bsz | 8 | 16 | 16 | 16 | 128 |
| #Epoch | 1600 | 40 | 100 | 450 | 750 |
| Dataset repeats | 1 | 150 | 1 | 1 | 1 |
| $s$ dim | 256 | 512 | 512 | 1024 | 32 |
| $d$ dim | 64 | 64 | 12 | 16 | 4 |
| hidden dim | 128 | 1024 | 1024 | 1024 | 128 |
| Base channels | 64 | 64 | 192 | 192 | 256 |
| Channel multipliers | | | $[1,2,2,2]$ | | $[4,4,4,4]$ |
| Attention placement | | | $[2]$ | | None |
| Encoder base ch | 64 | 64 | 192 | 192 | 128 |
| Encoder ch. mult. | | | $[1,2,2,2]$ | | $[4,4,4,4]$ |
| Enc. attn. placement | | | $[2]$ | | None |
| Input size | $3 \times 64 \times 64$ | $3 \times 64 \times 64$ | $3 \times 256 \times 256$ | $3 \times 256 \times 256$ | 80 |
| Seq len | 15 | 10 | 10 | 10 | 68 |
| Optimizer | | | AdamW (weight decay$= 1$e$-5$) | | |
| Backbone | | | Unet | | MLP |

Table 5: Network architecture of our latent DDIM.

| Parameter | MUG | TaiChi-HD | VoxCeleb | Celebv-HQ |
|---|---|---|---|---|
| Batch size | 128 | 128 | 128 | 128 |
| #Epoch | 500 | 500 | 200 | 1000 |
| MLP layers ($N$) | | | 10 | |
| MLP hidden size | 1216 | 5008 | 2528 | 4736 |
| $\beta$ scheduler | | | Linear | |
| Learning rate | | | 1e$-4$ | |
| Optimizer | | AdamW (weight decay$= 1$e$-5$) | | |
| Train Diff T | | | 1000 | |
| Diffusion loss | | L2 loss with noise prediction $\epsilon$ | | |

## C  DATASETS

**MUG.** The MUG facial expression dataset, introduced by Aifanti et al. (2010), contains image sequences from 52 subjects, each displaying six distinct facial expressions: anger, fear, disgust, happiness, sadness, and surprise. Each video sequence in the dataset ranges from 50 to 160 frames. To create sequences of length 15, as done in prior work (Bai et al., 2021), we randomly select 15 frames from the original sequences. We then apply Haar Cascade face detection to crop the faces and resize them to $64 \times 64$ pixels, resulting in sequences of $x \in \mathbb{R}^{15 \times 3 \times 64 \times 64}$. The final dataset comprises 3,429 samples. In the case of of the zero shot experiments we resize the images to $256 \times 256$ pixels.

**TaiChi-HD.** The TaiChi-HD dataset, introduced by Siarohin et al. (2019), contains videos of full human bodies performing Tai Chi actions. We follow the original preprocessing steps from FOMM (Siarohin et al., 2019) and use a $64 \times 64$ version of the dataset. The dataset comprises 3,081 video chunks with varying lengths, ranging from 128 to 1,024 frames. We split the data into 90% for training and 10% for testing. To create sequences of length 10, similar to the approach used for the MUG dataset, we randomly select 10 frames from the original sequences. The resulting sequences are resized to $64 \times 64$ pixels, forming $x \in \mathbb{R}^{10 \times 3 \times 64 \times 64}$.

**VoxCeleb.** The VoxCeleb dataset (Nagrani et al., 2017) is a collection of face videos extracted from YouTube. We used the preprocessing steps from Albanie et al. (2018), where faces are extracted, and the videos are processed at 25/6 fps. The dataset comprises 22,496 videos and 153,516 video chunks. We used the verification split, which includes 1,211 speakers in the training set and 40 different speakers in the test set, resulting in 148,642 video chunks for training and 4,874 for testing. To create sequences of length 10, we randomly select 10 frames from the original sequences. The videos are processed at a resolution of $256 \times 256$ resulting in sequences represented as $x \in \mathbb{R}^{10 \times 3 \times 256 \times 256}$.

**CelebV-HQ.** The CelebV-HQ dataset (Zhu et al., 2022) is a large-scale collection of high-quality video clips featuring faces, extracted from various online sources. The dataset consists of 35,666 video clips involving 15,653 identities, with each clip manually labeled with 83 facial attributes, including 40 appearance attributes, 35 action attributes, and 8 emotion attributes. The videos were initially processed at a resolution of $512 \times 512$. We then used Wang et al. (2021) to crop the facial regions, resulting in videos at a $256 \times 256$ resolution. To create sequences of length 10, we randomly selected 10 frames from the original sequences, producing sequences represented as $x \in \mathbb{R}^{10 \times 3 \times 256 \times 256}$.

**TIMIT.** The TIMIT dataset, introduced by Garofolo (1993), is a collection of read speech designed for acoustic-phonetic research and other speech-related tasks. It contains 6300 utterances, totaling approximately 5.4 hours of audio recordings, from 630 speakers (both men and women). Each speaker contributes 10 sentences, providing a diverse and comprehensive pool of speech data. To pre-process the data we use mel-spectogram feature extraction with 8.5ms frame shift applied to the audio. Subsequently, segments of 580ms duration, equivalent to 68 frames, are sampled from the audio and treated as independent samples.

## D METRICS

**Average Keypoint Distance (AKD).** To evaluate whether the motion in the reconstructed video is preserved, we utilize pre-trained third-party keypoint detectors on the TaiChi-HD, VoxCeleb, CelebV-HQ, and MUG datasets. For the VoxCeleb, CelebV-HQ and MUG datasets, we employ the facial landmark detector from Bulat & Tzimiropoulos (2017), whereas for the TaiChi-HD dataset, we use the human-pose estimator from Cao et al. (2017). Keypoints are computed independently for each frame. AKD is calculated by averaging the $L_1$ distance between the detected keypoints in the ground truth and the generated video. The TaiChi-HD and MUG datasets are evaluated at a resolution of $64 \times 64$ pixels, and the VoxCeleb and CelebV-HQ datasets at $256 \times 256$ pixels. If the model output is at a lower resolution, it is interpolated to $256 \times 256$ pixels for evaluation.

**Average Euclidean Distance (AED).** To assess whether the identity in the reconstructed video is preserved, we use the Average Euclidean Distance (AED) metric. AED is calculated by measuring the Euclidean distance between the feature representations of the ground truth and the generated video frames. We selected the feature embedding following the example set in Siarohin et al. (2019). For the VoxCeleb, CelebV-HQ, and MUG datasets, we use a VGG-FACE for facial identification using the framework of Serengil & Ozpinar (2020), whereas for TaiChi-HD, we use a network trained for person re-identification (Hermans et al., 2017). TaiChi-HD and MUG are evaluated at a resolution of $64 \times 64$ pixels, and VoxCeleb and CelebV-HQ at $256 \times 256$ pixels.

To ensure fairness when measuring AED and AKD, we created a predefined dataset of example pairs, ensuring that all models are evaluated on the exact same set of pairs. This is important because when measuring quantitative metrics, the results may vary depending on the dynamics swapped between two subjects, as e.g., the key points in AKD in the original video are influenced by the identity of the person. To address this issue, we establish a fixed set of pairs for a consistent comparison across all methods.

**Accuracy (Acc).** As in Naiman et al. (2023), we used this metric for the MUG dataset to evaluate a model's ability to preserve fixed features while generating others. For example, dynamic features are frozen while static features are sampled. Accuracy is computed using a pre-trained classifier, referred to as the "judge", which is trained on the same training set as the model and tested on

the same test set. For the MUG dataset, the classifier checks that the facial expression remains unchanged during the sampling of static features.

**Inception Score (IS).** The Inception Score is a metric used to evaluate the performance of the model generation. First, we apply the judge, to all generated videos $x_0^{1:V}$, obtaining the conditional predicted label distribution $p\left(y|x^{1:V}\right)$. Next, we compute $p(y)$, the marginal predicted label distribution, and calculate the KL-divergence $\text{KL}\left[p\left(y|x_0^{1:V}\right)\|p(y)\right]$. Finally, the Inception Score is computed as $\text{IS} = \exp\left(\mathbb{E}_x\text{KL}\left[p\left(y|x_0^{1:V}\right)\|p(y)\right]\right)$. We use this metric evaluate our results on MUG dataset.

**Inter-Entropy** ($H(y|x_0^{1:V})$)**.** This metric reflects the confidence of the judge in its label predictions, with lower inter-entropy indicating higher confidence. It is calculated by passing $k$ generated sequences $\{x_0^{1:V}\}^{1:k}$ into the judge and computing the average entropy of the predicted label distributions: $\frac{1}{k}\sum_{i=1}^{k} H(p(y|\{x_0^{1:V}\}^i))$. We use this metric evaluate our results on MUG dataset.

**Intra-Entropy** ($H(y)$)**.** This metric measures the diversity of the generated sequences, where a higher intra-entropy score indicates greater diversity. It is computed by sampling from the learned prior distribution $p(y)$ and then applying the judge to the predicted labels $y$. We use this metric to evaluate our results on the MUG dataset.

**EER.** Equal Error Rate (EER) metric is widely employed in speaker verification tasks. The EER represents the point at which the false positive rate equals the false negative rate, offering a balanced measure of performance in speaker recognition. This metric, commonly applied to the TIMIT dataset, provides a robust evaluation of the model's ability to disentangle features relevant to speaker identity.

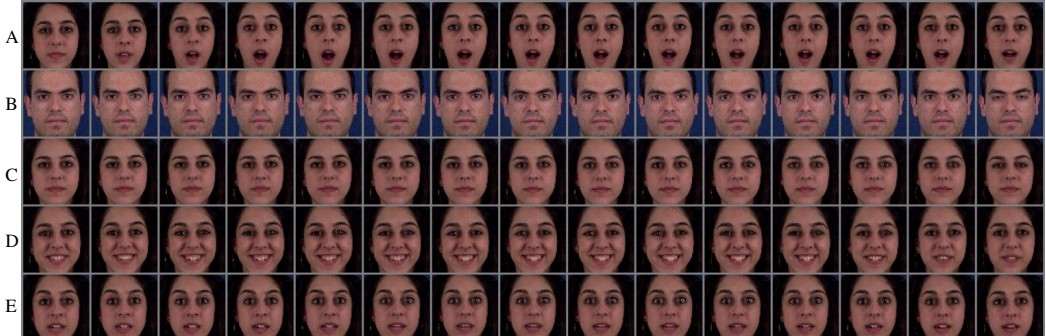

Figure 7: Rows A and B are two inputs from the test set. Row C shows a dynamic swap example, using the static of A and dynamics of B. In row D we extract the same person from A, but with the dynamics as labeled in B. Finally, in row E, we extract the same person from A with the dynamics that are predicted by the classifier.

# E  MUG AND JUDGE METRIC ANALYSIS

While our results show significant improvement over previous methods on VoxCeleb (Nagrani et al., 2017), CelebV-HQ (Zhu et al., 2022), and TaiChi-HD (Siarohin et al., 2019), both in terms of disentanglement and reconstruction, our performance on MUG (Aifanti et al., 2010) is only on par with the state-of-the-art methods. Since MUG is a labeled dataset, the traditional evaluation task involves the unconditional generation of static factors while freezing the dynamics, resulting in altering the appearance of the person. The generated samples are then evaluated using an off-the-shelf judge model (See App. D), which is a neural network trained to classify both static and dynamic factors. If the disentanglement method disentangles these factors effectively, we expect the judge to correctly identify the dynamics while outputting different predictions for the static features, since the latter were randomly sampled and should differ from the original static factor.

Figure 8: Rows A and B are two inputs from the test set. Row C shows a dynamic swap example, using the static of A and dynamics of B. In row D we extract the same person from A, but with the dynamics as labeled in B. Finally, in row E, we extract the same person from A with the dynamics that are predicted by the classifier.

Table 6: Judge benchmark disentanglement metrics on MUG.

| Method | Acc↑ | IS↑ | $H(y\|x)$↓ | $H(y)$↑ | Reconstruction (MSE) ↓ |
|---|---|---|---|---|---|
| | | | MUG | | |
| MoCoGAN | 63.12% | 4.332 | 0.183 | 1.721 | – |
| DSVAE | 54.29% | 3.608 | 0.374 | 1.657 | – |
| R-WAE | 71.25% | 5.149 | 0.131 | 1.771 | – |
| S3VAE | 70.51% | 5.136 | 0.135 | 1.760 | – |
| SKD | 77.45% | 5.569 | 0.052 | 1.769 | – |
| C-DSVAE | 81.16% | 5.341 | 0.092 | 1.775 | – |
| SPYL | 85.71% | 5.548 | 0.066 | 1.779 | 1.311e−3 |
| DBSE | **86.90%** | **5.598** | **0.041** | **1.782** | 1.286e−3 |
| Ours | 81.15% | 5.382 | 0.090 | 1.773 | **2.669e−7** |

Surprised by our results on MUG, we investigated the failure cases to understand the limitations of our model. In particular, we examined scenarios where we freeze the dynamics and swap the static features between two samples, and then we generate the corresponding output. In Fig. 7, we show an example where the static features of the second row are swapped with those of the first row, and the resulting generation is displayed in the third row. We observe that while the dynamics from the second row are well-preserved, the generated person retains the identity of the first row. However, the classifier incorrectly predicts the dynamics for the sequence. To further investigate this, we extracted a ground-truth example of the person from the first row in the dataset expressing the expected emotion and the predicted one. In the last two rows of Fig. 7, we show the same person with predicted dynamics (fourth row) and the same person with the dynamics that the classifier predicted (fifth row). We provide another example of the same phenomenon in Fig. 8.

We observe that while the judge predicts the wrong label for our generated samples in rows C, the facial expressions of the people there align better with the actual dynamics in rows B. This suggests that the classifier is biased towards the identity when predicting dynamics, potentially forming a discrete latent space where generalization to nearby related expressions is not possible. Importantly, the judge attains $> 99\%$ accuracy on the test set. We conclude that utilizing a judge can be problematic for measuring new and unseen variations in the data. This analysis motivates us to present the AKD and AED, as detailed above in App. D.

# F  ADDITIONAL RESULTS

## F.1  RECONSTRUCTION RESULTS

In Figs. 9 to 12, we present several qualitative reconstruction examples across all datasets.

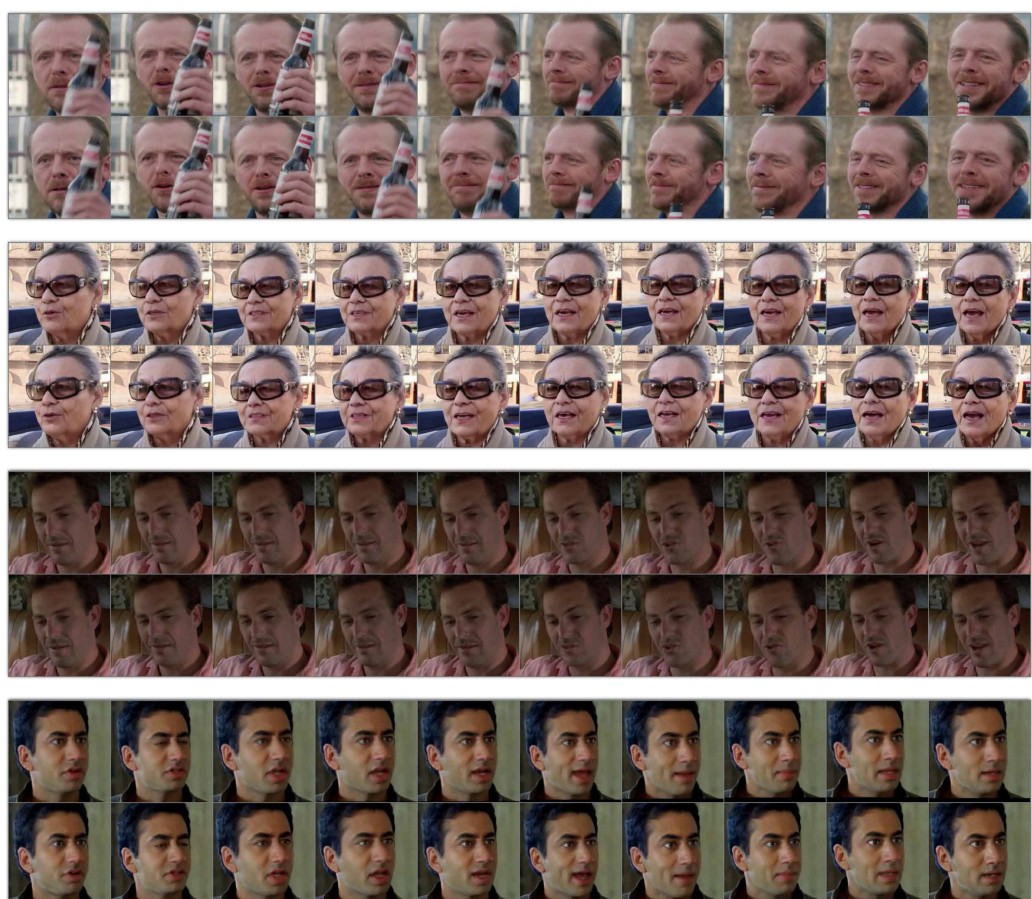

Figure 9: Reconstruction results of CelebV-HQ ($256 \times 256$). The first row for each pair is the original video and the second row is its reconstruction.

### F.2 ADDITIONAL RESULTS: CONDITIONAL SWAP

In what follows, we present more results for the conditional swapping experiment from the main text (Sec. 5.1). In each figure, the first two rows show the original sequences (real videos). The third and fourth rows are the results of the conditional swap where we change the dynamic and static factors, respectively. We show our results for all datasets in Figs. 13 to 16.

### F.3 ADDITIONAL RESULTS: UNCONDITIONAL SWAP

In this section, we present more results for the unconditional swapping experiment from the main text (Sec. 5.4). Each figure is composed of separate panels. In each panel, the middle row represents the original sequence. In the top row, we sample new dynamic factors and freeze the static factor. In the bottom row below, we sample a new static factor and freeze the dynamics. We show our results on all datasets in Figs. 17 to 20.

### F.4 ADDITIONAL RESULTS: ZERO-SHOT DISENTANGLEMENT

Here we extend the results from Sec. 5.2. We provide additional examples of conditional swapping when the model is trained on one dataset and evaluated on another dataset, unseen during training. Specifically, in Fig. 21, we show examples where the model is trained on VoxCeleb and tested on MUG. Additionally, in Fig. 22, the model is trained on VoxCeleb and tested on CelebV-HQ. Finally, in Fig. 23, the model is trained on CelebV-HQ and tested on VoxCeleb.

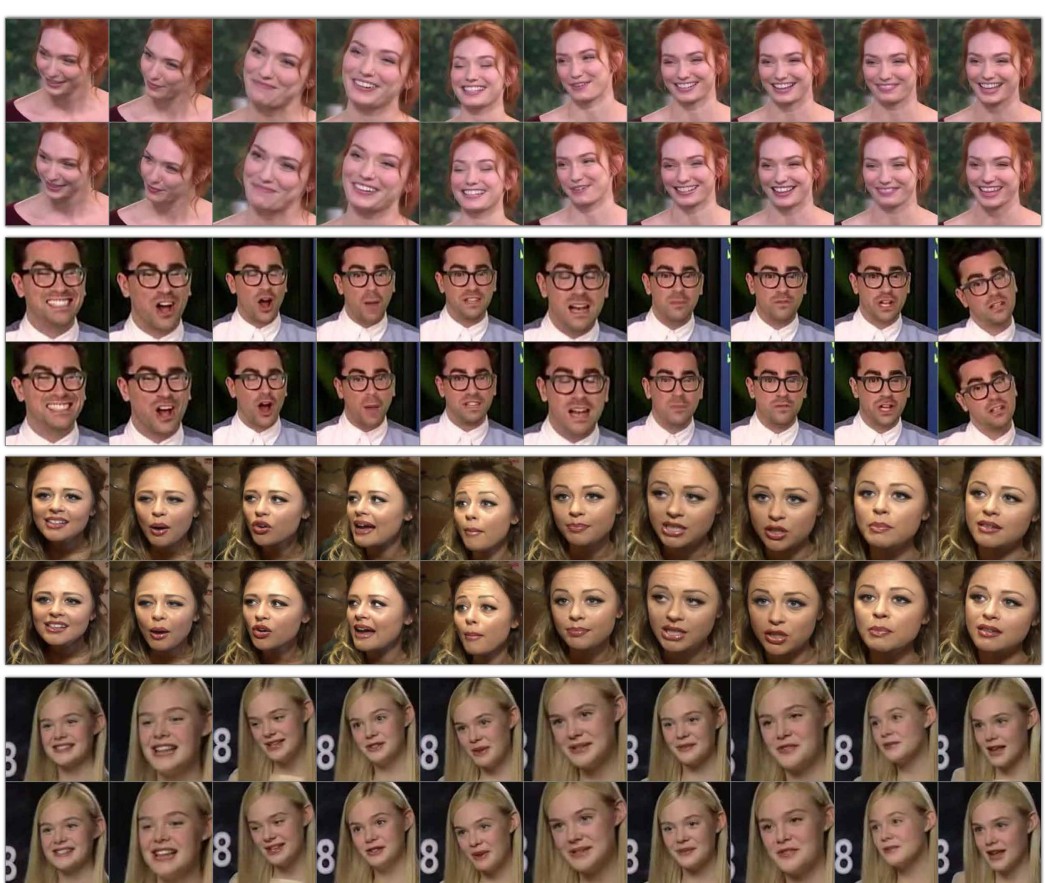

Figure 10: Reconstruction results of VoxCeleb ($256 \times 256$). The first row for each pair is the original video and the second row is its reconstruction.

### F.5 ADDITIONAL RESULTS: MULTIFACTOR DISENTANGLEMENT

In this section, we present more examples for traversing the latent space, separately for the static and dynamic factors. For static factors, we show in Figs. 24 to 35. There, we find different factors of variation such as Male to Female, younger to older, brighter and darker hair color, and more. Each row in the figure is a video, and the different columns represent the traversal in $\alpha$ values (see Eq. 9). In addition, we present full examples of dynamic factor traversal in Figs. 36 to 47, demonstrating various factors of variation. Among the factors are facial expressions, camera angles, head rotations, eyes and mouth control, etc.

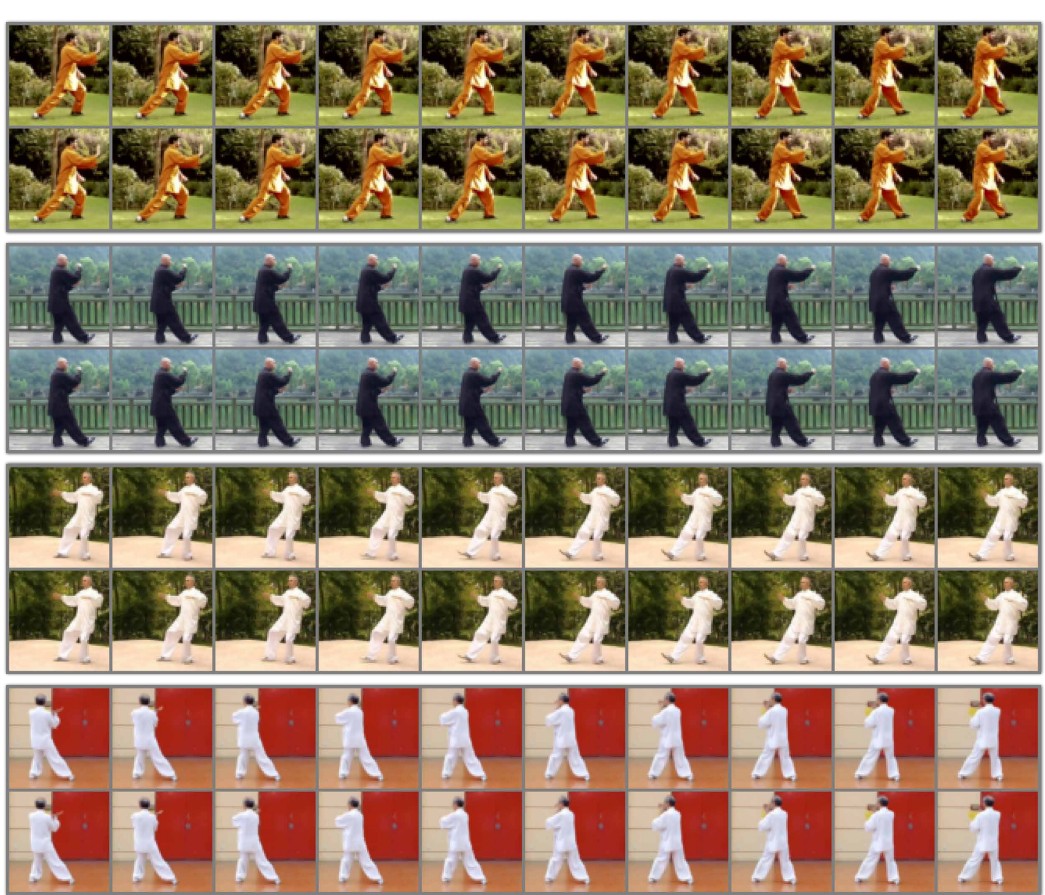

Figure 11: Reconstruction results of TaiChi-HD. The first row for each pair is the original video and the second row is its reconstruction.

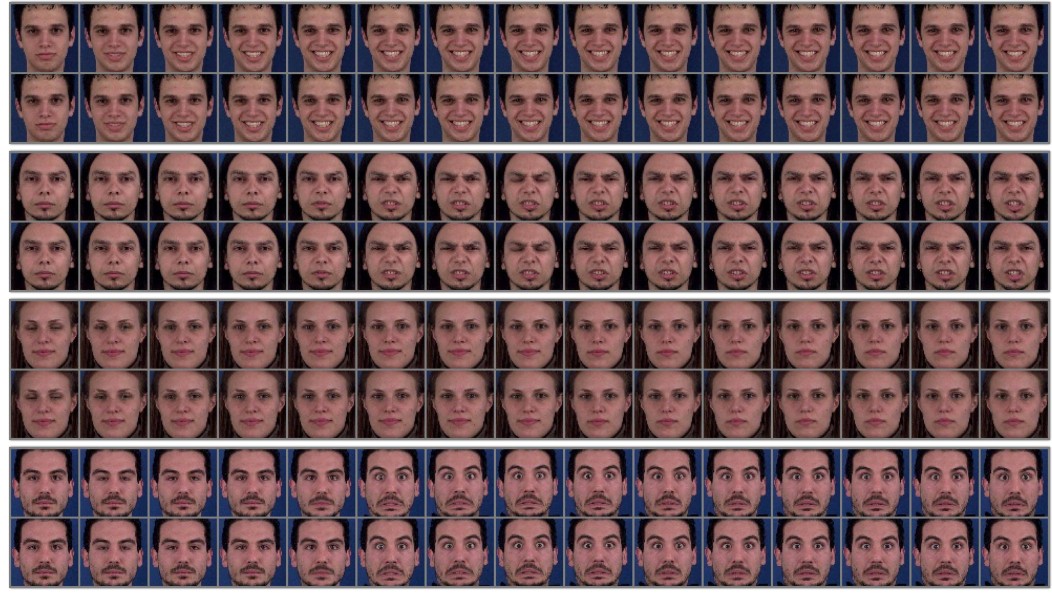

Figure 12: Reconstruction results of MUG. The first row for each pair is the original video and the second row is its reconstruction.

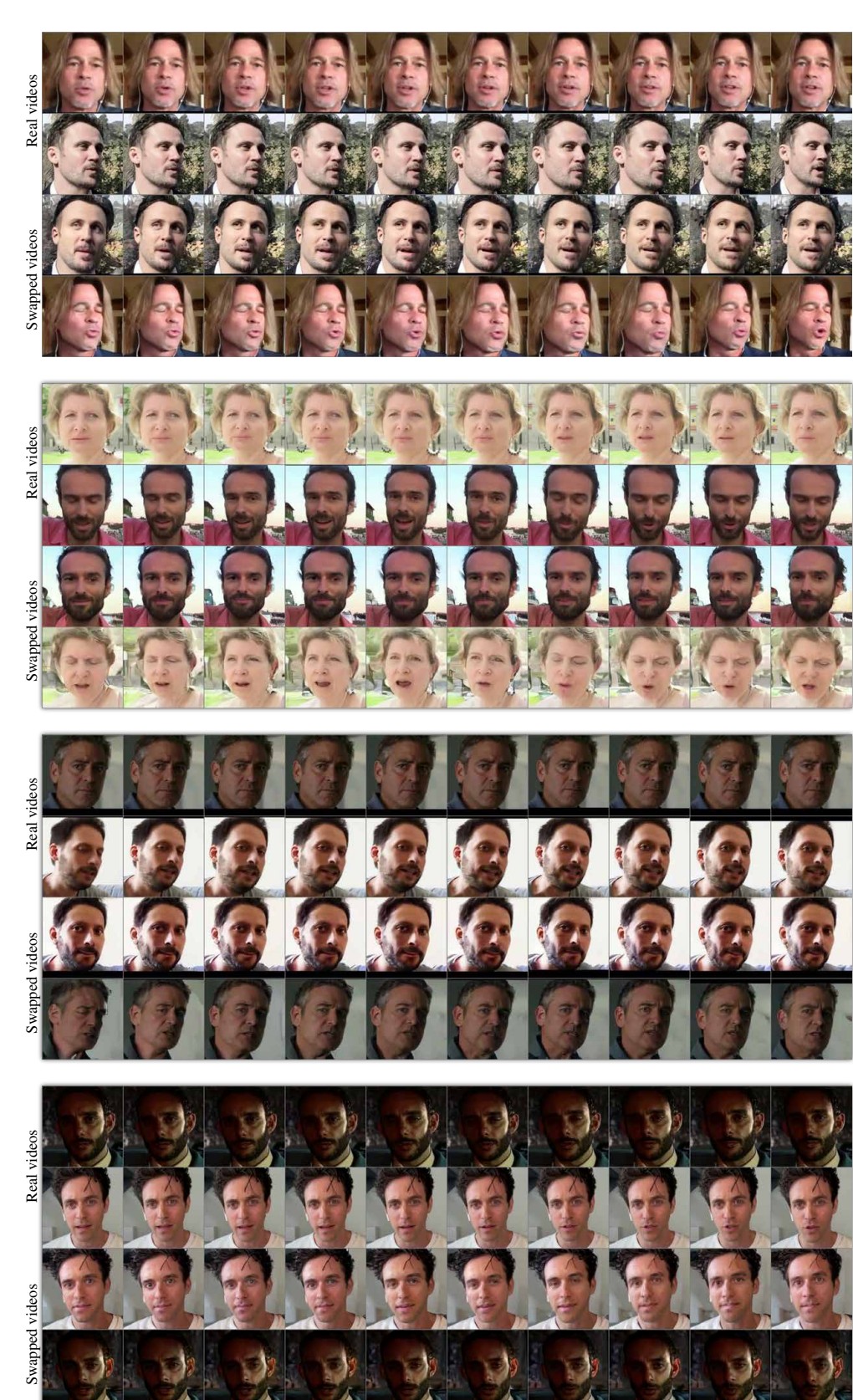

Figure 13: Each panel contains a pair of original videos from CelebV-HQ (Real videos), and a pair of conditional swapping of the dynamic and static factors (Swapped videos).

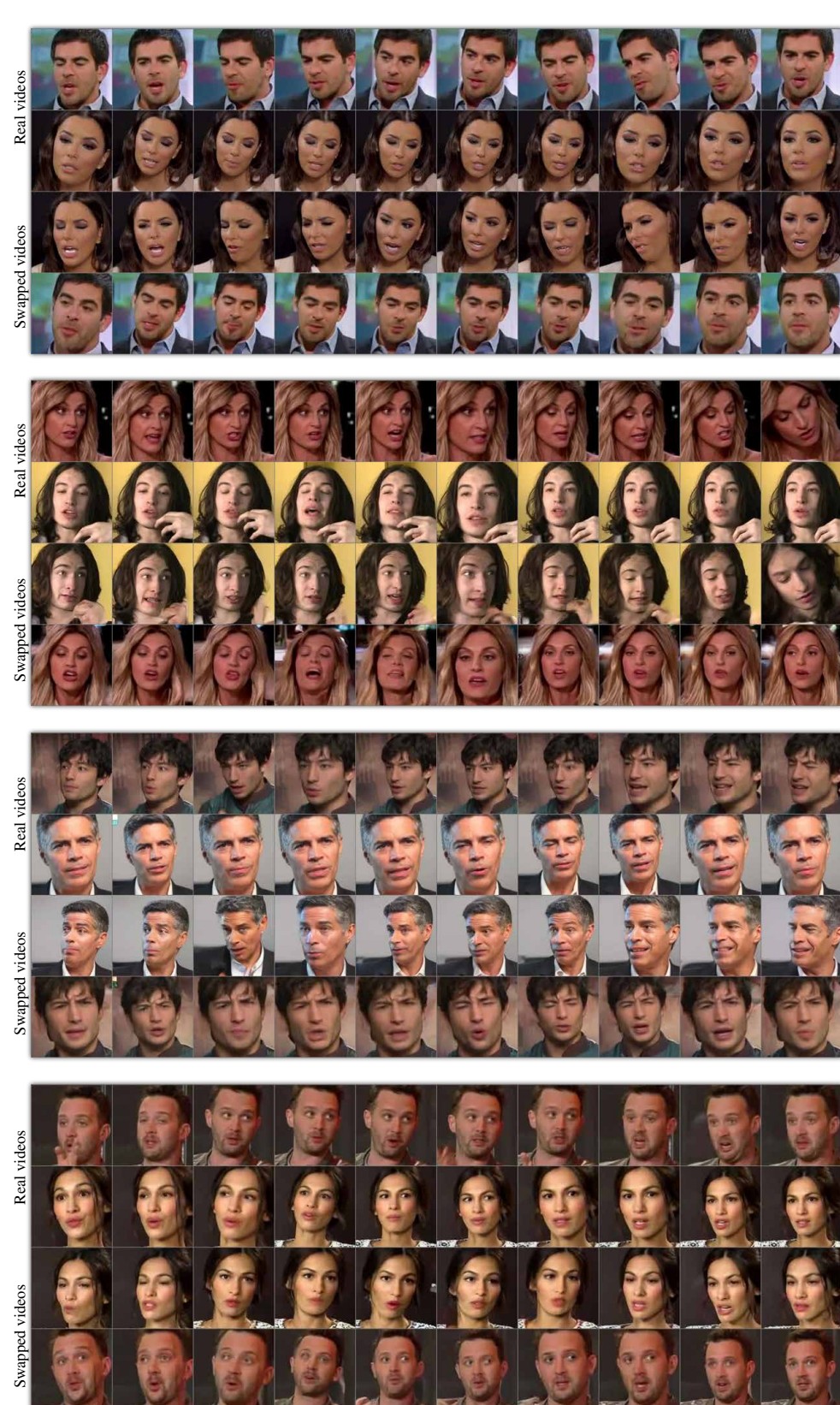

Figure 14: Each panel contains a pair of original videos from VoxCeleb (Real videos), and a pair of conditional swapping of the dynamic and static factors (Swapped videos).

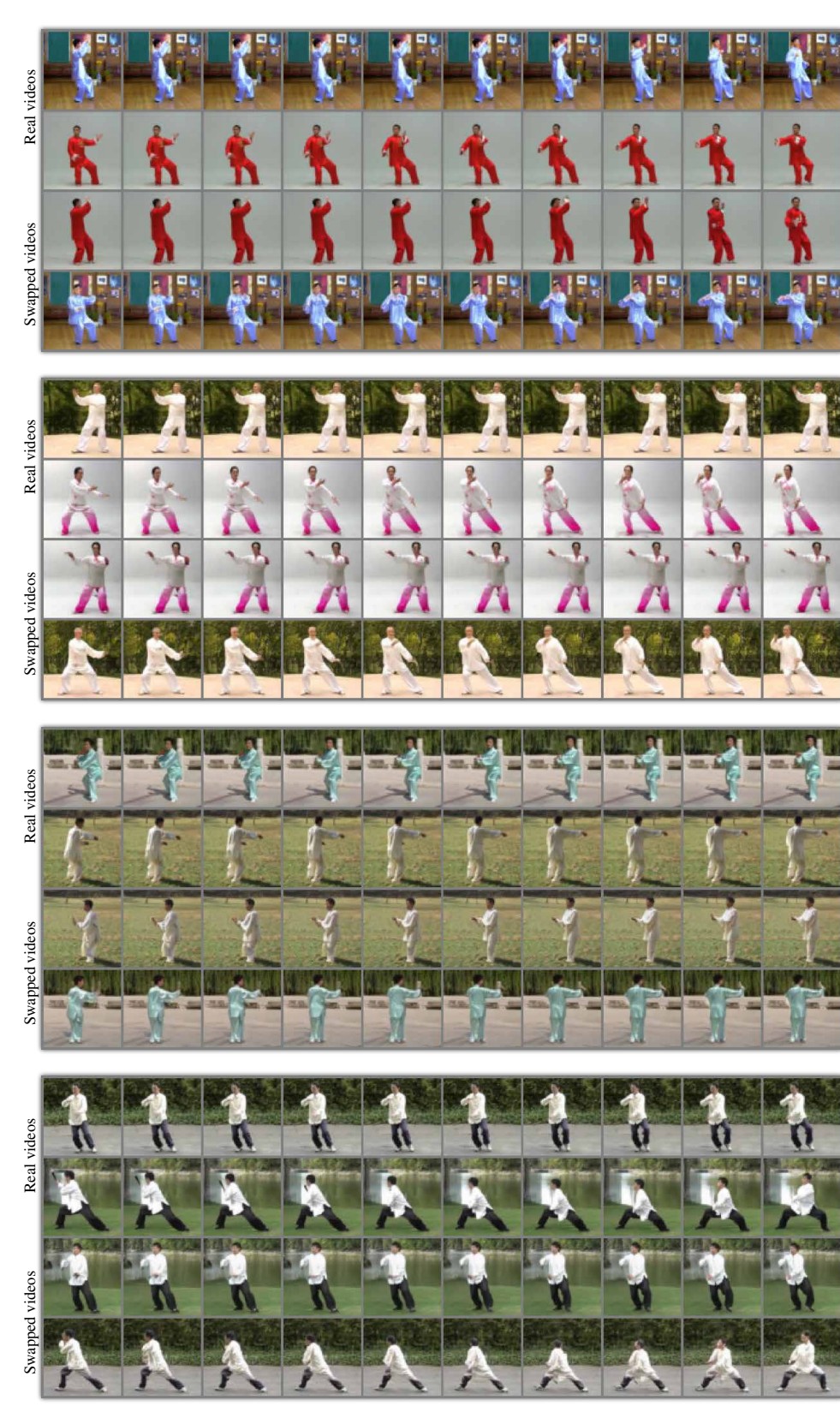

Figure 15: Each panel contains a pair of original videos from TaiChi-HD (Real videos), and a pair of conditional swapping of the dynamic and static factors (Swapped videos).

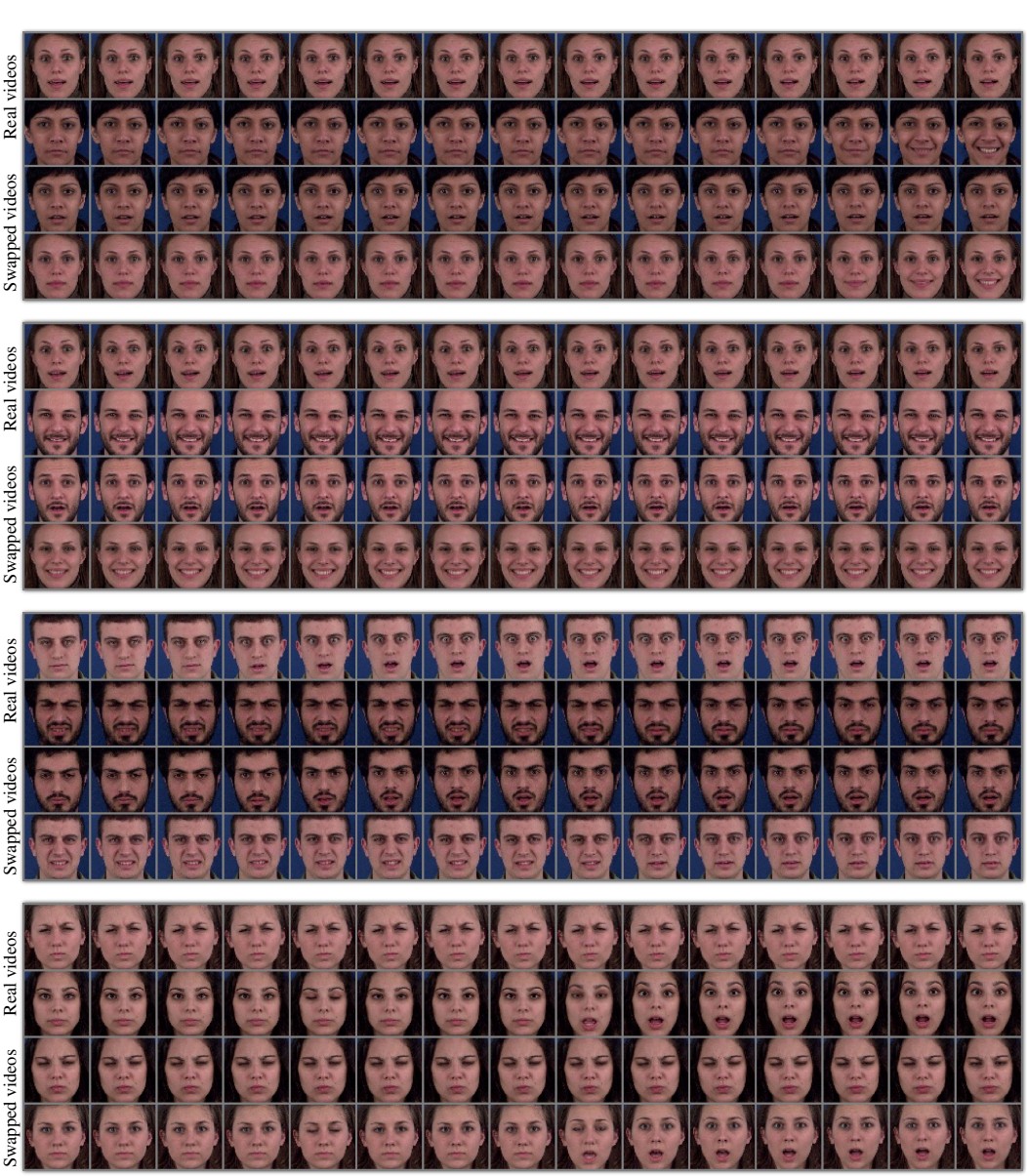

Figure 16: Each panel contains a pair of original videos from MUG (Real videos), and a pair of conditional swapping of the dynamic and static factors (Swapped videos).

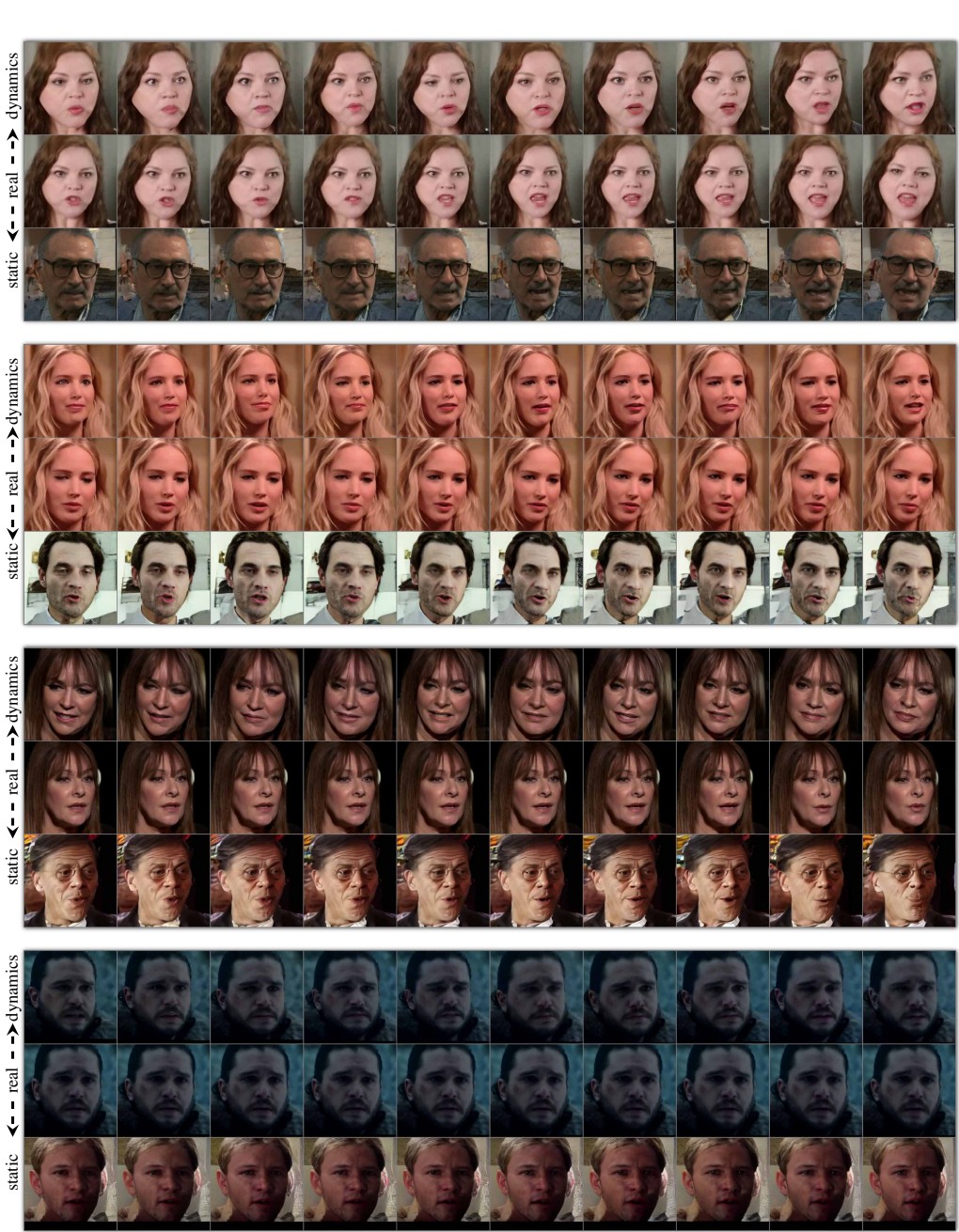

Figure 17: CelebV-HQ unconditional swapping. The middle row represents the original video (real), the row above shows a dynamic swap (dynamics), and the row below shows a static swap (static).

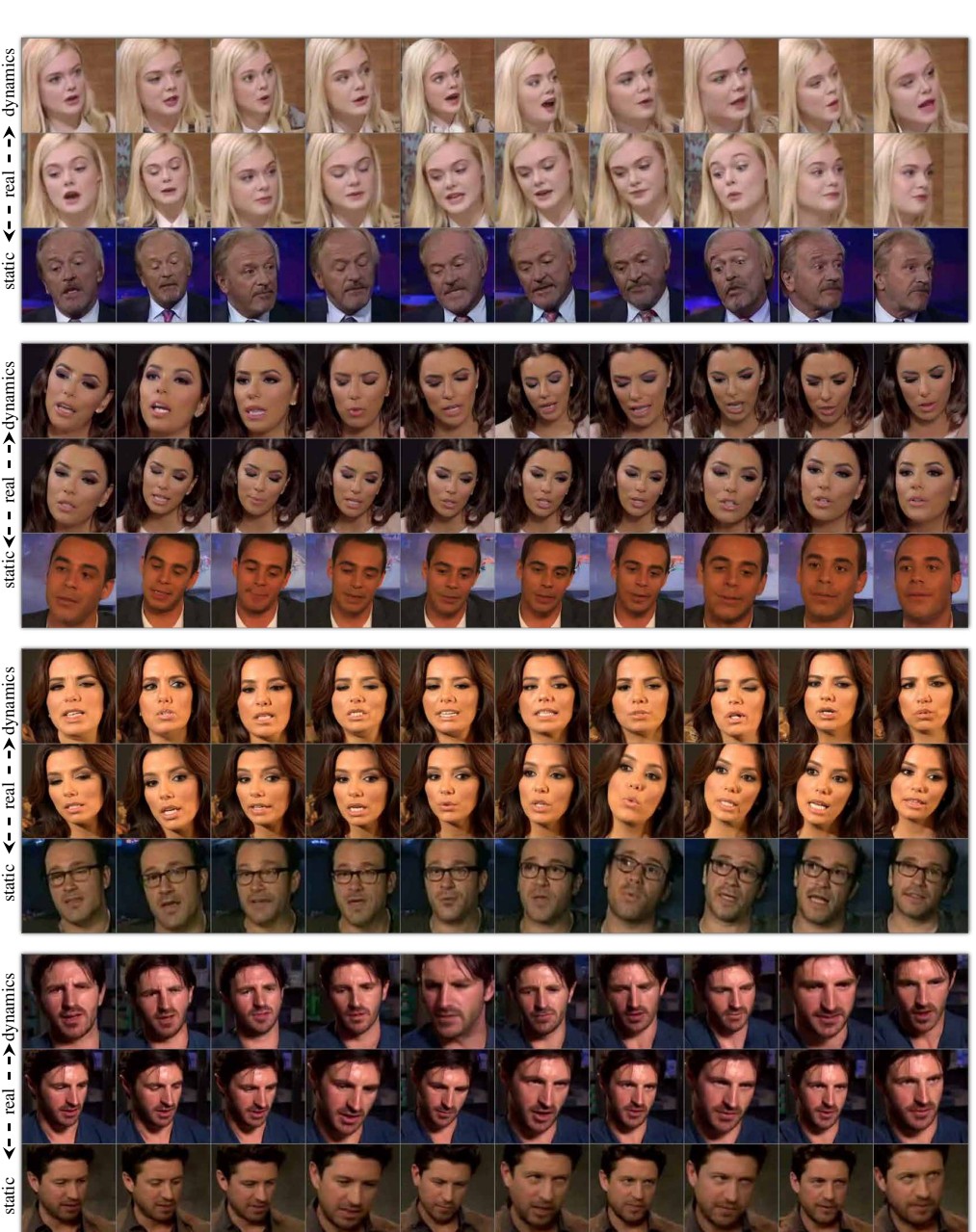

Figure 18: VoxCeleb unconditional swapping. The middle row represents the original video (real), the row above shows a dynamic swap (dynamics), and the row below shows a static swap (static).

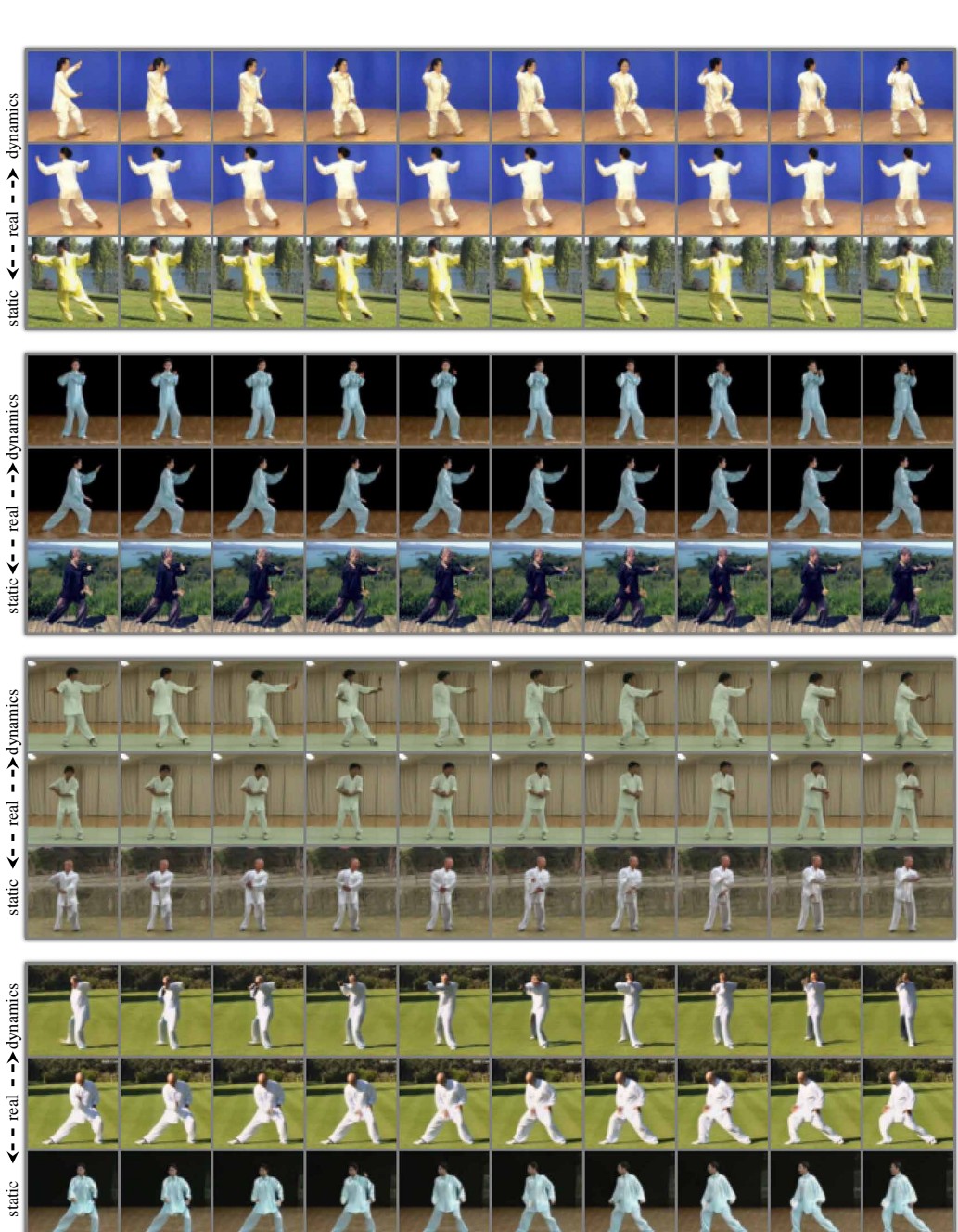

Figure 19: TaiChi-HD unconditional swapping. The middle row represents the original video (real), the row above shows a dynamic swap (dynamics), and the row below shows a static swap (static).

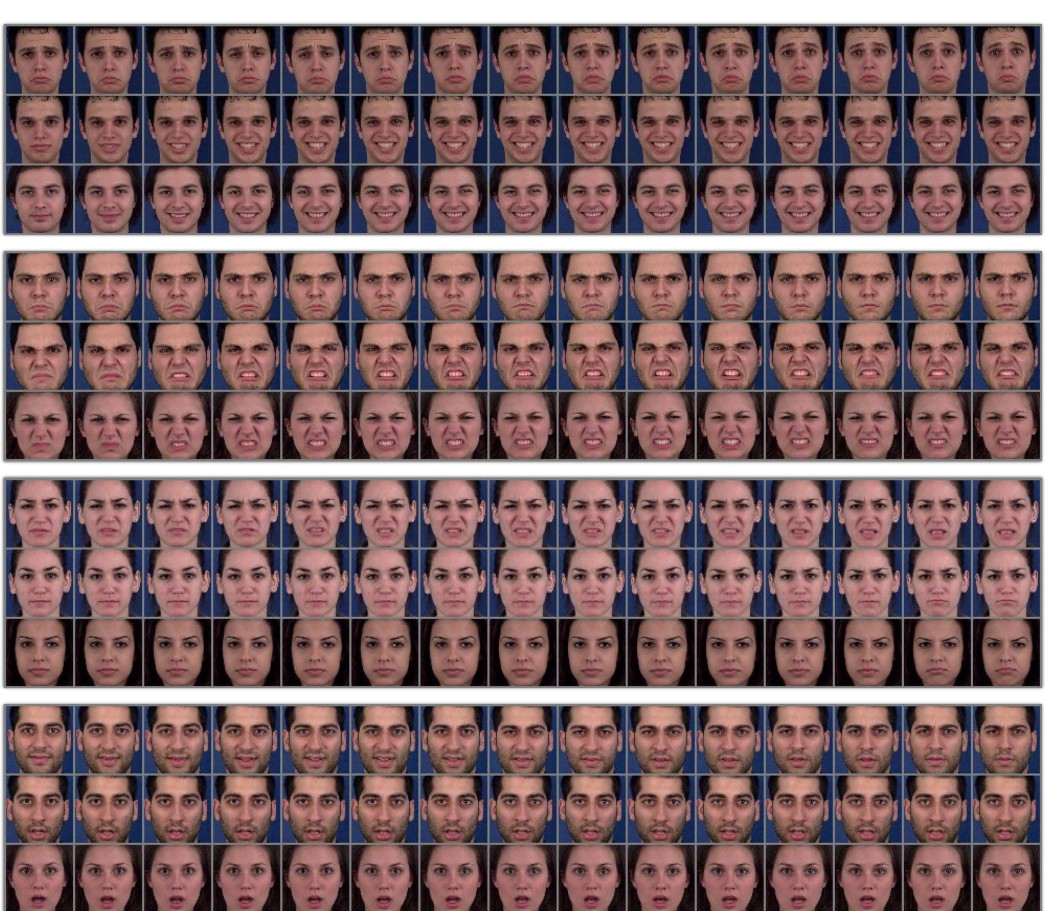

Figure 20: MUG unconditional swapping. The middle row represents the original video (real), the row above shows a dynamic swap (dynamics), and the row below shows a static swap (static).

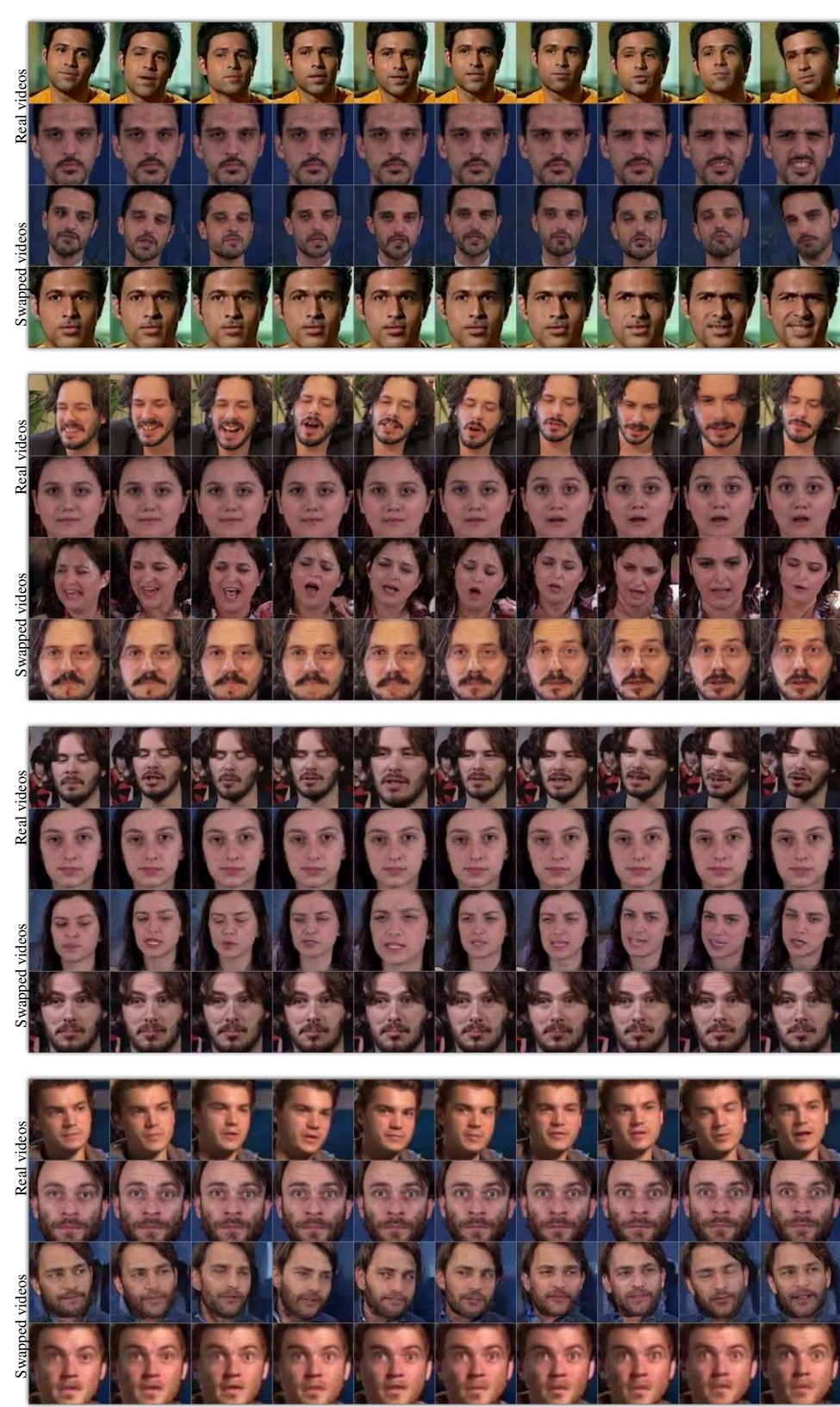

Figure 21: Each panel contains in its first and second rows a pair of real videos from VoxCeleb and MUG, respectively. We perform conditional swapping using a model that was trained on VoxCeleb, but we zero-shot swap the dynamic and static factors of a MUG example (Swapped videos).

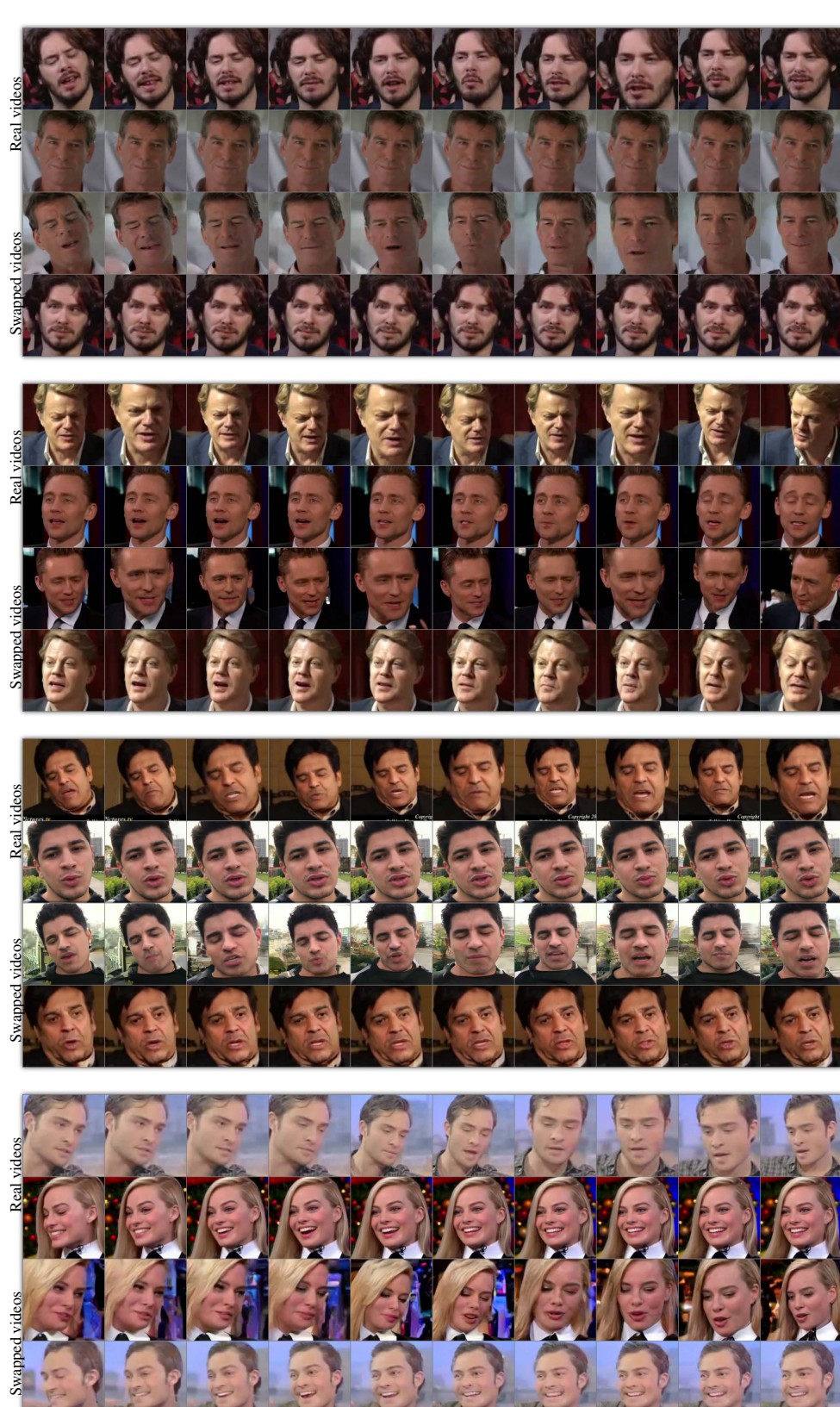

Figure 22: Each panel contains in its first and second rows a pair of real videos from VoxCeleb and CelebV-HQ. We perform conditional swapping using a model that was trained on VoxCeleb, but we zero-shot swap the dynamic and static factors of a CelebV-HQ example (Swapped videos).

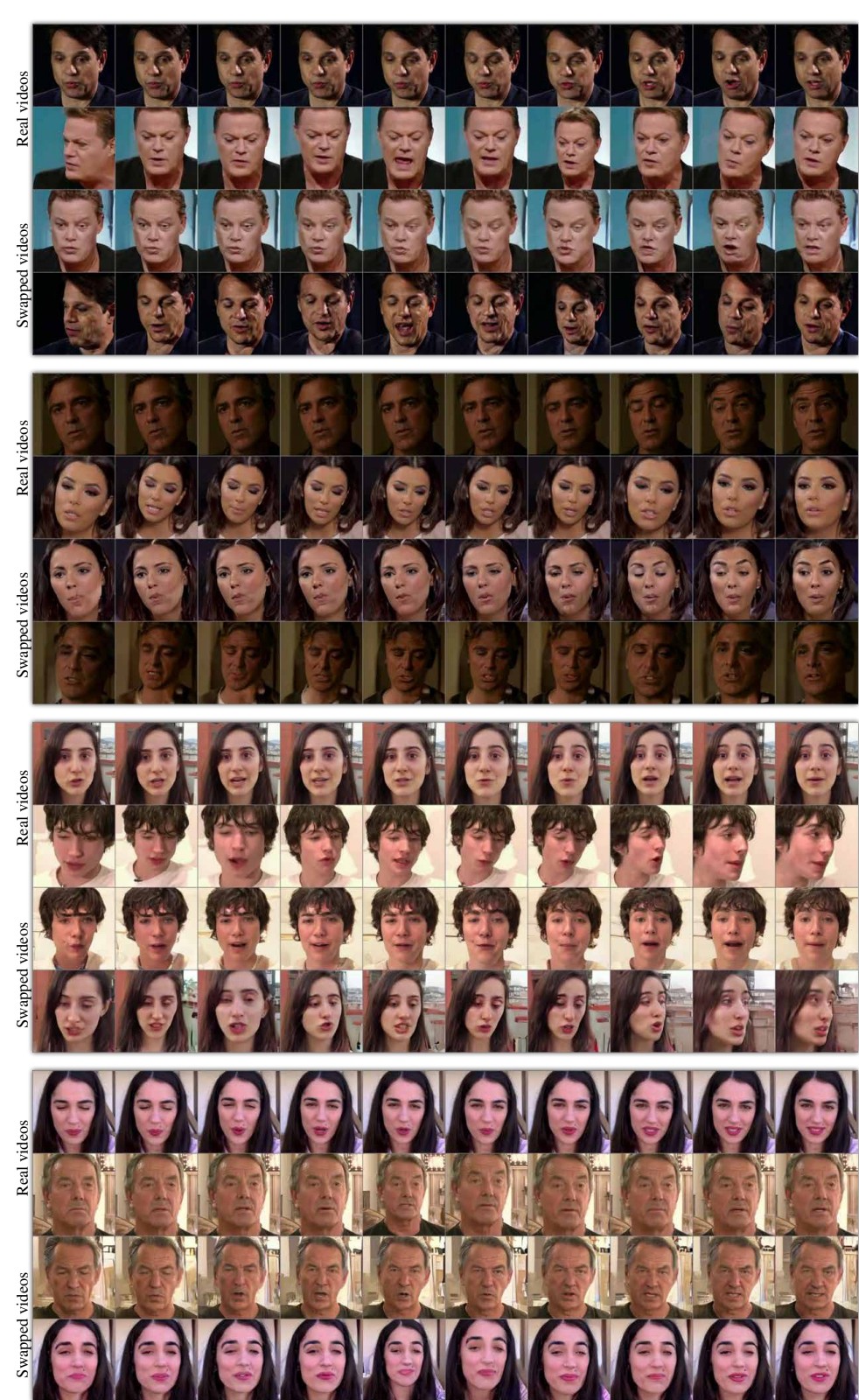

Figure 23: Each panel contains in its first and second rows a pair of real videos from CelebV-HQ and VoxCeleb. We perform conditional swapping using a model that was trained on CelebV-HQ, but we zero-shot swap the dynamic and static factors of a VoxCeleb example (Swapped videos).

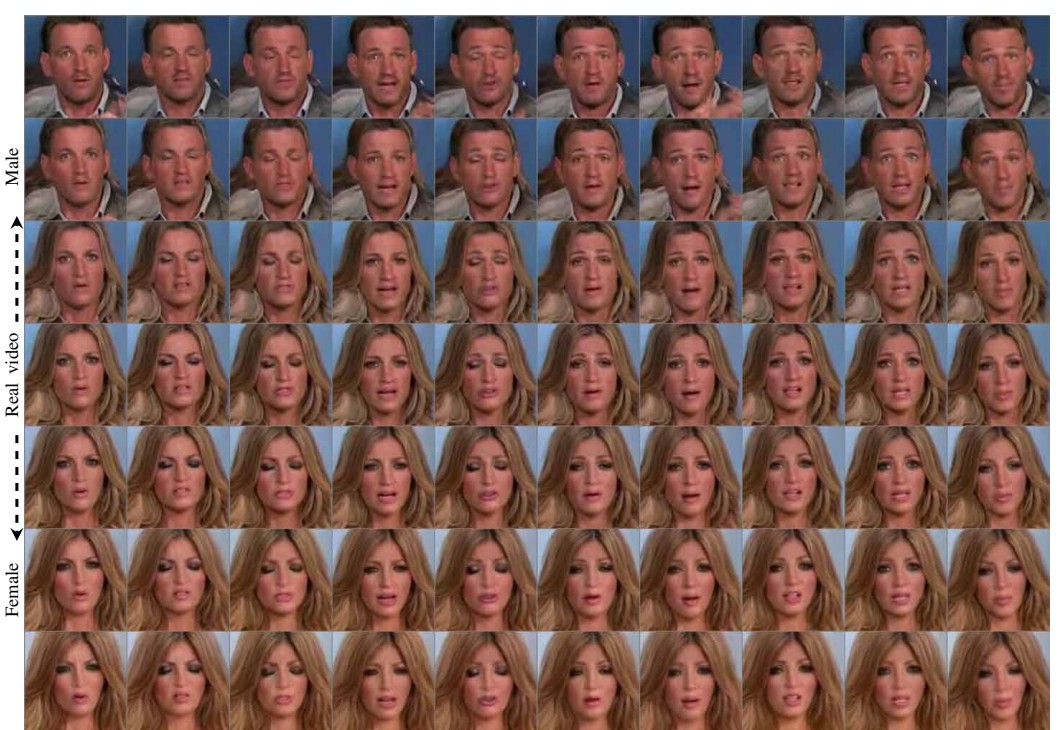

Figure 24: Traversing between Male appearances and Female appearances.

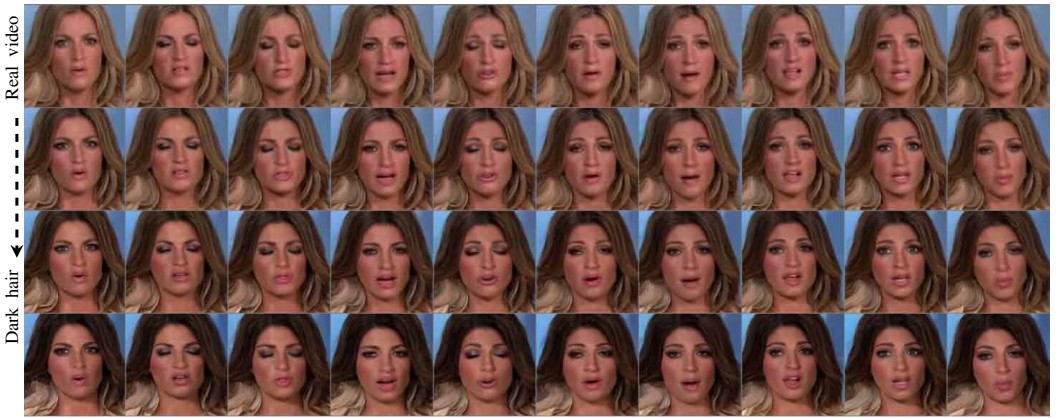

Figure 25: Traversing over a darker hair factor.

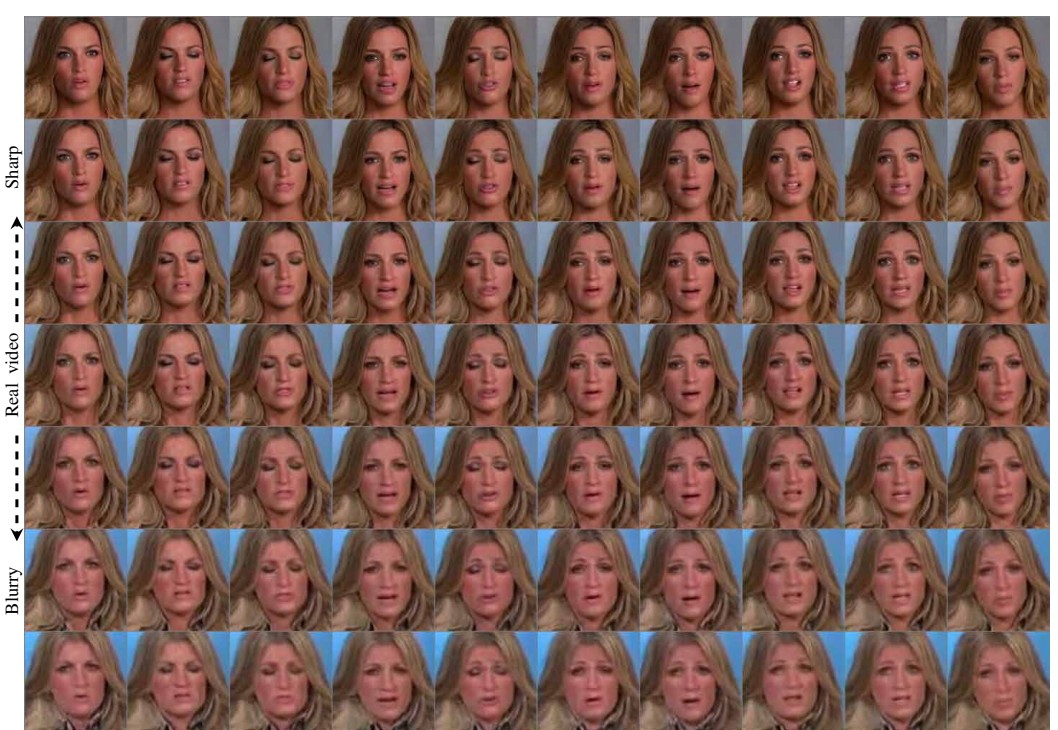

Figure 26: Traversing between sharper and blurry videos.

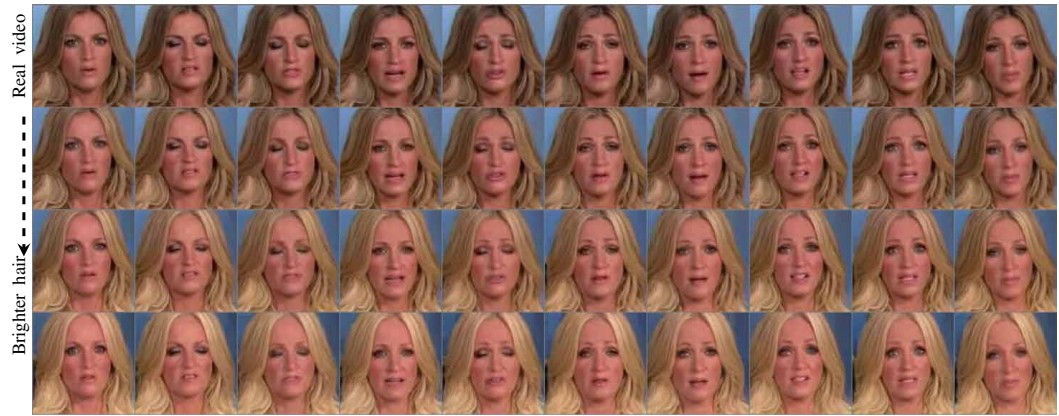

Figure 27: Traversing over a brighter hair factor.

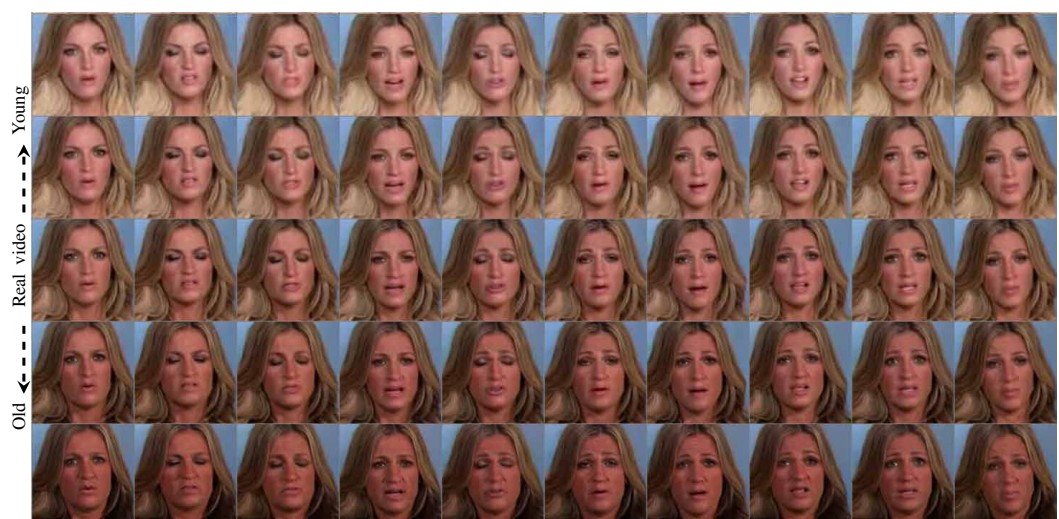

Figure 28: Traversing between younger and older appearances.

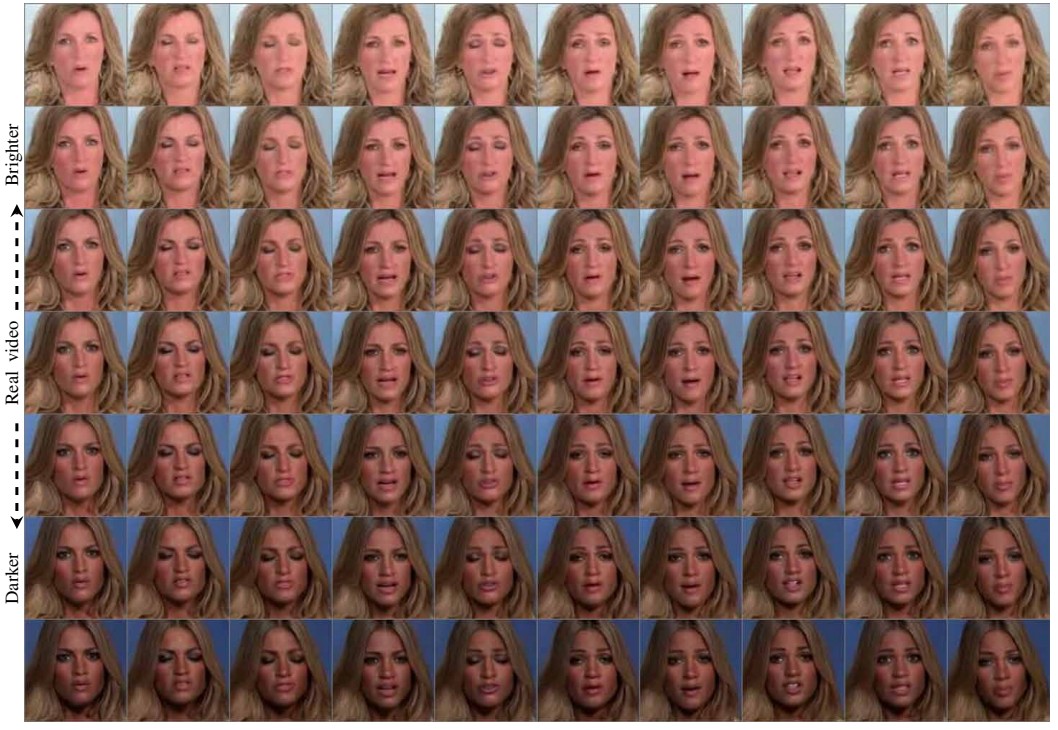

Figure 29: Traversing over skin color variations.

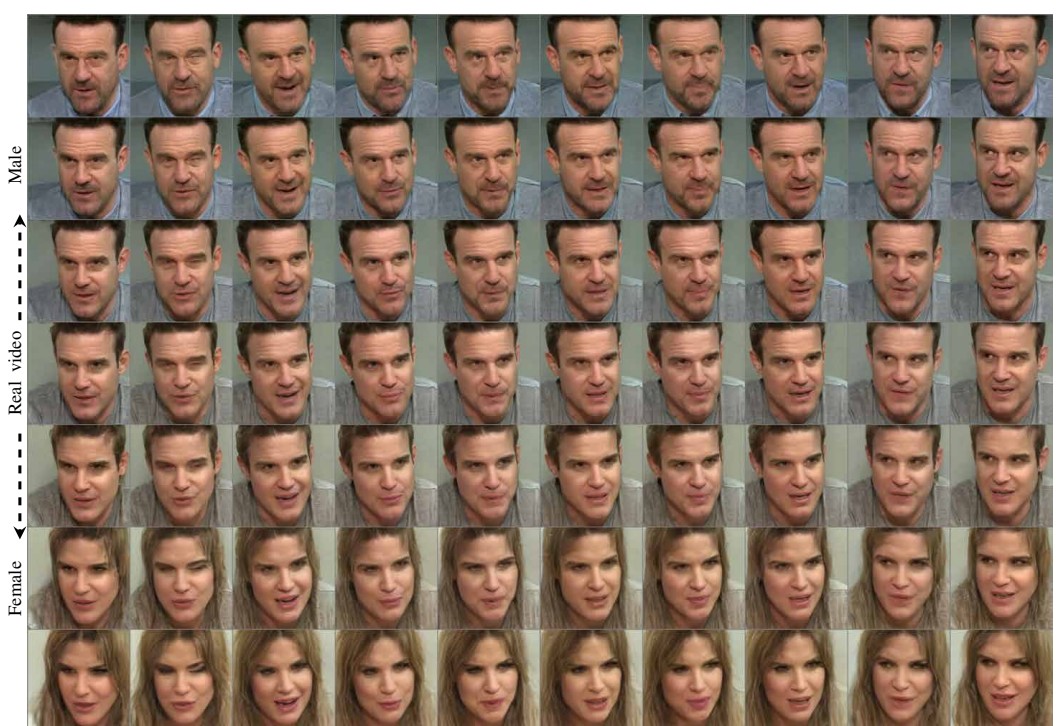

Figure 30: Traversing between Male appearances and Female appearances.

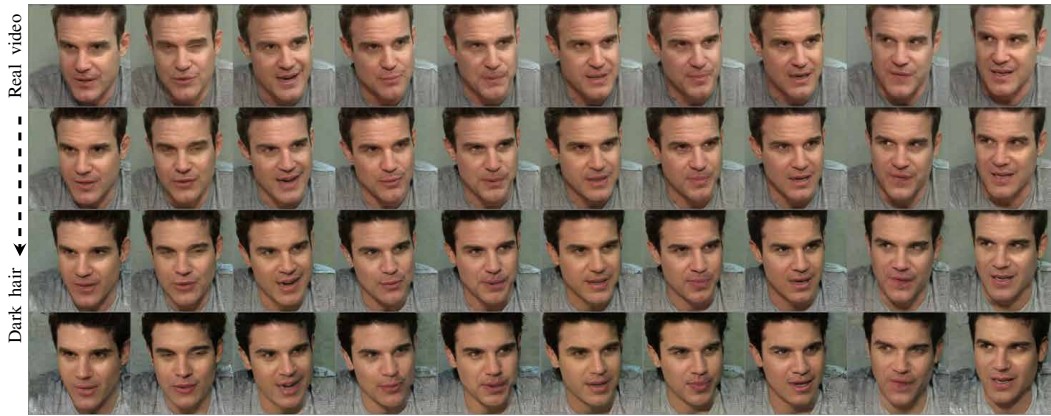

Figure 31: Traversing over a darker hair factor.

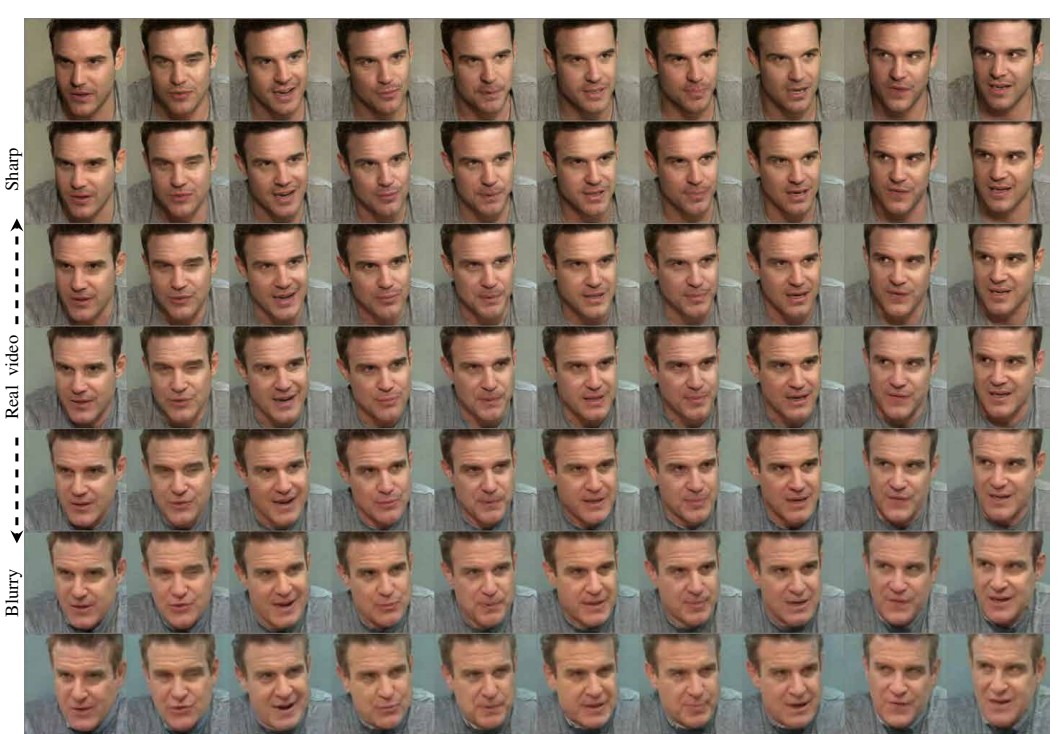

Figure 32: Traversing between sharper and blurry videos.

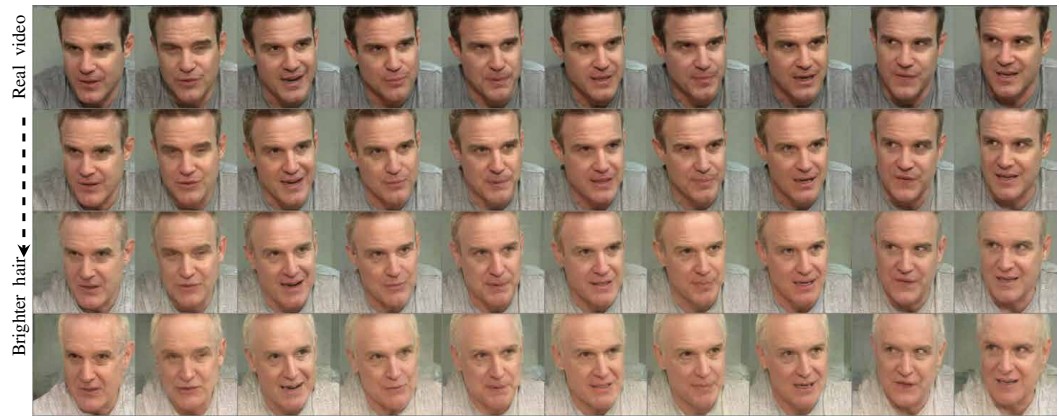

Figure 33: Traversing over a brighter hair factor.

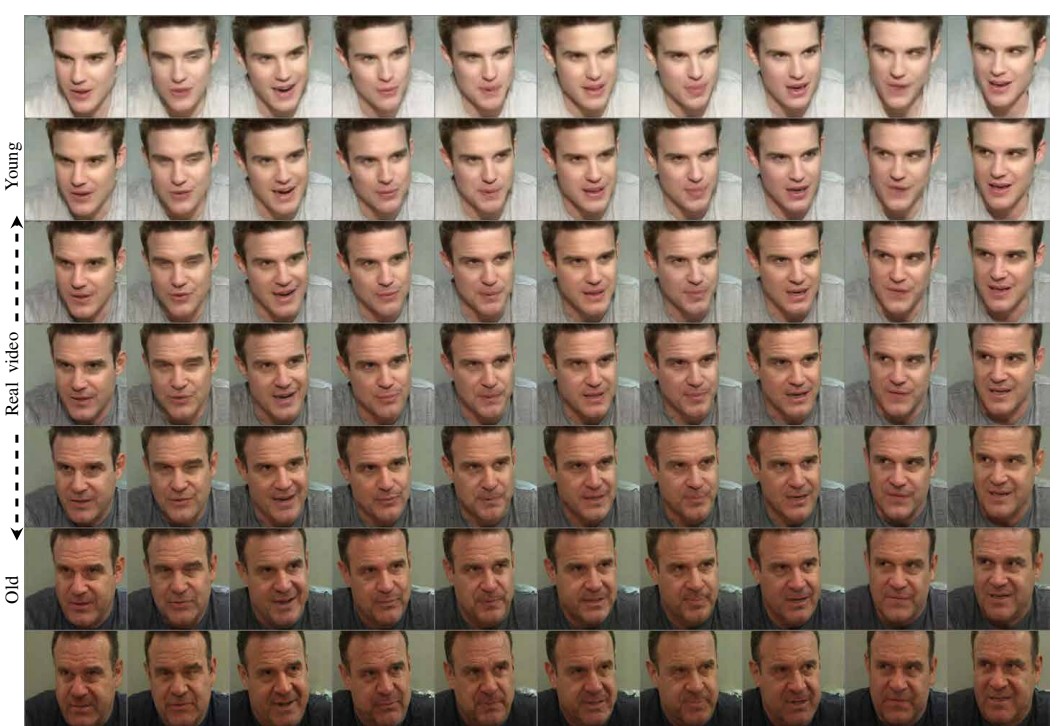

Figure 34: Traversing between younger and older appearances.

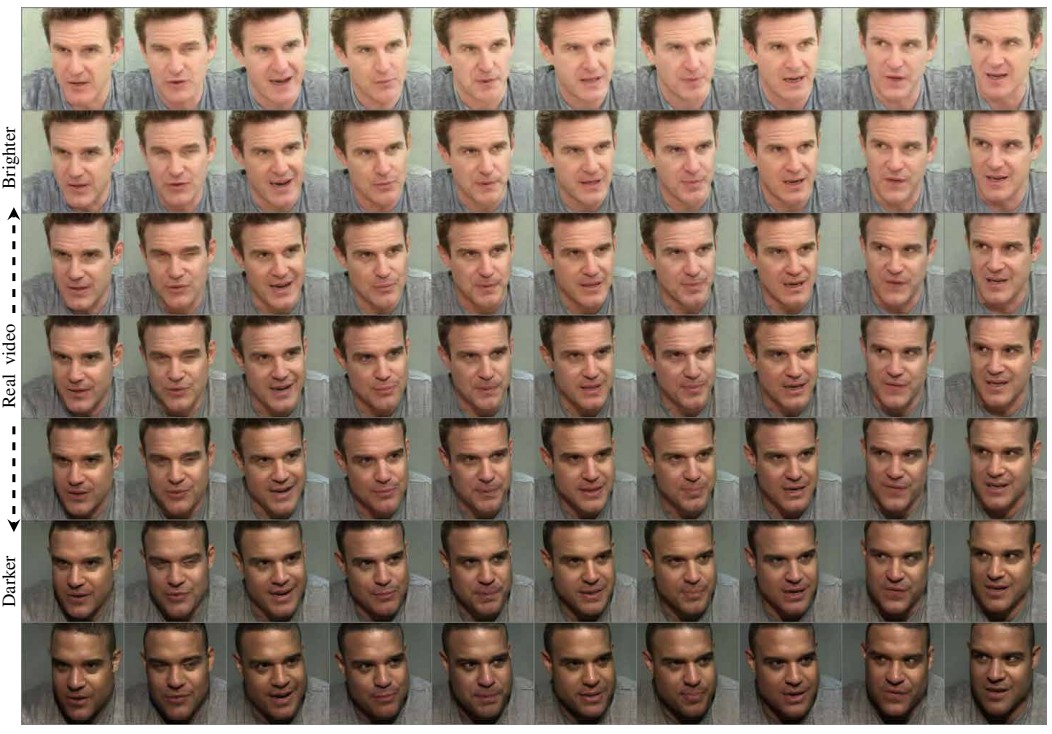

Figure 35: Traversing over skin color variations.

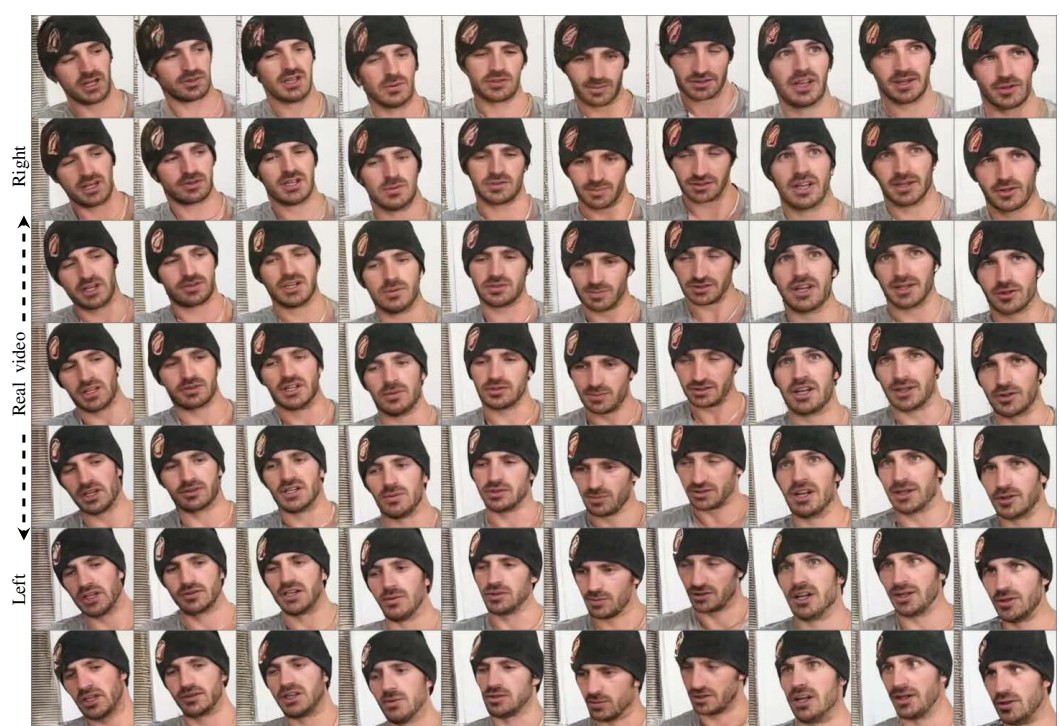

Figure 36: Traversing a head rotation factor.

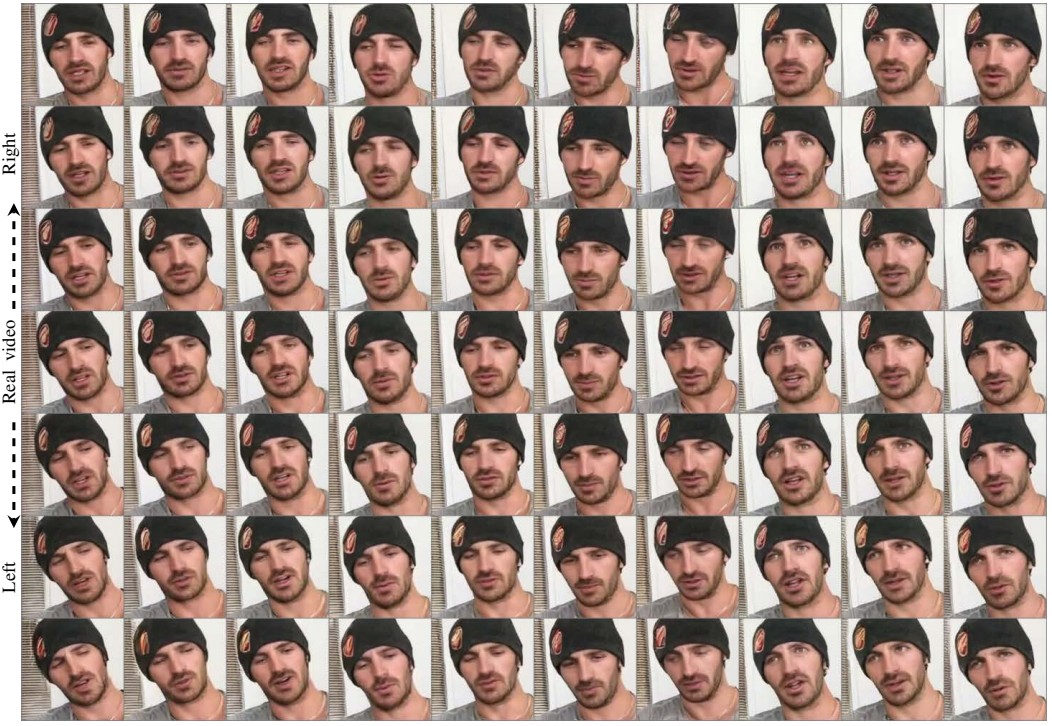

Figure 37: Traversing over head angles.

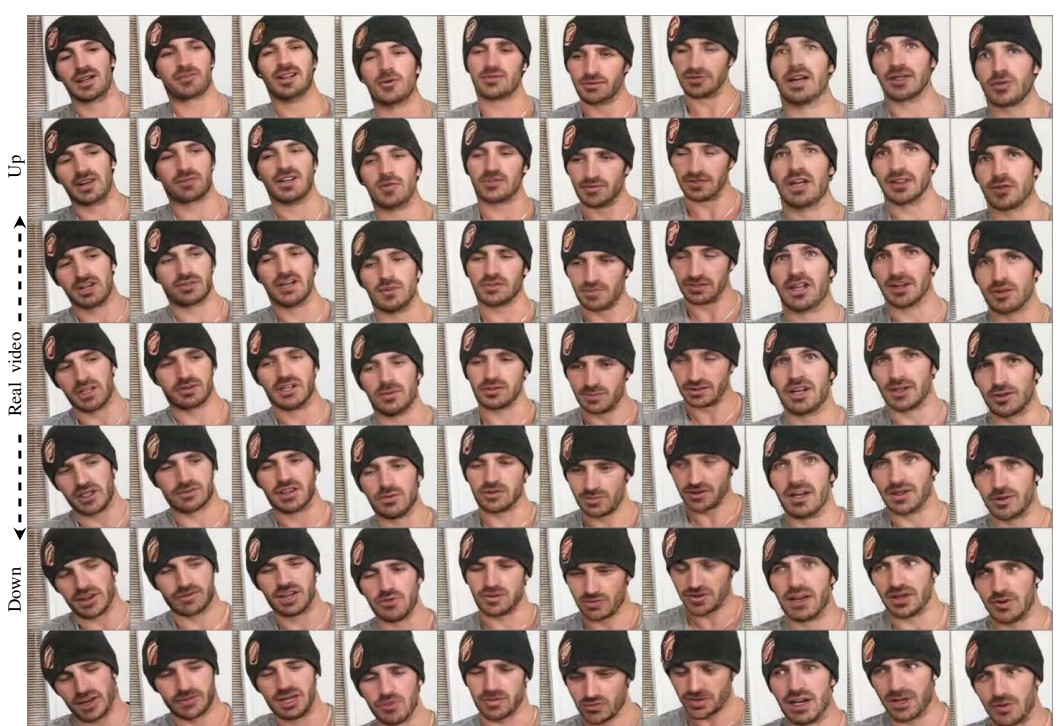

Figure 38: Traversing over up and down rotations.

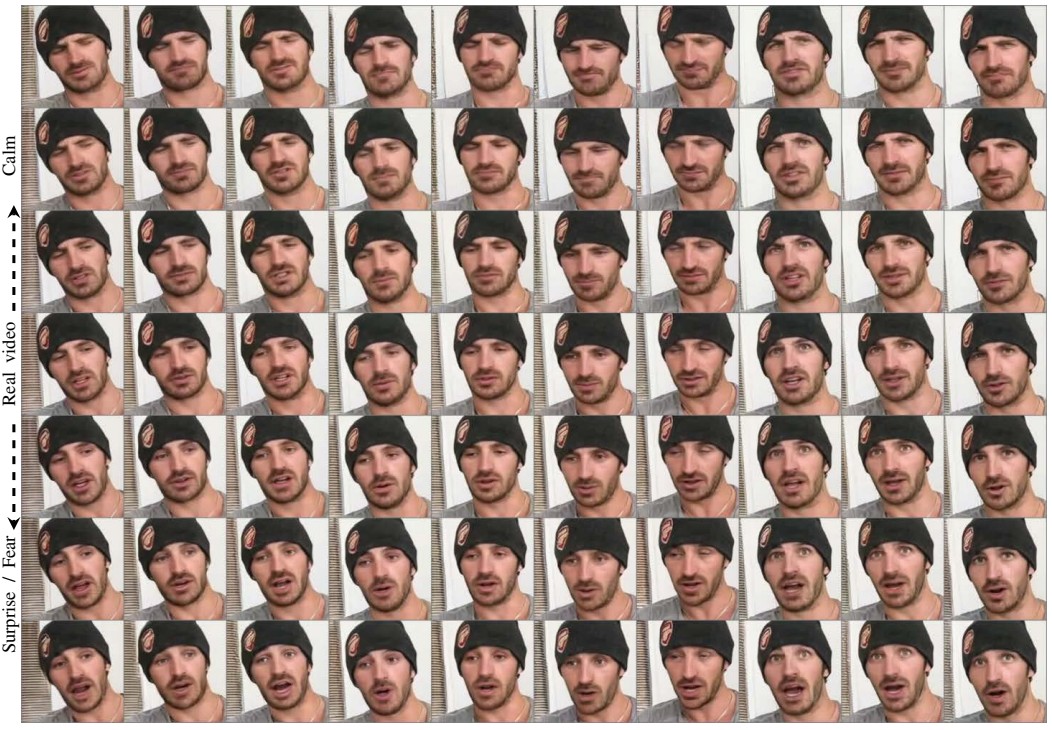

Figure 39: Traversing over facial expressions.

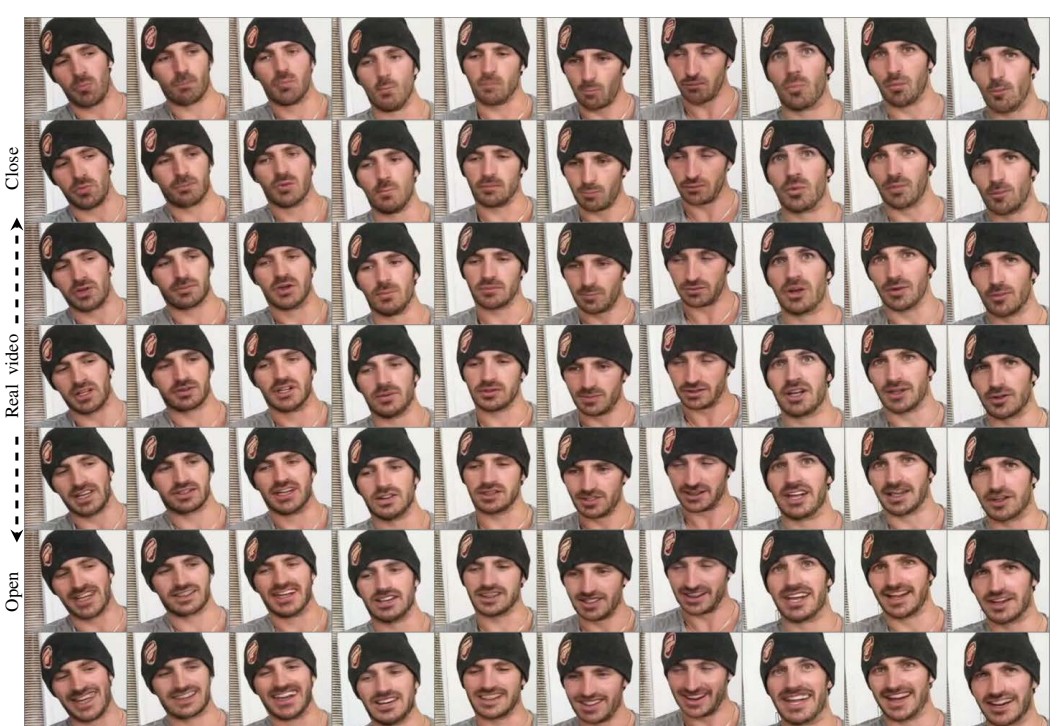

Figure 40: Traversing over mouth openness factor.

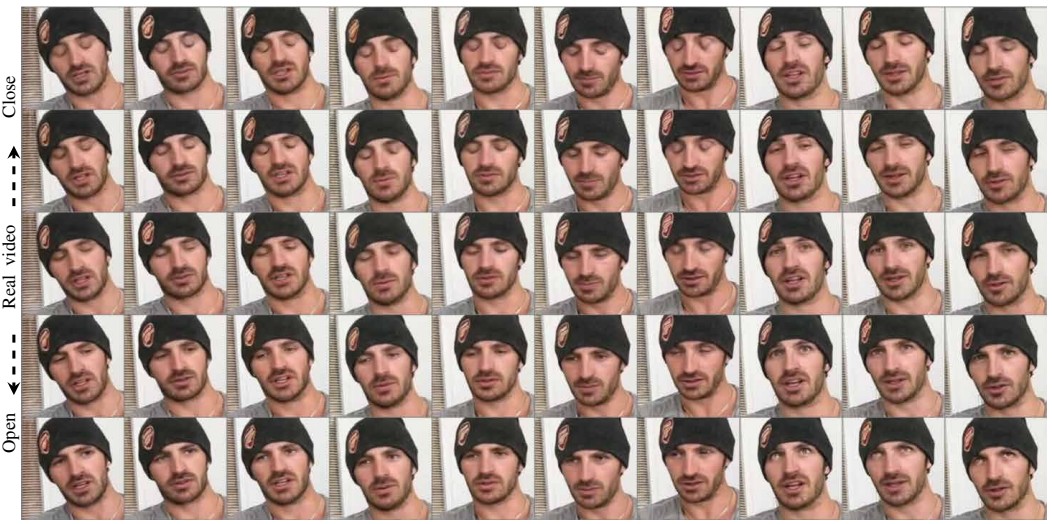

Figure 41: Traversing over eyes openness factor.

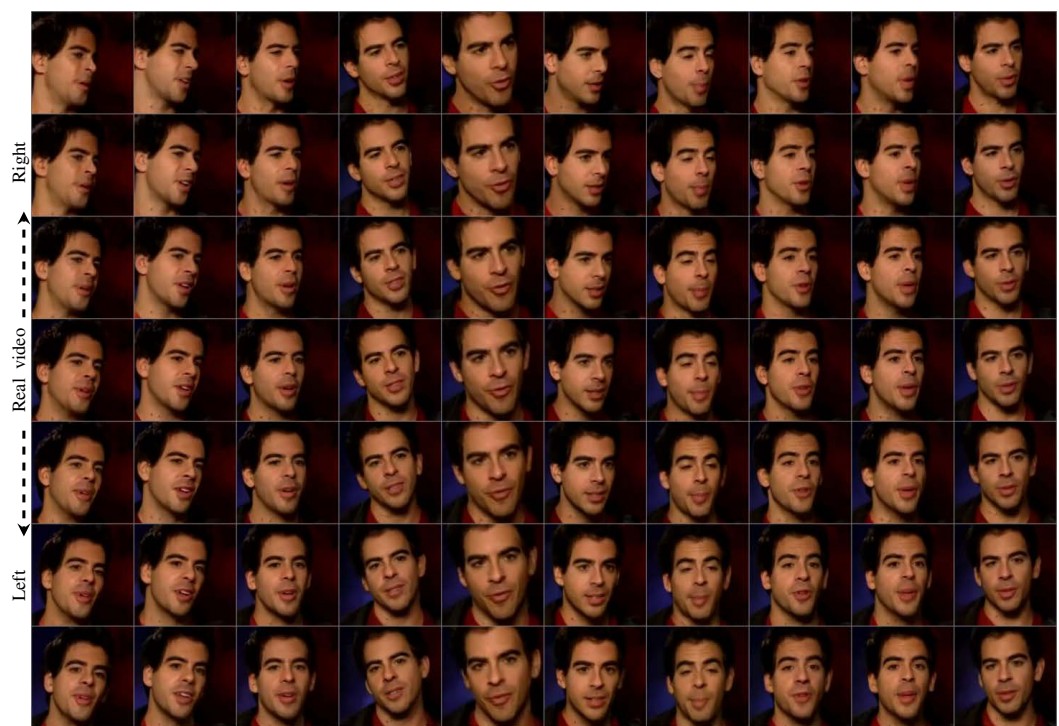

Figure 42: Traversing over a head rotation factor.

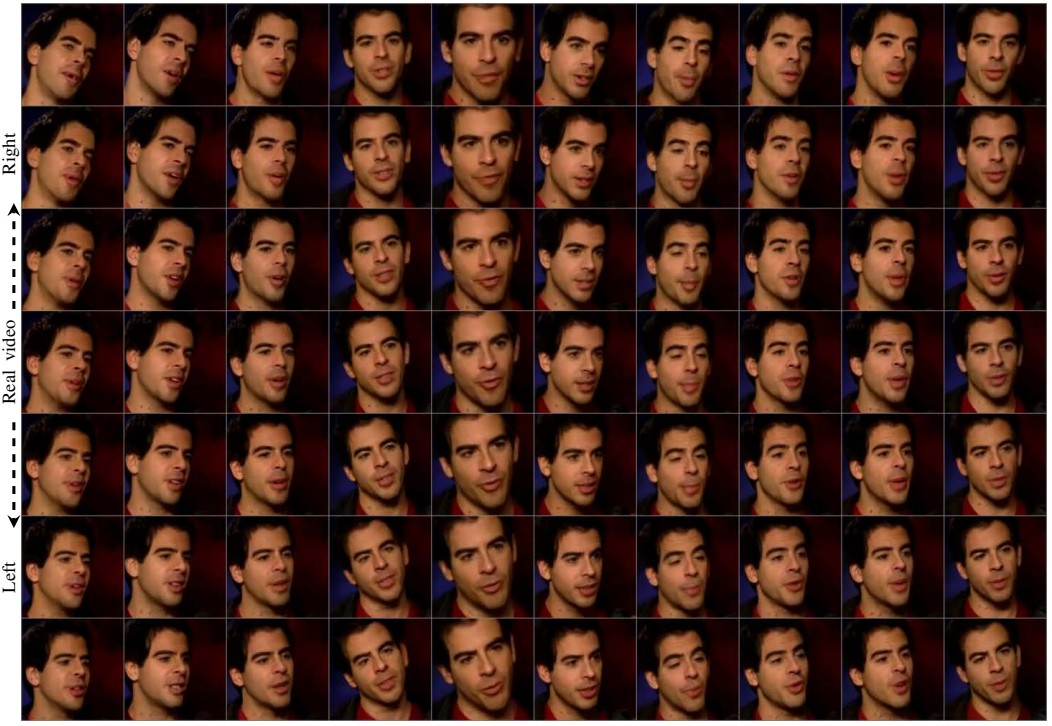

Figure 43: Traversing over various head angles.

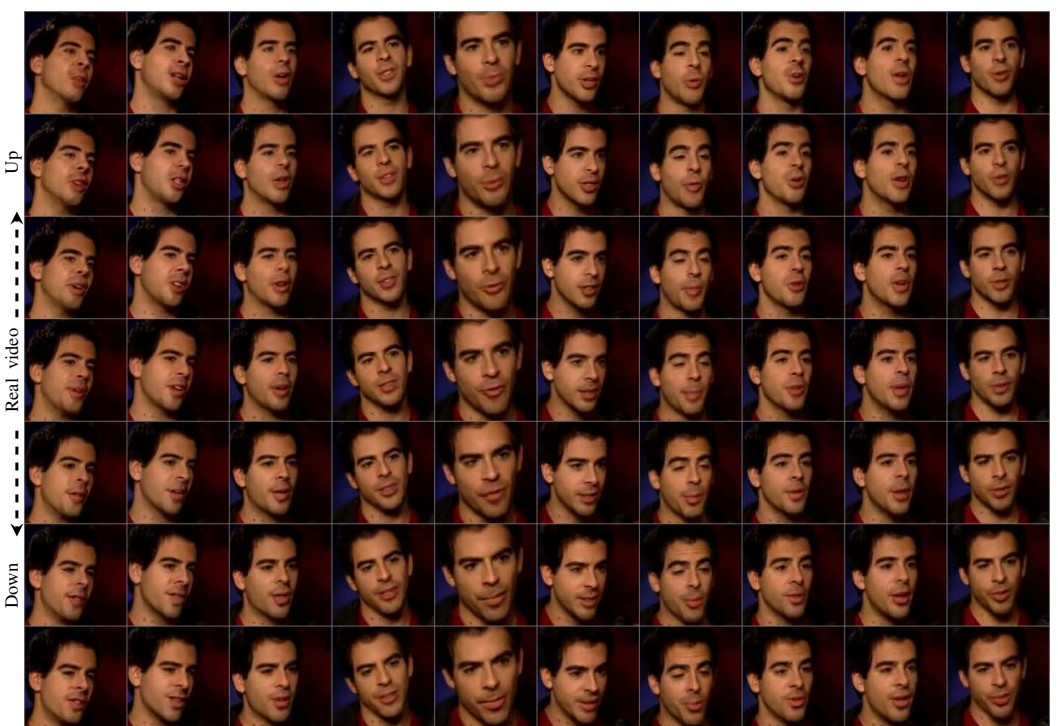

Figure 44: Traversing over up and down head rotations.

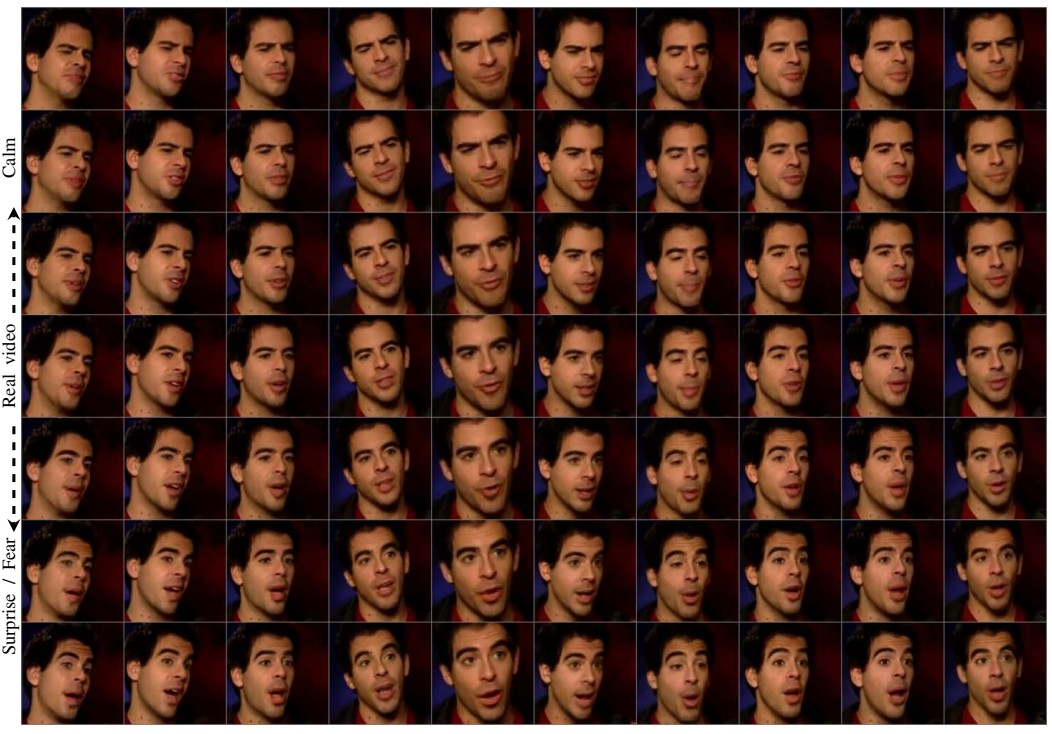

Figure 45: Traversing over facial expressions.

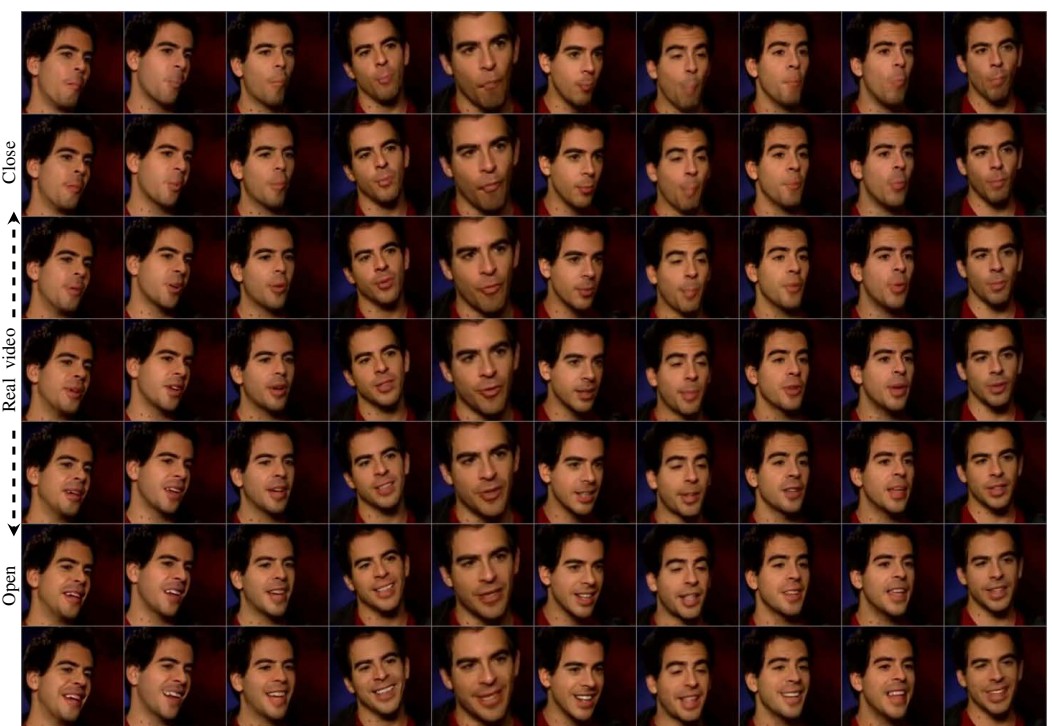

Figure 46: Traversing over mouth openness factor.

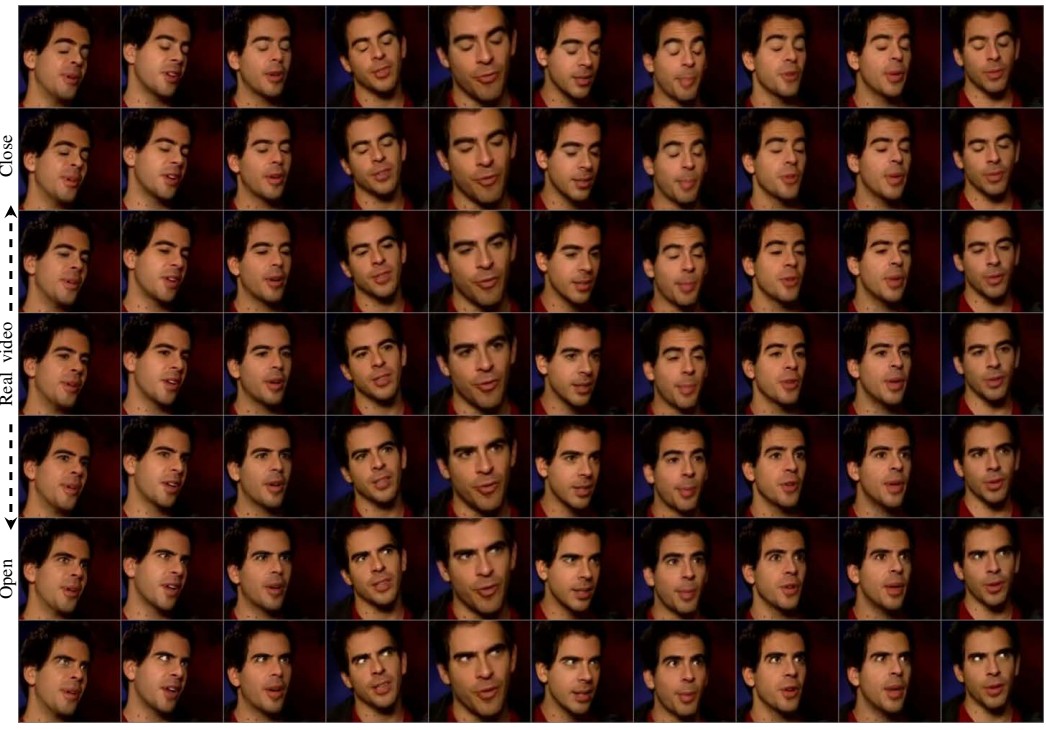

Figure 47: Traversing over eyes openness factor.

