# OpenReview forum: "From Noise to Factors: Diffusion-based Unsupervised Sequential Disentanglement"
_ICLR.cc/2025/Conference — Submitted to ICLR 2025_

### Official Review · Reviewer_GRap · 2024-10-29

**Soundness:** 2
**Presentation:** 3
**Contribution:** 2
**Rating:** 5
**Confidence:** 3

**Summary:**

The paper proposes a decoder framework for the latent spaces within diffusion generative models. By incorporating sequential aware neural networks (LSTM) into the proposed DiffSDA method, it is able to extract some sequential features that allow for video editing tasks.

**Strengths:**

The paper extends the static diffusion space decoders to their dynamic sequential version, which can be applied to video editing tasks. It is an interesting application scenario.

**Weaknesses:**

- Above all, I believe that it is not appropriate to show a video example with gender being modified in the first illustration figure, the authors can choose many other alternative examples to convey the same idea other than this particular case. On top of that, the results shown in this first figure are some qualitative examples of "video editing" (at least to me), and there is a misalignment between what is “disentanglement” and “editing or semantic manipulation tasks” within this context.
- A major weakness of this paper is the lack of a clear problem definition. The abstract and introduction start with an ambitious, high-level goal—representation disentanglement in an unsupervised learning setting—but the focus quickly shifts to generative modeling across a range of architectures, including VAEs, GANs, and DMs. There is a significant gap in explaining how this broad question is formulated within a generative framework.
- Building on my previous point, the authors may risk overclaiming the contribution of the proposed framework. It functions as a post-hoc decoder that applies additional disentanglement to the diffusion latent space, rather than as an unsupervised learning framework that directly addresses disentanglement challenges.
- Even within this diffusion generative context, many recent works have demonstrated the intrinsic ability of diffusion models to learn disentangled features, such as [a,b,c]. The authors fail to discuss and/or compare these related works in the paper.
- I see a strong implicit assumption in the problem formulation in Section 4.1, where the authors assume the data distribution can be factorized into a "state-independent distribution density of static (time-invariant) and dynamic (time-variant) factors," treating them as independent variables. This assumption seems questionable and requires, at a minimum, some justification. Additionally, Eq. (4) is somewhat unclear and unprofessional—what is V? It is mentioned in Line 185 but not defined.


[a] "Diffusion models already have a semantic latent space." ICLR 2023.

[b] “Boundary Guided Learning-Free Semantic Control with Diffusion Models”. NeurIPS 2023.

[c] “Continuous, Subject-Specific Attribute Control in T2I Models by Identifying Semantic Directions.”. 2024

**Questions:**

Please see my detailed Weaknesses section. While there are several minor issues throughout the manuscript, I have highlighted what I consider to be the major concerns. Overall, I believe the paper is not yet ready for publication.

**Details Of Ethics Concerns:**

Although I did not flag this paper for additional ethical review, I believe there are potential ethical concerns, especially given the qualitative examples presented. I am noting this here to raise the authors' awareness.

---

> ### Author Response · Authors · 2024-11-22
> **Response to Reviewer GRap (1/2)**
>
> We sincerely thank you for your detailed review of our paper and for taking the time to provide thoughtful feedback. Your comments have highlighted several important areas for improvement, and we appreciate your constructive critique, which will help us refine both our framework and its presentation. Given the opportunity, we will incorporate the discussions and modifications below into our final version.
>
> * ***Above all, I believe that it is not appropriate to show a video example with gender being modified in the first illustration figure, the authors can choose many other alternative examples to convey the same idea other than this particular case.***
>
> Thank you for raising this concern. We understand the importance of ethical considerations in presenting research, particularly when sensitive attributes such as gender are involved. The example we chose is commonly used in the literature to illustrate algorithmic capabilities and was selected for its effectiveness in demonstrating the functionality of our proposed method [1, 2]. That said, we recognize the potential for ethical sensitivities in such examples and respect your perspective.
>
> To address your concern, we are open to revising the example to avoid any potential sensitivities. If necessary, we can replace it with an alternative example, such as modifying attributes like hairstyle, lighting, or age, which would similarly demonstrate the core idea without raising ethical questions.
>
> Additionally, we would be happy for the ethics committee to review our submission and will align with any recommendations it provides to ensure our work adheres to ethical standards. We appreciate your feedback and remain committed to presenting our research responsibly and thoughtfully.
>
> [1] Preechakul et al. ``Diffusion Autoencoders: Toward a Meaningful and Decodable Representation''. CVPR 2022.
>
> [2] Wang et al. ``InfoDiffusion: Representation Learning Using Information Maximizing Diffusion Models''. ICML 2023.
>
>
> * ***On top of that, the results shown in this first figure are some qualitative examples of "video editing" (at least to me), and there is a misalignment between what is “disentanglement” and “editing or semantic manipulation tasks” within this context.***
>
> The first figure presents results from several challenging downstream tasks applied to real-world datasets. While the figure is not intended to explicitly illustrate the concept of disentanglement, it serves to highlight novel outcomes that are unattainable with existing unsupervised sequential disentanglement methods, showcasing the unique capabilities of our approach.
>
>
> * ***A major weakness of this paper is the lack of a clear problem definition. The abstract and introduction start with an ambitious, high-level goal—representation disentanglement in an unsupervised learning setting—but the focus quickly shifts to generative modeling across a range of architectures, including VAEs, GANs, and DMs. There is a significant gap in explaining how this broad question is formulated within a generative framework.***
>
> A prevalent paradigm in the literature on unsupervised sequential disentanglement (and disentanglement in general) involves framing the problem as a generative modeling task. In this formulation, a data sample $x$ is assumed to be governed by an underlying factor $z$, and the objective is to probabilistically design and implement the corresponding generative distributions. Following the reviewer’s recommendation, we have included a brief discussion in Appendix A, elaborating on the problem definition of unsupervised (sequential) disentanglement through the lens of generative modeling.

---

> ### Author Response · Authors · 2024-11-22
> **Response to Reviewer GRap (2/2)**
>
> * ***Building on my previous point, the authors may risk overclaiming the contribution of the proposed framework. It functions as a post-hoc decoder that applies additional disentanglement to the diffusion latent space, rather than as an unsupervised learning framework that directly addresses disentanglement challenges.***
>
> We emphasize that our framework is not a mere "post-hoc decoder" but a diffusion-based autoencoder explicitly designed to enable unsupervised sequential disentanglement, i.e., factorizing input sequences into static and dynamic representations. Moreover, our approach does not impose `additional disentanglement on the diffusion latent space'. Instead, disentanglement occurs within the latent space of a novel semantic encoder, which is entirely separate from the stochastic (diffusion) encoder and stochastic (diffusion) decoder. Importantly, our framework directly addresses the unsupervised disentanglement challenge posed in this work: given a time series, our method generates a static vector and multiple dynamic vectors in an unsupervised manner, capturing the time-independent and time-dependent semantic aspects of the input sequence.
>
> * ***Even within this diffusion generative context, many recent works have demonstrated the intrinsic ability of diffusion models to learn disentangled features, such as [a,b,c]. The authors fail to discuss and/or compare these related works in the paper.***
>
> Thank you for bringing these papers to our attention and providing us the opportunity to improve the quality of our related works section. While the papers mentioned by the reviewer are indeed relevant to extracting semantics using diffusion models, they differ significantly from our work in several key aspects. Specifically, these works focus on non-sequential data and do not address the decomposition of input sequences into static and dynamic factors of variation. Additionally, these approaches are tailored to image data, whereas our method is modal-agnostic and we demonstrated its effectiveness on both video and audio data. While we briefly mention these works in the related work section, we believe a direct comparison is not necessary, as they address fundamentally different problems from the one tackled in our framework.
>
> * ***I see a strong implicit assumption in the problem formulation in Section 4.1, where the authors assume the data distribution can be factorized into a "state-independent distribution density of static (time-invariant) and dynamic (time-variant) factors," treating them as independent variables. This assumption seems questionable and requires, at a minimum, some justification.***
>
> In the literature on unsupervised sequential disentanglement, a common assumption is the independence of static and dynamic factors. Our work aligns with the literature, where we explicitly assume it in our original submission in Lines 211-212 and posterior modeling in Eq. (5). Moreover, the independence assumption holds in many significant scenarios, offering a robust foundation for studying and developing sequential disentanglement methodologies. Importantly, most existing disentanglement work assumes independence also in the prior model (Eq. (4) in our manuscript), however, we do not make such assumption. While certain approaches extend beyond independence (in prior and posterior modeling), it still remains a key challenge, representing an important frontier in advancing unsupervised sequential disentanglement approaches.
>
> * ***Additionally, Eq. (4) is somewhat unclear and unprofessional—what is V? It is mentioned in Line 185 but not defined.***
>
> We added to Sec. 4 in the revised version the sentence: ``$T$ and $V$ represent the maximum diffusion and sequence times, respectively''. We would be happy to further clarify Eq. (4) if needed.
>
> * ***Although I did not flag this paper for additional ethical review, I believe there are potential ethical concerns, especially given the qualitative examples presented. I am noting this here to raise the authors' awareness.***
>
> Please see our response above.

---

> > ### Comment · Reviewer_GRap · 2024-11-24
> > **Thank you for the author responses**
> >
> > I have reviewed the author responses as well as the comments from other reviewers. Some of my concerns regarding the problem formulation of unsupervised sequential disentanglement have been adequately addressed in the responses, and I am raising my rating from 3 to 5 to reflect this.
> >
> > As a reviewer primarily focused on generative models, I view this work as more of an applied contribution to a specific task that incorporates generative models as part of its framework. However, the work does not offer significant insights or advancements to core generative modeling itself, which limits my ability to assign a higher score. I am also lowering my confidence score to reflect this perspective.

---

> ### Author Response · Authors · 2024-11-24
> **Response to Reviewer GRap**
>
> * ***I have reviewed the author responses as well as the comments from other reviewers. Some of my concerns regarding the problem formulation of unsupervised sequential disentanglement have been adequately addressed in the responses, and I am raising my rating from 3 to 5 to reflect this.***
>
>     ***As a reviewer primarily focused on generative models, I view this work as more of an applied contribution to a specific task that incorporates generative models as part of its framework. However, the work does not offer significant insights or advancements to core generative modeling itself, which limits my ability to assign a higher score. I am also lowering my confidence score to reflect this perspective.***
>
>
> We sincerely thank you for your thoughtful feedback, constructive comments, and for raising the score.
>
> While it is true that unsupervised sequential disentanglement branches from generative modeling, we respectfully disagree with the notion that our work constitutes merely an ``applied contribution'' to the field of generative models. Vanilla generative models such as VAEs, GANs, and diffusion models inherently encode entangled representations, and disentangling them requires distinct theoretical frameworks. Our work advances the theory of sequential disentanglement by introducing a novel probabilistic model grounded in score matching (Eqs. 4-5), which is a significant theoretical contribution in its own right. Furthermore, the implementation of this theoretical model represents another layer of contribution, as it demonstrates the practical realization of the probabilistic framework and enables its application to real-world sequential data. To the best of our knowledge, we are the first to propose an unsupervised sequential disentanglement framework (i.e., model and implementation) grounded in diffusion models.
>
> To clarify, our work does not aim to advance generative modeling itself; rather, it builds upon generative models as the theoretical framework to design, model, and implement disentanglement specifically for sequential information.
>
> We hope this explanation better positions our contribution in the broader context of generative modeling and disentanglement theory. We would be happy to address any remaining concerns or questions you may have.

---

### Official Review · Reviewer_FuiQ · 2024-10-30

**Soundness:** 3
**Presentation:** 3
**Contribution:** 2
**Rating:** 5
**Confidence:** 3

**Summary:**

In this paper, the authors present a diffusion sequential disentanglement autoencoder (DiffSDA) for unsupervised sequential disentanglement. The proposed DiffSDA is based on DiffAE but improves DiffAE from three perspectives. Moreover, the authors introduce three new datasets for the evaluation and verify the effectiveness of the proposed DiffSDA by comparing with several SOTA baselines.

**Strengths:**

1. The paper proposes to deal with a practical, meaningful and challenging problem setting, that is unsupervised sequential disentanglement.
2. The paper presents a diffusion sequential disentanglement autoencoder (DiffSDA) framework for the unsupervised sequential disentanglement problem.
3. The paper conduct experiments on three practical datasets, and demonstrate the effectiveness of the proposed DiffSDA compared with given baselines.

**Weaknesses:**

1. One major concern of the work is on the technical novelty. The disentanglement function is decoupled with the diffusion module but is done in the semantic encoder. However, the way of extracting the dynamic and static information in the semantic structure, i.e., using LSTM to explore the temporal relations and its last hidden to calculate the static one, has been well studied in existing studies, e.g., disentangled sequential VAE and S3VAE. To this reviewer, this work is a conditional diffusion with the extracted dynamic and static feature representations as conditions. In this sense, the technical significance of the proposed DiffSDA is limited.
2. The paper misses some important related works. Using diffusion for animation generation, e.g., Animate Anyone, MagicAnimate, not only transfers face/appearance but also changes the motion to follow the driven sequence. There are also a lot of image2video works that generate (zero-shot) target video sequences using a single target image and a source driving videos. Although these methods do not explicitly use the terminology ‘disentanglement’, they have similar application scenarios with the paper.
3. It is always encouraging to introduce and test new datasets. However, unsupervised sequential disentanglement is a well-defined/studied problem setting, and testing on some benchmark datasets (maybe naïve) is necessary. The authors may refer to the datasets used in disentangled sequential VAE and S3VAE.
4. To show the effectiveness of the disentanglement, it is necessary to show the reconstruction results with only one latent factor, e.g., keeping dynamic one and set the static one as zeros. Moreover, a project page with some video results is encouraged, which clearly helps the readers to understand your results as well as to compare with the other baselines.
5. Since the method is built on DiffAE, the reviewer wonders how the performance of the proposed DiffSDA compared with DiffAE where frame-by-frame swap can be done.
6. To show the superoirty of the proposed method in the fast sampling, complexity analysis or time-cost analyses, e.g. rtf, need to be done.

**Questions:**

Please refer to weakness.

---

> ### Author Response · Authors · 2024-11-12
>
> Dear Reviewer FuiQ,
>
> We would like to bring to your attention that the review provided for our submission appears to pertain to a different submission. We kindly request that you revise your response to accurately reflect the content of our submission. We sincerely appreciate the time and effort you have dedicated to reviewing our work and thank you for your attention to this matter.
>
> Best regards,
> The Authors

---

> > ### Comment · Reviewer_FuiQ · 2024-11-13
> > **Update of the review.**
> >
> > Sorry for the inconvenience caused. I have updated my review.

---

> ### Comment · Area_Chair_DeKD · 2024-11-13
>
> Reviewer FuiQ,
>
> Can you correct your reviews ASAP to allow the authors sufficient time to respond?
>
> AC

---

> ### Author Response · Authors · 2024-11-22
> **Response to Reviewer FuiQ (1/3)**
>
> We sincerely thank you for your detailed and constructive review of our work. We appreciate your recognition of the meaningful and challenging problem setting addressed in our paper, as well as your acknowledgment of the DiffSDA framework and its evaluation across three datasets. Your insights and suggestions have provided valuable direction for improving the technical rigor and presentation of our work. We will update our final revision with the discussions and modifications mentioned below, given the chance.
>
> * ***One major concern of the work is on the technical novelty. The disentanglement function is decoupled with the diffusion module but is done in the semantic encoder. However, the way of extracting the dynamic and static information in the semantic structure, i.e., using LSTM to explore the temporal relations and its last hidden to calculate the static one, has been well studied in existing studies, e.g., disentangled sequential VAE and S3VAE. To this reviewer, this work is a conditional diffusion with the extracted dynamic and static feature representations as conditions. In this sense, the technical significance of the proposed DiffSDA is limited.***
>
> Indeed, our approach for extracting static and dynamic factors aligns with a commonly used strategy in the literature on unsupervised sequential disentanglement. However, the technical novelty of our work extends beyond this component. The primary contribution lies in the design of a diffusion-based probabilistic model tailored for sequential disentanglement, coupled with its implementation through a novel autoencoder framework. Both the theoretical formulation of the probabilistic model and the practical neural network architecture are original and have not been presented in prior work.
>
> * ***The paper misses some important related works. Using diffusion for animation generation, e.g., Animate Anyone, MagicAnimate, not only transfers face/appearance but also changes the motion to follow the driven sequence. There are also a lot of image2video works that generate (zero-shot) target video sequences using a single target image and a source driving videos. Although these methods do not explicitly use the terminology ‘disentanglement’, they have similar application scenarios with the paper.***
>
> Following the reviewer’s suggestion, we have expanded the related work section to discuss the mentioned body of work and clarify the distinctions with respect to our approach. Our method is modal-agnostic, and therefore, it is applicable to any sequential data. To support this claim empirically, we added to the revised version a new experiment on the audio modality, where our approach achieves state-of-the-art performance on the TIMIT speaker identification benchmark in comparison to existing approaches. In contrast, the animation and image2video works mentioned above are primarily designed to the video modality, making their adaptation to other data modalities challenging. Additionally, our approach enables certain multifactor disentanglement capabilities, which may not be achievable in some of the mentioned works. Finally, our model is generative, allowing it to sample novel, unseen examples. These distinctions highlight our approach's broader scope and versatility compared to the mentioned work.

---

> ### Author Response · Authors · 2024-11-22
> **Response to Reviewer FuiQ (2/3)**
>
> * ***It is always encouraging to introduce and test new datasets. However, unsupervised sequential disentanglement is a well-defined/studied problem setting, and testing on some benchmark datasets (maybe naïve) is necessary. The authors may refer to the datasets used in disentangled sequential VAE and S3VAE.***
>
> Following the reviewer’s suggestion, we have expanded our evaluation to include an entirely new modality. Specifically, the revised version incorporates the audio dataset TIMIT, along with its associated downstream task of speaker identification, commonly considered in recent unsupervised sequential disentanglement works. The main other dataset in prior works like DSVAE and S3VAE is Sprites. We would like to clarify our evaluation choices with respect to omitting Sprites. First, our work focuses on real-world video data--a complex and challenging setting that has yet to be thoroughly explored in prior research. Second, the Sprites dataset has reached a saturation point, with recent works consistently achieving 99.9\% accuracy, offering limited scope for further differentiation.
>
>
> | **Method**   | **Static EER ↓** | **Dynamic EER ↑** | **Disentanglement Gap ↑** |
> |--------------|------------------|-------------------|---------------------------|
> | FHVAE        | 5.06%           | 22.77%           | 17.71%                   |
> | DSVAE        | 5.64%           | 19.20%           | 13.56%                   |
> | R-WAE        | 4.73%           | 23.41%           | 18.68%                   |
> | S3VAE        | 5.02%           | 25.51%           | 20.49%                   |
> | SKD          | 4.46%           | 26.78%           | 22.32%                   |
> | C-DSVAE      | 4.03%           | 31.81%           | 27.78%                   |
> | SPYL         | **3.41%**       | 33.22%           | 29.81%                   |
> | DBSE         | 3.50%           | 34.62%           | 31.11%                   |
> | **Ours**     | 4.43%           | **46.72%**       | **42.29%**               |
>
>
> * ***To show the effectiveness of the disentanglement, it is necessary to show the reconstruction results with only one latent factor, e.g., keeping dynamic one and set the static one as zeros.***
>
> We agree with the reviewer that assessing the effectiveness of disentanglement should involve fixing one factor while varying the other. However, we believe this assessment should not rely on using the zero vector. Specifically, due to the nonlinearity of our neural network, the zero vector may correspond to meaningful static or dynamic variations. Furthermore, our modeling framework does not assign any special significance to the zero vector, making its behavior in such scenarios unclear.
>
> To evaluate disentanglement more effectively, we proposed the swap experiment in the paper. In this setup, one factor, such as the dynamic component, is held fixed, while the static component is replaced with that of a different example. The qualitative and quantitative results from these experiments confirm that our approach achieves effective disentanglement.
>
> * ***Moreover, a project page with some video results is encouraged, which clearly helps the readers to understand your results as well as to compare with the other baselines.***
>
> We added a zip container with several video and audio examples, demonstrating the disentanglement capabilities of our work with respect to competitive works.
>
> * ***Since the method is built on DiffAE, the reviewer wonders how the performance of the proposed DiffSDA compared with DiffAE where frame-by-frame swap can be done.***
>
> A direct comparison between our approach and DiffAE is not feasible, as DiffAE is not designed to handle sequential information. Even when applied frame-by-frame, DiffAE does not produce a static and dynamic factorization of the data. Consequently, it is unclear how to perform swaps using DiffAE’s $z_\text{sem}$. Specifically, $z_\text{sem}$ represents a high-dimensional, potentially entangled vector that does not inherently capture the distinct notions of static and dynamic factors of variation.

---

> ### Author Response · Authors · 2024-11-22
> **Response to Reviewer FuiQ (3/3)**
>
> * ***To show the superoirty of the proposed method in the fast sampling, complexity analysis or time-cost analyses, e.g. rtf, need to be done.***
>
>     Thank you for the opportunity to clarify this point. The fast sampling efficiency of our method is considered within the context of diffusion-based approaches. Specifically, our approach leverages the EDM framework [1], which enables high-quality sampling with only 63 neural function evaluations, a significant improvement over the 1000 evaluations required by DDPM [2], as utilized in prior DiffAE works [3, 4]. We hope this explanation clarifies the advantage, and we are happy to include this discussion in the revised version of the manuscript.
>
> [1] Karras et al. ``Elucidating the Design Space of Diffusion-Based Generative Models''. NeurIPS 2022.
>
> [2] Ho et al. ``Denoising Diffusion Probabilistic Models''. NeurIPS 2020.
>
> [3] Preechakul et al. ``Diffusion Autoencoders: Toward a Meaningful and Decodable Representation''. CVPR 2022.
>
> [4] Wang et al. ``InfoDiffusion: Representation Learning Using Information Maximizing Diffusion Models''. ICML 2023.

---

> > ### Comment · Reviewer_FuiQ · 2024-11-25
> > **Comments after the rebuttal**
> >
> > I appreciate the authors' detailed response to my reviews. However, my major concern is not well handled. The reviewer understands that the work uses diffusion, a more powerful generative model, compared with VAE, as the backbone of the overall framework. However the core of disentanglement, that is the structure of extracting the static and dynamic factors, is not significantly novel. Moreover, if taking the paper as the application of the diffsuion model, there are quite a number of related works, as mentioned in my comments, needs to be considered and the current version is not sufficent to support the superority of the proposed method.

---

> ### Author Response · Authors · 2024-11-25
> **Kindly Follow-Up: Reviewer FuiQ**
>
> Dear Reviewer FuiQ, We sincerely thank you for your comprehensive and valuable review, which has significantly helped improve the quality of our paper. As a kind reminder, we have carefully addressed your questions and concerns in our rebuttal response, including performing experiments with new modalities from previous benchmarks and clarifying key points. We hope that our response has provided sufficient clarification. Please let us know if you have any remaining concerns or require additional information.

---

> ### Author Response · Authors · 2024-11-26
> **Response to Reviewer FuiQ**
>
> We thank Reviewer FuiQ for their response.
>
> * ***I appreciate the authors' detailed response to my reviews. However, my major concern is not well handled. The reviewer understands that the work uses diffusion, a more powerful generative model, compared with VAE, as the backbone of the overall framework. However the core of disentanglement, that is the structure of extracting the static and dynamic factors, is not significantly novel.***
>
> We understand that the reviewer focuses on the module for extracting static and dynamic factors using LSTM as the central technical novelty of our work. However, we respectfully argue that the assertion regarding a lack of significant novelty in our disentanglement approach does not fully capture the scope and impact of our contributions. In particular, subsequent works following DSVAE, such as S3VAE, C-DSVAE, SPYL, and others, have employed similar static-dynamic extraction modules with relatively minor variations.
>
> Our manuscript introduces a novel approach to unsupervised sequential disentanglement, offering advancements in both probabilistic modeling and practical implementation. Specifically, our theoretical framework (Equations (4)–(5)) diverges from existing methods by formalizing disentanglement within a diffusion generative modeling framework. This approach unlocks two key benefits: (1) Unlike most prior methods, we impose no prior constraints on the latent representations $s$ and $d$, allowing us to learn more expressive and flexible representations.
>     (2) In contrast to DSVAE, S3VAE, C-DSVAE, SPYL, and DBSE, our model employs a simple loss function with a single loss term, eliminating the need for hyperparameter balancing—a major challenge in previous works.
>
> To the best of our knowledge, our work is the first to model and implement unsupervised sequential disentanglement using diffusion models, a direction that has not been previously explored. This innovation in both modeling and implementation forms the foundation of our contributions.
>
> Moreover, the assertion that our work merely replaces the backbone of existing models is fundamentally incorrect and oversimplifies the challenges we addressed. Even if such a substitution were attempted, it would not be straightforward. For instance, during the early stages of our research, we investigated NVAE as a potential candidate for disentanglement on high-quality real-world data. However, as discussed in our introduction, NVAE’s hierarchical and high-dimensional latent representation posed significant challenges for disentanglement. Despite our efforts to apply standard static-dynamic extraction modules within this framework, the results ultimately fell short, with poor performance observed in both reconstruction quality and disentanglement. This highlights the inherent difficulty of achieving effective disentanglement in such settings. Moreover, it underscores the necessity of meticulous design and modeling to develop an effective disentanglement framework, as demonstrated by our approach, which performs robustly on real-world data.
>
> We kindly ask the reviewer to reconsider the novelty and positioning of our work, particularly its unique contributions to the field of unsupervised sequential disentanglement.
>
>
> * ***Moreover, if taking the paper as the application of the diffsuion model, there are quite a number of related works, as mentioned in my comments, needs to be considered and the current version is not sufficent to support the superority of the proposed method.***
>
> We do not consider our work to be merely an application of diffusion models; rather, it represents a novel framework for sequential disentanglement that builds upon the formalism of diffusion models. Consequently, the most relevant comparisons are to other unsupervised sequential disentanglement approaches. To the best of our knowledge, no prior work has explored diffusion-based unsupervised sequential disentanglement, and in our manuscript, we benchmark against state-of-the-art sequential disentanglement methods based on VAEs.
>
> While recent video animation methods may achieve better results on specific tasks, our work stands out in three key aspects: (1) we provide a disentangled representation that enables a broad range of tasks beyond those included in our evaluation, (2) we demonstrate multifactor disentanglement in video data, a capability not addressed by many of the animation methods cited, and (3) our approach is modality-agnostic, making it adaptable to a wide variety of data modalities. For example, during the rebuttal phase, we showcased its effectiveness on audio data. In contrast, the animation methods referenced by the reviewer are highly specialized and require significant modifications to their modeling, architecture, and foundational design to extend to new modalities.

---

### Official Review · Reviewer_3qxX · 2024-11-04

**Soundness:** 3
**Presentation:** 3
**Contribution:** 2
**Rating:** 5
**Confidence:** 2

**Summary:**

The paper is concerned with the problem of learning disentangled generative models for sequential (e.g. video) data. It proposes a method based on the Diffusion VAE framework that splits the latents into two groups - static (does not vary with frame) and dynamic (varies per frame). The proposed method is benchmarked on a number of (single) person-centric datasets.

**Strengths:**

* Generally well-written and clear.
* Strong empirical results on the chosen benchmarks.
* The proposed solution to obtaining disentangled representations is interesting in that it relies on the structure of the model rather than auxiliary losses.

**Weaknesses:**

**Major**
* The paper touches on the problem of disentangled representation for sequential data in general. However, the presented experiments are limited - they only make use of simple (single person) video data. This does not give a fair picture of the methods usefulness and applicability because (i) it targets only a single modality; and (ii) because even within this modality the model targets very structured datasets (e.g. single view, single object).

**Minor**
* The Introduction section gives a skewed picture of the state of image generation with VAEs and GANs (e.g. citing works from 2016 and 2018) to support the claims of GANs being unstable and VAEs blurry. A large body of work has been introduced since the introduction of these methods that makes GAN training very stable in practice, and improves VAE generation quality.
* Please double-check citations. For example, line 651 has an incorrect author name.

**Questions:**

* It would be great to see the method shine in more (real-world) domains, for example audio (e.g. speaker vs content) disentanglement, multi-object video (controlling aspects of objects as is shown in this work, but then on the object rather than video level), multi-view video (e.g. controlling viewpoint)

Relevant works:
* View point synthesis: "DORSAL: DIFFUSION FOR OBJECT-CENTRIC REPRESENTATIONS OF SCENES"
* Object disentanglement: "Neural Assets: 3D-Aware Multi-Object Scene Synthesis with Image Diffusion Models"
* Audio benchmark: "Sample and Predict Your Latent: Modality-free Sequential Disentanglement via Contrastive Estimation"

---

> ### Author Response · Authors · 2024-11-22
> **Response to Reviewer 3qxX (1/2)**
>
> We sincerely thank the reviewer for their thorough review, which highlights important points and raises concerns that helped us enhancing the quality of the paper. We greatly appreciate the acknowledgment of the paper's clarity, the comprehensiveness of the empirical results, and the interest in the proposed solution. Below, we address the raised concerns.
>
> * ***The paper touches on the problem of disentangled representation for sequential data in general. However, the presented experiments are limited - they only make use of simple (single person) video data. This does not give a fair picture of the methods usefulness and applicability because (i) it targets only a single modality; and (ii) because even within this modality the model targets very structured datasets (e.g. single view, single object)***
>
> In our original submission, we focused exclusively on a single modality (videos), but we emphasize that our approach is modality-agnostic, as it does not rely on any specific assumptions about the underlying data modality. However, we acknowledge the limitation of evaluating only on video data in our initial experiments. To address this within the constraints of the rebuttal period, we conducted an additional experiment to incorporate a new modality into our evaluation.  Specifically, we extended our experiments to audio data by utilizing the standard TIMIT benchmark, where the task is speaker identification. The results, now included in Sec. 5.5 of the revised version, demonstrate state-of-the-art performance on this benchmark in terms of the disentanglement gap when compared to existing SOTA methods. Additionally, we have included a zip file of audio samples for qualitative comparison with results from [1] (publicly available), further highlighting the high-quality performance of our approach.
>
> Regarding structured datasets, we agree that current unsupervised sequential disentanglement benchmarks predominantly focus on structured data, which is inherently simpler to model. We also fully concur that tackling unstructured data, such as scenarios involving multiple views or multiple objects, remains an open and critically important challenge in the field of sequential disentanglement. To address this, we have added a discussion in Sec. 6 of the main text highlighting this challenge and its broader significance to the research community. We also intend to incorporate this direction into our future research agenda. That said, we would like to emphasize that even on structured data, as demonstrated in our paper, existing state-of-the-art methods often struggle to effectively process real-world information. While our model achieves significant progress compared to these methods, it also reveals that there is still considerable room for improvement, even within the domain of structured data.
>
> [1] Han et al. ``Disentangled Recurrent Wasserstein Autoencoder''. ICLR 2021.
>
> * ***The Introduction section gives a skewed picture of the state of image generation with VAEs and GANs (e.g. citing works from 2016 and 2018) to support the claims of GANs being unstable and VAEs blurry. A large body of work has been introduced since the introduction of these methods that makes GAN training very stable in practice, and improves VAE generation quality.***
>
> Thank you for bringing this to our attention. We have revised the introduction by omitting the discussion of unstable training and mode collapse in GANs, instead focusing on the superior quality of results achieved by diffusion models compared to GANs. Regarding VAEs, we would like to highlight that our original submission acknowledges the existence of high-quality VAEs (now marked in red in the revised version). However, we also point out that the hierarchical nature of the latent spaces in these approaches can complicate disentanglement, an aspect noted also in existing work [1]. Additionally, we have revised the introduction to emphasize that even advanced VAE approaches are still outperformed by diffusion-based methods.
>
> [1] Preechakul et al. ``Diffusion Autoencoders: Toward a Meaningful and Decodable Representation''. CVPR 2022.
>
> * ***Please double-check citations. For example, line 651 has an incorrect author name.***
>
> Thank you for highlighting this important issue. We have addressed it in the revised version and have thoroughly reviewed all other citations to ensure their accuracy and correctness with the text.

---

> ### Author Response · Authors · 2024-11-22
> **Response to Reviewer 3qxX (2/2)**
>
> * ***It would be great to see the method shine in more (real-world) domains, for example audio (e.g. speaker vs. content) disentanglement, multi-object video (controlling aspects of objects as is shown in this work, but then on the object rather than video level), multi-view video (e.g., controlling viewpoint)***
>
> Following your suggestion, we have evaluated our method on a standard sequential disentanglement benchmark for audio, and the results are presented below. As highlighted in our previous comment, our method achieves a state-of-the-art disentanglement gap compared to existing approaches, showcasing its ability to adapt effectively to diverse real-world modalities.
>
> TIMIT results:
>
> | **Method**   | **Static EER ↓** | **Dynamic EER ↑** | **Disentanglement Gap ↑** |
> |--------------|------------------|-------------------|---------------------------|
> | FHVAE        | 5.06%           | 22.77%           | 17.71%                   |
> | DSVAE        | 5.64%           | 19.20%           | 13.56%                   |
> | R-WAE        | 4.73%           | 23.41%           | 18.68%                   |
> | S3VAE        | 5.02%           | 25.51%           | 20.49%                   |
> | SKD          | 4.46%           | 26.78%           | 22.32%                   |
> | C-DSVAE      | 4.03%           | 31.81%           | 27.78%                   |
> | SPYL         | **3.41%**       | 33.22%           | 29.81%                   |
> | DBSE         | 3.50%           | 34.62%           | 31.11%                   |
> | **Ours**     | 4.43%           | **46.72%**       | **42.29%**               |
>
>
> * ***Relevant works: View point synthesis: "DORSAL: DIFFUSION FOR OBJECT-CENTRIC REPRESENTATIONS OF SCENES"; Object disentanglement: "Neural Assets: 3D-Aware Multi-Object Scene Synthesis with Image Diffusion Models"; Audio benchmark: "Sample and Predict Your Latent: Modality-free Sequential Disentanglement via Contrastive Estimation"***
>
> Thank you for bringing these references to our attention. The first two works are non-sequential and primarily focus on scene synthesis, while our approach specifically addresses unsupervised sequential disentanglement and is broadly applicable to any single-object sequential information. We have briefly mentioned these two works in the revised version for completeness. The third work, however, is highly relevant to our paper and was cited and discussed in our original submission. Additionally, we conduct similar evaluations on the MUG and TIMIT datasets to provide a comparable analysis.

---

> ### Author Response · Authors · 2024-11-25
> **Kindly Follow-Up: Reviewer 3qxX**
>
> Dear Reviewer 3qxX, Thank you for your review and valuable feedback, which have greatly contributed to enhancing the quality of our paper. As a kind reminder, we recently submitted a thorough rebuttal addressing your questions, including conducting experiments with new modalities. We hope that our response has provided sufficient clarification. Please let us know if you have any further inquires or require additional information.

---

> > ### Comment · Reviewer_3qxX · 2024-12-02
> > **Thank you**
> >
> > Dear authors.
> >
> > Thank you for a thorough rebuttal. It is great to see the additional results on the speaker disentanglement task; and to see that your method also shines on it.
> >
> > I am also grateful for the additional discussion both - here and in the updated paper - on the relationship between your work and works tackling unstructured data, and on the relationship to more recent generative modeling literature.

---

> > > ### Author Response · Authors · 2024-12-03
> > > **Response**
> > >
> > > Thank you, for your thoughtful response. We are pleased that we were able to address your concerns during the rebuttal phase. As the rebuttal phase is now coming to an end, we would greatly appreciate your reconsideration of the score in light of the clarifications provided. Thank you once again for your time and effort.

---

### Official Review · Reviewer_4tn8 · 2024-11-07

**Soundness:** 3
**Presentation:** 3
**Contribution:** 3
**Rating:** 6
**Confidence:** 4

**Summary:**

The paper proposes a probabilistic model for disentangling static and dynamic features from a series of videos, expanding on the idea of DiffAE. Specifically, the authors expand DiffAE to denoising a series of images, and factorize $z_{sem}$ to $s$ and $d$ which stands for static and dynamic features. A simplified score matching object is also utilized and empirically shows reasonable results for reconstruction. Moreover, the authors experimented on various disentanglement-oriented experiments including conditional/unconditional and zero-shot swapping, and also latent probing.

**Strengths:**

1. The PGM factorization is natural and elegant, and the network design are simple but effective.
2. Quantitative results shows good reconstruction and swapping. Qualitative results also demonstration good disentanglement and generation capacity.

**Weaknesses:**

1. Experiments are mostly shown on short sequence face/motion datasets, where the static feature is appearance and dynamic feature being the keypoints/expressions. Under the framwork that the author proposed, ideally longer sequences should facilitates the disentanglement of $s$ and $d$.
2. It's not clear if the effect of disentanglement comes mostly from the factorization, or from the fact that $d$ has small size and $s$ has larger size.

**Questions:**

1. It's not required, but would be good if the authors include the derivation of the training objective from the score matching objectives.
2. Would like to see the comparison and results on longer sequences.

---

> ### Author Response · Authors · 2024-11-22
> **Response to Reviewer 4tn8 (1/2)**
>
> Thank you for your thoughtful review and for highlighting the strengths of our work, including the elegant factorization in our probabilistic model, effective network design, and strong disentanglement results. Your feedback provides valuable guidance for improving our work. Given the chance, we will incorporate the discussions and modifications below into a final version.
>
>
> * ***Experiments are mostly shown on short sequence face/motion datasets, where the static feature is appearance and dynamic feature being the keypoints/expressions. Under the framwork that the author proposed, ideally longer sequences should facilitates the disentanglement of $s$ and $d$.***
>
> First, to address the reviewers' concern that our experiments primarily focus on short sequences from face and motion datasets, as well as similar concerns raised by other reviewers, we perform additional experiments to demonstrate our model's state-of-the-art ability to disentangle audio data. These results are included in the revised version of our paper.
>
> Second, regarding the use of short sequences for video data, this limitation arises due to the computational cost of training on longer sequences. For instance, VideoMAE V2 [1] trains on 16-frame sequences for over a week using 64 A100 GPUs. Scaling models to handle longer sequences remains an open challenge in the literature [2, 3, 4]. Methods such as [2] address this issue by employing state-space models, which scale to 60-frame inputs. However, it is important to note that scaling to longer sequences was not the focus of our work.
>
> Finally, we agree with the reviewers that longer sequences could intuitively facilitate disentanglement, as they provide more information about the sample. For example, in videos, additional frames allow static content to be better captured by $s$ rather than $d$. However, processing longer sequences remains computationally demanding, as discussed above, and is therefore left as future work.
>
> TIMIT results:
>
> | **Method**   | **Static EER ↓** | **Dynamic EER ↑** | **Disentanglement Gap ↑** |
> |--------------|------------------|-------------------|---------------------------|
> | FHVAE        | 5.06%           | 22.77%           | 17.71%                   |
> | DSVAE        | 5.64%           | 19.20%           | 13.56%                   |
> | R-WAE        | 4.73%           | 23.41%           | 18.68%                   |
> | S3VAE        | 5.02%           | 25.51%           | 20.49%                   |
> | SKD          | 4.46%           | 26.78%           | 22.32%                   |
> | C-DSVAE      | 4.03%           | 31.81%           | 27.78%                   |
> | SPYL         | **3.41%**       | 33.22%           | 29.81%                   |
> | DBSE         | 3.50%           | 34.62%           | 31.11%                   |
> | **Ours**     | 4.43%           | **46.72%**       | **42.29%**               |
>
>
>
>
> [1] "VideoMAE V2: Scaling Video Masked Autoencoders with Dual Masking," Limin Wang et al.
>
> [2] "Long Movie Clip Classification with State-Space Video Models," Md Mohaiminul Islam and Gedas Bertasius.
>
> [3] "A Simple Recipe for Contrastively Pre-training Video-First Encoders Beyond 16 Frames," Pinelopi Papalampidi et al.
>
> [4] "MeMViT: Memory-Augmented Multiscale Vision Transformer for Efficient Long-Term Video Recognition," Chao-Yuan Wu et al.

---

> ### Author Response · Authors · 2024-11-22
> **Response to Reviewer 4tn8 (2/2)**
>
> * ***It's not clear if the effect of disentanglement comes mostly from the factorization, or from the fact that $d$ has small size and $s$ has larger size.***
>
>
> We would like to clarify this point. In our original submission, we discussed disentanglement and its emergence by writing below Eq. (8) the following: ``While our loss in Eq.~8 does not include auxiliary terms, it promotes disentanglement due to two main reasons: i)  the static factor $s_0$ is shared across $\tau$ , and thus it will not hold dynamic information, and ii)  the dynamic factors $d_0^\tau \in \mathbb{R}^k$ are low-dimensional (i.e., $k$ is small), making it difficult for $d_0^\tau$ to store static features.''
>
> Thus, the disentanglement effect arises from the factorization mechanism, which is driven by two primary reasons. Specifically: 1. The static factor $s_0$ is shared across all frames in the sequence. Since all frames are conditioned on the same $s_0$, it is impossible to reconstruct temporal dynamics using $s_0$ alone, as no variations across time would be captured. This promotes $s_0$ to exclusively represent static information. Importantly, each frame is then reconstructed using both $s_0$ and its corresponding dynamic vector $d_0^\tau$, with the dynamic component $d_0^\tau$ responsible for capturing temporal variations needed for reconstruction. 2. To further facilitate disentanglement and prevent $d_0^\tau$ from encoding static information, we deliberately constrain $d_0^\tau$ to be low-dimensional. This encourages $d_0^\tau$ to focus solely on capturing temporal variations, as its limited capacity restricts it from representing too many semantic features.
>
> We hope this explanation clarifies that the disentanglement stems from the overall factorization design rather than solely from the dimensional differences between $s$ and $d$. We would be happy to include this discussion in the revised version.
>
>
> * ***It's not required, but would be good if the authors include the derivation of the training objective from the score matching objectives.***
>
> Thank you for pointing this out. While we strive for completeness in our manuscript, we believe that including the derivation of the training objective would primarily repeat well-established results in the diffusion methods literature. Specifically, the general optimization objective presented in Eq. (6) of our submission is a standard result in score matching [1] and is not unique to our approach. Similarly, the practical training objective detailed in Eq. (8) follows a recent result from EDM [2], which is thoroughly derived in their Eqs. (104--112), with additional supporting derivations in Eqs. (40--50). The key distinction in our approach lies in its usage of disentangled representations, $z_0^\tau$, as an additional input to $F_\theta$, which is absent in EDM’s formulation. However, this modification is intrinsic to $F_\theta$ and does not alter the rigorous derivation. A redefinition of the conditioning input for $F_\theta$ ensures compatibility with the original derivation in [2].
>
> [1] Vincent. ``A connection between score matching and denoising autoencoders''. Neural Computation, 2011.
>
> [2] Karras et al. ``Elucidating the Design Space of Diffusion-Based Generative Models''. NeurIPS 2022.
>
>
> * ***Would like to see the comparison and results on longer sequences.***
>
> As explained above, training our model on longer sequences is computationally intensive. To address this, we conduct zero-shot experiments on longer sequences from the CelebV-HQ dataset. In these experiments, we fix the static features while sampling the dynamic features to compute the AED score. Similarly, we freeze the dynamic features while sampling the static features to compute the AKD score. The table presents results for different sequence lengths of 5, 10, 20, and 40 frames as input.
>
> Notably, even when increasing the sequence length to much longer videos, such as 40 frames, our model substantially outperforms previous work. However, there is a slight degradation in the AKD metric. We believe, as suggested by the reviewer, that training on longer sequences could further improve disentanglement and enhance robustness across varying sequence lengths. We leave this exploration for future research.
>
> Finally, we attach a zip container with several longer video and audio examples, demonstrating the disentanglement capabilities of our work with respect to competitive works.
>
> | Length         | AED↓ (static frozen) | AKD↓ (dynamics frozen) |
> | -------------- | -------------------- | ---------------------- |
> | 5              | 0.561                | 5.9143                 |
> | 10             | 0.546                | 7.2723                 |
> | 20             | 0.523                | 9.5124                 |
> | 40             | 0.514                | 10.1940                |
> | 10 (DBSE)    | 0.766                | 38.362                 |

---

> ### Author Response · Authors · 2024-11-25
> **Kindly Follow-Up: Reviewer 4tn8**
>
> Dear Reviewer 4tn8, We highly appreciate your time and thank you for your detailed and constructive review, which has significantly improved our paper. As a kind reminder, we recently submitted a detailed rebuttal addressing your concerns, including experiments with longer sequences. We hope that our response has provided sufficient clarification. Please let us know if you have any further questions or require additional information.

---

> > ### Comment · Reviewer_4tn8 · 2024-11-27
> >
> > I appreciate the authors' detailed responses and additional experiments that addressed some concerns on experimentation with longer sequences, which adds to the completeness of the paper. Tehrefore, I will maintain my score and confidence and recommend for boarderline accept.

---

### Meta-Review · Area_Chair_DeKD · 2024-12-22

**Metareview:**

This paper proposes a diffusion autoencoder based approach for unsupervised sequential disentanglement. It conducts evaluations on video (as well as audios, included during rebuttal) and shows promising results. Reviewers acknowledge that the method is valid and results are strong, however several concerns are raised. In particular, there are questions around the applicability of the method to realistic settings, as well as the positioning of this work in the context of the diffusion generative modeling literature. After going through the paper and rebuttal, the AC shares similar views to the reviewers. The approach seems logical, but is on the incremental side in terms of novelty; also, there is a large overlap between the proposed method and diffusion model variants that focus on semantic editing and related matters. The AC believes that the authors need to carefully position their work and thoroughly discuss and potentially compare against the diffusion modeling literature.

**Additional Comments On Reviewer Discussion:**

The authors provided additional experiments on audios to address some of the generality concerns, and this is much appreciated by the reviewers and the AC. However, concerns regarding novelty and positioning agains diffusion modeling methods still remain.

---

### Decision · Program_Chairs · 2025-01-22

Reject